# Relaxed Recursive Transformers: Effective Parameter Sharing with Layer-wise LoRA

**Sangmin Bae**[1*], **Adam Fisch**[2], **Hrayr Harutyunyan**[3], **Ziwei Ji**[3], **Seungyeon Kim**[2], **Tal Schuster**[2†]

[1]KAIST AI    [2]Google DeepMind    [3]Google Research
bsmn0223@kaist.ac.kr, talschuster@google.com

## Abstract

Large language models (LLMs) are expensive to deploy. Parameter sharing offers a possible path towards reducing their size and cost, but its effectiveness in modern LLMs remains fairly limited. In this work, we revisit "layer tying" as form of parameter sharing in Transformers, and introduce novel methods for converting existing LLMs into smaller "Recursive Transformers" that share parameters across layers, with minimal loss of performance. Here, our Recursive Transformers are efficiently initialized from standard pretrained Transformers, but only use a single block of unique layers that is then repeated multiple times in a loop. We further improve performance by introducing Relaxed Recursive Transformers that add flexibility to the layer tying constraint via depth-wise low-rank adaptation (LoRA) modules, yet still preserve the compactness of the overall model. We show that our recursive models (e.g., recursive Gemma 1B) outperform both similar-sized vanilla pretrained models (such as TinyLlama 1.1B and Pythia 1B) and knowledge distillation baselines—and can even recover most of the performance of the original "full-size" model (e.g., Gemma 2B with no shared parameters). Finally, we propose Continuous Depth-wise Batching, a promising new inference paradigm enabled by the Recursive Transformer when paired with early exiting. In a theoretical analysis, we show that this has the potential to lead to significant (2-3$\times$) gains in inference throughput.

## 1 Introduction

Efficient deployment of large language models (LLMs) demands a balance between performance and resources (Raposo et al., 2024; Leviathan et al., 2023; Rivière et al., 2024; Wan et al., 2024; Zhou et al., 2024). While larger models with more parameters consistently demonstrate superior performance (Rosenfeld et al., 2020; Rae et al., 2021; Hoffmann et al., 2022), their substantial memory and computational demands are expensive (Pope et al., 2023). Parameter sharing approaches (e.g. Dehghani et al., 2019; Xia et al., 2019; Lan et al., 2020; Takase & Kiyono, 2023), wherein weights are reused across model layers, can lower these costs by reducing memory footprint, and thereby allow for the use of fewer (or lower-grade) accelerators, or larger batch sizes for better throughput. While parameter sharing has shown encouraging capabilities in previous work (Lan et al., 2020; Giannou et al., 2023), its application to modern LLMs has yielded limited reported success.

In this work, we revisit parameter sharing for LLMs, and propose novel methodologies to *convert* existing, unshared models into smaller, and more efficient, Recursive Transformers. These models use a single block of unique layers that are recursively reused across multiple loops, yet still achieve impressive performance relative to their reduced size. To mitigate the potential performance degradation associated with parameter sharing, we first initialize the shared block of layers based on the original model's pre-trained parameters, and then finetune the resulting recursive model for a limited number of "uptraining" steps. Importantly, we show that our initialization strategies allow us to achieve strong performance with minimal training time. This is aligned with observations that model compression techniques such as layer skipping (Zhang et al., 2024a; Zeng et al., 2023; Fan et al., 2020; Elhoushi et al., 2024), pruning (Frankle & Carbin, 2019; Ramanujan et al., 2020) or nesting (Devvrit et al., 2023) can preserve surprisingly high performance—further motivating our approach of compressing models to more compact yet performant architectures (here, repeated layers with low-rank adapters).

---

*Work done during an internship at Google DeepMind. [†]Corresponding author.

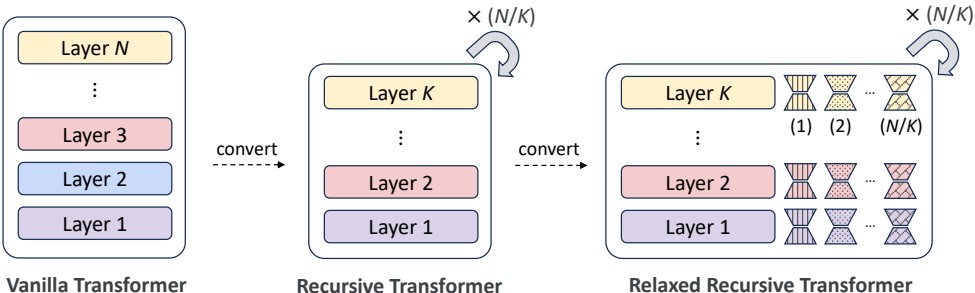

Figure 1: Overview of the conversion from a vanilla $N$-layer Transformer to a Recursive Transformer with $N/K$ blocks of $K$ shared layers. The Recursive Transformer is obtained by repeating a single block of $K$ layers multiple times, resulting in a looped architecture. The Recursive Transformer can also be converted into a Relaxed Recursive Transformer by adding layer-specific LoRA modules. This preserves many of the advantages of weight sharing, but also allows for better performance.

As depicted in Figure 1, we further propose the Relaxed Recursive Transformer, an extension of the Recursive Transformer in which the weight tying across repeated layer blocks is slightly relaxed through the incorporation of multiple layer-specific, low-rank adaptation (LoRA) modules (Hu et al., 2022). Despite its simplicity, this strategy offers several non-trivial advantages. First, it allows for low-rank deltas between shared layers, while only adding minimal overhead. Second, the rank of the LoRA matrices can be adjusted to control the degree of relaxation, which directly influences model capacity. Furthermore, since the relaxed model has the same overall shape as the original Transformer, we can efficiently initialize LoRA modules via truncated Singular Value Decomposition (Hansen, 1987) on the residual matrices between the original layer weights and the shared layer weights. Hence, the rank values serve as a pivotal hyperparameter, enabling the Relaxed Recursive Transformer to seamlessly transition between the two extremes of the vanilla and Recursive Transformer architectures.

While the primary focus of this paper lies in how to formulate and train Recursive Transformers, we also highlight their potential to achieve significant throughput gains via a new batched inference paradigm, Continuous Depth-wise Batching, that their recursive nature enables. Prior work introduced continuous sequence-wise batching (Yu et al., 2022; Kwon et al., 2023), which leverages the fact that the computation performed to compute a new token is functionally the same (and uses the same model parameters) regardless of the token position within the sequence. This allows new requests to be continuously scheduled when slots within a batch become available. For example, when one response is completed, the start of the next response to be formed can immediately take the finished response's place in the batch, without waiting for the rest of the batch responses that might be longer. In our Recursive Transformer, parameter sharing occurs not only across different timesteps, but also across different depths (loop iterations). This enables an extra dimension of dynamic grouping: jointly computing different iterations of the looped layer blocks per individual responses within the same batch.

Our key contributions are as follows:

- We introduce a framework for initializing and training Relaxed Recursive Transformers and demonstrate strong performance compared to non-recursive models of comparable size. For example, when we uptrained a recursive Gemma 1B model converted from a pretrained Gemma 2B (Team et al., 2024), we observed up to 13.5 absolute accuracy improvement (22% error reduction) on few-shot tasks compared to a non-recursive Gemma 1B model (pretrained from scratch). Furthermore, we show that by incorporating knowledge distillation (Hinton et al., 2015; Kim & Rush, 2016), our recursive Gemma model, uptrained on 60 billion tokens, achieves performance on par with the full-size Gemma model trained on a massive 3 trillion token corpus (see §3.3 for details).

- Based on our Relaxed Recursive Transformer, we also evaluate a key use case for continuous depth-wise batching with early-exiting (Bae et al., 2023; Schuster et al., 2022; Elbayad et al., 2020; Graves, 2016a), which opportunistically makes predictions for samples with high confidence at earlier stages. From our simulation, Early Exits reveal a substantial throughput improvement of up to 2-3× compared to a vanilla Transformer with the same architecture. Notably, the recursive Gemma model, which outperforms the vanilla Pythia model, can theoretically achieve a nearly 4× increase in throughput (see §3.8 for details).

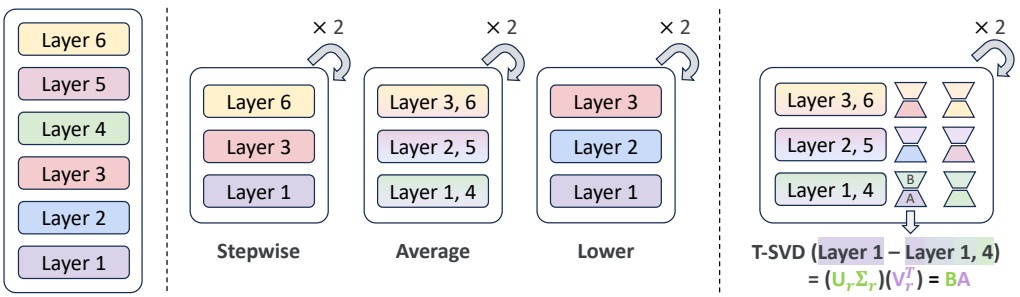

Figure 2: **Left:** An example of unshared, full-size model with 6 layers. **Middle:** Three proposed methodologies for initializing looped layers in a Recursive Transformer. Each layer number indicates the source layer in the full-size model used for initialization. **Right:** Example of a Relaxed Recursive Transformer initialized by SVD method. Here, looped layers are initialized using the Average method.

## 2 EFFECTIVE MODEL COMPRESSION WITH RECURSIVE PATTERNS

In this section, we present the main details of our method for converting a vanilla Transformer model into a parameter-shared model that outperforms models of equivalent size. We first provide a short overview of the Transformer architecture (§2.1). Then, we introduce the Recursive Transformer and present effective techniques to initialize its looped layers by leveraging the weights of the original pretrained model (§2.2). In §2.3, we relax the parameter-sharing constraint in the model design, and add a limited set of layer-specific parameters to further improve the model's accuracy while maintaining compact representations. Finally, we show how, beyond reduced memory, Recursive Transformers readily support further throughput optimizations via a novel inference paradigm (§2.4).

### 2.1 BASIC TRANSFORMER ARCHITECTURE

Large language models (Rivière et al., 2024; Reid et al., 2024; OpenAI, 2023; Dubey et al., 2024) typically leverage the Transformer architecture (Vaswani et al., 2017). A Transformer consists of $L$ layers, where the hidden states at each time step $t$ are computed by running through the series of layers:

$$\mathbf{h}_t^\ell = f(\mathbf{h}_t^{\ell-1}; \Phi_\ell), \ \ell \in [1, L], \tag{1}$$

with $\mathbf{h}_t^0$ representing the embedding of the token $y_{t-1}$ from the previous time step, and $\Phi_\ell$ denoting the trainable parameters of the $\ell$-th layer. Each layer has two core components: a multi-head attention (MHA) mechanism and a feed-forward network (FFN). MHA employs multiple attention heads to capture diverse relationships within the input sequence via linear attention weights and scaled dot-product attention mechanisms. The FFN structure typically consists of two linear transformations, but different models exhibits distinct structural variations. See Appendix C for further details.

### 2.2 RECURSIVE TRANSFORMER: LOOPED LAYER TYING

In this work, we revisit parameter sharing in the context of LLMs and propose the Recursive Transformer architecture. Among various looping strategies (refer to Appendix D), we specifically adopt the CYCLE strategy (Takase & Kiyono, 2023) for Recursive Transformers, wherein a single block of unique layers is recursively reused. This inherent design aligns seamlessly with early-exiting mechanisms, potentially offering substantial speedup. The model's hidden states are computed as:

$$\mathbf{h}_t^\ell = f(\mathbf{h}_t^{\ell-1}; \Phi'_{((\ell-1) \bmod L/B)+1}), \ \ell \in [1, L], \tag{2}$$

where the parameter-shared model is parameterized by $\Phi'$, and $B$ denotes the number of looping blocks (we restrict $B$ to be a factor of $L$). For example, Gemma 2B (Team et al., 2024) with 18 layers can be converted to a recursive variant with 2 blocks by storing weights for only the first 9 layers. The forward pass will loop twice through these 9 layers. We tie all trainable parameters, including the weights of the linear layers in the Transformer blocks and the weights of the RMSNorm (Zhang & Sennrich, 2019).

**Initialization techniques for looped layers** To mitigate the potential performance drop associated with reduced capacity in parameter-shared models, we propose several novel initialization method-ologies to facilitate effective knowledge transfer from unshared, pretrained models to Recursive

Transformers. Figure 2 illustrates three such techniques. The Stepwise method selects intermediate layers at specific intervals while keeping the first and last layer fixed. This is motivated by prior work (Liu et al., 2023; Zhang et al., 2024a; Zeng et al., 2023; Fan et al., 2020) showing minimal impact on generation quality when skipping a few layers in LLMs. The Average method initializes the shared weights among tied layers by averaging their weight matrices, whereas the Lower method directly uses weights from the first $K$ layers of the unshared model. We conducted a brief uptraining on 15 billion tokens to investigate the extent of performance recovery in these initialized models (§3.4) and found the Stepwise approach to perform best for Recursive Transformers. However, we found the Average method to perform best for Relaxed Recursive Transformers, discussed next.

## 2.3 RELAXED RECURSIVE TRANSFORMER: MULTI-LORA LAYERS

While full layer-tying is effective for compressing the model's size while maintaining strong capabilities, it has two noticeable limitations: (1) the set of possible model sizes is limited to scaling the number of layers, and (2) each model layer ends up having to serve multiple roles associated with different depths of the model. To address this, we introduce Relaxed Recursive Transformers in which we incorporate independent adapter modules (Hu et al., 2022; Houlsby et al., 2019) for each layer, relaxing the strict parameter sharing. While we experiment with various approaches like layer-specific prefixes (Liu et al., 2021) (see Appendix K), we find low-rank adaptation (LoRA) modules (Hu et al., 2022) to efficiently capture the subtle variations between tied layers. Specifically, we modify Eq. 2 to:

$$\mathbf{h}_t^\ell = f(\mathbf{h}_t^{\ell-1}; \Phi'_{((\ell-1) \bmod L/B)+1}, \Delta\Phi'_\ell), \ \ell \in [1, L], \tag{3}$$

where $\Delta\Phi'$ is the (small) set of parameters for the LoRA modules.

In this relaxed model, each looped layer is augmented with multiple LoRA modules. For example, a recursive model with two loop iterations has a single block of shared layers, and two different LoRA modules are attached to each layer within this block. The first and second LoRA modules per layer are used during the first and second loop iterations, respectively. Functionally, these LoRA modules introduce low-rank deltas to all of the shared, linear weight matrices. More concretely, for a base transformation $\mathbf{h} = \mathbf{W}'\mathbf{x}$, our modified forward pass yields $\mathbf{h} = \mathbf{W}'\mathbf{x} + \Delta\mathbf{W}'\mathbf{x} = \mathbf{W}'\mathbf{x} + \mathbf{B}\mathbf{A}\mathbf{x}$, where $\mathbf{A} \in \mathbb{R}^{(r \times k)}$ and $\mathbf{B} \in \mathbb{R}^{(d \times r)}$ denote the weight matrices of LoRA with rank $r$.

**LoRA initialization via truncated SVD** Unlike typical LoRA finetuning setups that train only the LoRA parameters, here we train all model parameters to let the shared parameters learn an optimal centroid for all of the layer depths that they support. Therefore, instead of following standard zero initialization for adaptation to the frozen base model, we propose novel initialization methods, especially designed for Relaxed Recursive Transformers. To effectively match the performance of the original full-size model after initializing the tied weights as described in §2.2, we aim for the sum of the tied weights ($\Phi'$) and LoRA weights ($\Delta\Phi'$) to approximately recover the full-size model's weights ($\Phi$). We exploit truncated Singular Value Decomposition (SVD) (Hansen, 1987) on residual matrices between original weights and tied weights:

$$\mathbf{U}_r^\ell, \mathbf{\Sigma}_r^\ell, \mathbf{V}_r^\ell = \text{Truncated SVD}(\mathbf{W}_\ell - \mathbf{W}'_{((\ell-1) \bmod L/B)+1}; \ r), \ \ell \in [1, L], \tag{4}$$

where outputs retain the first $r$ columns corresponding to the $r$ largest singular values. $\mathbf{W}$ denotes the weight matrices of the full-size model, and $\mathbf{W}'$ denotes those of the Recursive Transformer. We initialize the LoRA's weights with principal components in Eq. 4: $\mathbf{B}$ as the product of $\mathbf{U}_r$ and $\mathbf{\Sigma}_r$, and $\mathbf{A}$ as the transpose of the right singular vectors $\mathbf{V}_r$ (see Figure 2). With sufficiently large ranks, our Relaxed Recursive Transformer (Eq. 3) approximates the full-size vanilla model (Eq. 1):

$$\mathbf{W}\mathbf{x} \approx \mathbf{W}'\mathbf{x} + (\mathbf{U}_r\mathbf{\Sigma}_r)(\mathbf{V}_r^\top)\mathbf{x} = \mathbf{W}'\mathbf{x} + \mathbf{B}\mathbf{A}\mathbf{x} = \mathbf{W}'\mathbf{x} + \Delta\mathbf{W}'\mathbf{x}, \tag{5}$$

Meanwhile, setting the rank to zero reduces the model to a Recursive Transformer, as the LoRA modules contribute no additional parameters, highlighting the flexibility of this relaxation approach.

## 2.4 CONTINUOUS DEPTH-WISE BATCHING AND EARLY-EXITING

In real-world deployments, user requests arrive sequentially and asynchronously. Recent research has introduced continuous sequence-wise batching (Yu et al., 2022; Kwon et al., 2023), a serving strategy that allows new requests to immediately replace completed (terminated) sequence within a batch. This approach exploits the fact that the computation performed for a new token is functionally

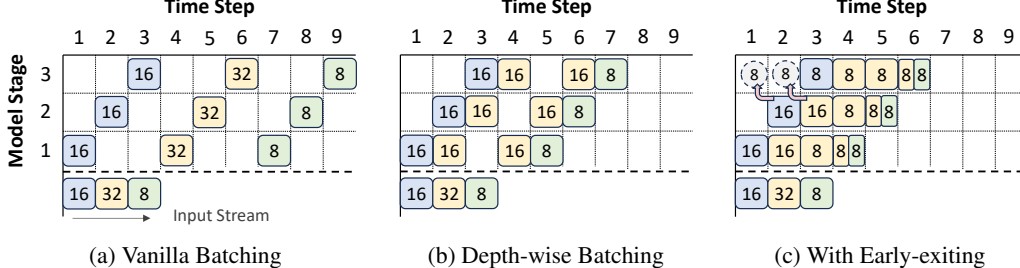

(a) Vanilla Batching  (b) Depth-wise Batching  (c) With Early-exiting

Figure 3: An illustrative example of a continuous depth-wise batching strategy together with early-exiting. We assume a maximum batch size of 32, three model "stages" (e.g., layer blocks), and a stream of batched inputs that arrive sequentially in time. In **(a)**, all three model stages must complete for the first (non-maximal) batch of 16 before the second batch of 32 examples that arrives next can be started. In **(b)**, however, half of second batch of 32 examples can share computation with the first batch of 16 that is still finishing. Finally, **(c)** demonstrates a situation where some examples within each batch can early-exit after stage 2; their vacant slots in the batch are then immediately filled.

the same and utilize the same model parameters. By continuously scheduling requests in this manner, models can operate at their maximum batch capacity, thereby enhancing serving efficiency.

The repetitive structure of Recursive Transformers allows for the same function to be applied not just across sequences, but also across depths (loop iterations). This introduces a new dimension for continuous batching, which we call Continuous Depth-wise Batching. This technique enables the simultaneous computation of different iterations of the looped layer block for different samples (See Figure 3 for an example with a single forward pass; this easily extends to multiple decode iterations per request.) With a maximum batch size of 32, a standard Transformer must wait for all model stages to complete before processing new requests. In contrast, our Recursive Transformer, because it shares layer functions across all stages, can immediately schedule new incoming requests at timestep 2, maximizing batch size utilization. This strategy can yield a substantial speedup in generation and reduce the time to first token (Fu et al., 2024; Miao et al., 2023) through faster scheduling.

Throughput improvements from depth-wise batching are further amplified when combined with early-exiting (Bae et al., 2023; Schuster et al., 2022; Elbayad et al., 2020). As depicted in Figure 3c, once some samples exit after certain looping iterations, queued requests can then be immediately scheduled. While Recursive Transformers leverage the speedup from early-exiting, they also inherently address a key challenge of batched inference in early-exiting approaches: the synchronization issue when serving large batches, as early-exited tokens might wait for others to complete processing through the entire model. We demonstrate that Recursive Transformers, equipped with this dynamic sample scheduling at various depths, can theoretically allow up to 2-3× speedup on evaluated LLMs.

## 3 EXPERIMENTS

### 3.1 EXPERIMENTAL SETUP

We evaluate our method on three popular pretrained LLMs: Gemma 2B (Team et al., 2024), TinyL-lama 1.1B (Zhang et al., 2024b), and Pythia 1B (Biderman et al., 2023). Table 2 summarizes each model's architecture and pretraining recipes, and their few-shot performance is summarized in Appendix F. After converting to Recursive Transformers, we uptrained models on the SlimPajama dataset (Soboleva et al., 2023). We used the Language Model Evaluation Harness framework (Gao et al., 2023) to evaluate accuracy on seven few-shot tasks, and averaged them for performance comparison. Detailed experimental setup for uptraining or evaluation can be found in Appendix G.

### 3.2 NON-RECURSIVE MODEL BASELINES

**Full-size model** Our ultimate goal is for the Recursive Transformer to achieve performance comparable to the original, full-size pretrained model, without much uptraining. However, we observed that the distribution divergence between the pretraining and uptraining datasets can hinder achieving the desired performance. In particular, uptraining on new datasets, particularly those of comparatively lower quality, sometimes led to performance degradation on certain benchmarks. Table 4 summarizes

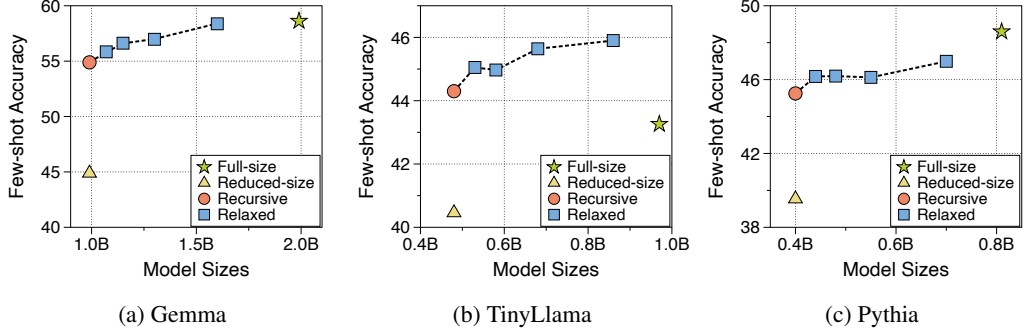

(a) Gemma  (b) TinyLlama  (c) Pythia

Figure 4: Recursive and Relaxed Recursive Transformers achieve comparable performance to full-size models, and significantly outperform reduced-size models. Recursive models were initialized using the Stepwise method, while relaxed models utilized Average and SVD methods for looped layers and LoRA modules. We show the performance of four different rank values: 64, 128, 256, and 512. Recursive and reduced-size models were either uptrained (recursive model) and pretrained from scratch (reduced-size model) on 60 billion tokens using a knowledge distillation objective.

the evaluation results of full-size models based on the number of uptraining tokens. For instance, in the case of Gemma, where the pretraining dataset is unreleased but potentially well-curated (Team et al., 2024), all few-shot performance metrics gradually decreased after uptraining on the SlimPajama dataset. This suggests that the achievable upper bound performance with the SlimPajama dataset might be considerably lower than the original model performance. Therefore, we set the target performance for Gemma and Pythia models as the performance achieved by uptraining a full-size pretrained model with an equivalent number of tokens. Since TinyLlama was already pretrained on SlimPajama—which is the same dataset we use for uptraining (eliminating any distribution shift)—for slightly longer than our runs, we use the performance of the original checkpoint as reference.

**Reduced-size model**   To demonstrate the performance advantages of Recursive Transformers compared to models with an equivalent number of parameters, we introduce another baseline: reduced-size models. These models have either half or one-third the parameters of their full-sized counterparts, matching the parameter count of our recursive models. However, these reduced models are pretrained from scratch on the same training recipe (number of training tokens and distillation from full-size model), but without the benefits of the pretrained weights and the looping mechanism. This comparison serves to highlight the efficacy of our initialization techniques and the recursive function itself in attaining strong performance, even with a constrained model size.

## 3.3   MAIN RESULTS

Figure 4 presents the few-shot performance of Recursive Transformers with two blocks and their relaxed variants. Recursive Transformers, even without relaxation, demonstrate remarkably high performance despite having only half the parameters of the full-size model. The Gemma model achieved a 10%p performance gain compared to the reduced-size model, which was also trained on 60 billion tokens using distillation loss. Remarkably, the recursive TinyLlama model even surpassed the vanilla model's performance, even though the latter was pretrained on a larger corpus of 105 billion tokens. Our initialization techniques proved highly effective in achieving this superior result, along with the benefit of the uptraining dataset (SlimPajama) being the same as its pretraining dataset.

The relaxed models effectively interpolate between the full-size model and the Recursive Transformer, depending on the LoRA rank. As the model size increases with larger LoRA modules, SVD initialization methods allow for a more precise approximation of full-rank matrices, resulting in improved performance. Notably, the relaxed Gemma model with a rank of 512 achieves performance on par with the original model pretrained on 3 trillion tokens (58.4% vs. 58.6%), despite using fewer parameters and uptraining on only 60 billion tokens. This trade-off provides flexibility in selecting the best configuration for various deployment scenarios. We believe that additional uptraining and higher-quality datasets could yield better performance with even more streamlined models.

In the subsequent sections, we provide a comprehensive overview of extensive ablation studies conducted prior to achieving this final performance. In §3.4, we delve into the analysis of various initialization methodologies for Recursive Transformers. Insights into the relaxation model are detailed in §3.5. Finally, we explore enhanced training strategies like knowledge distillation (§3.6).

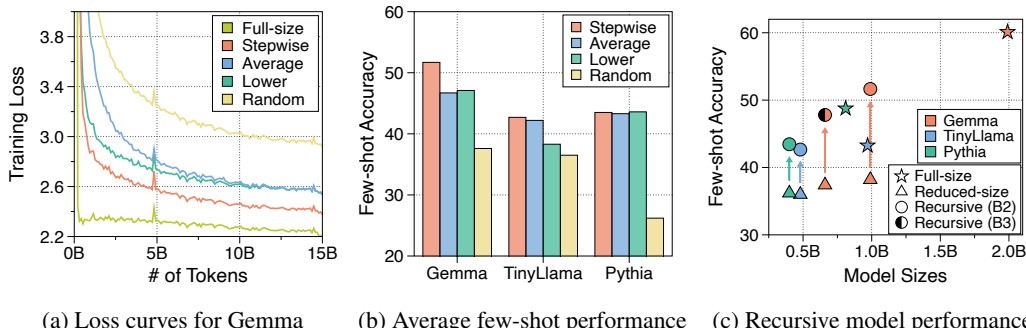

(a) Loss curves for Gemma    (b) Average few-shot performance    (c) Recursive model performance

Figure 5: **(a)** Among the proposed methods, the Stepwise method obtains the lowest training loss on the SlimPajama dataset. **(b)** The Stepwise method consistently demonstrate the highest average few-shot accuracy across three architectures. **(c)** Recursive Transformers initialized with the Stepwise method demonstrated significant performance gains compared to non-recursive model baselines.

## 3.4 Initialization Techniques for Looped Layers

**Stepwise initialization serves as the best initial point for Recursive Transformers**  We present the training loss of Gemma models initialized using three different methods in Figure 5a, and their few-shot performance in Figure 5b. Our proposed methods significantly outperformed random initialization, which simply adds recursion to a reduced-size model, suggesting that leveraging pretrained weights in any manner is beneficial for performance boost. Moreover, the Stepwise methodology consistently demonstrated best performance, aligning with insights that LLMs can preserve performance even with a few layers skipped (Raposo et al., 2024; Zhang et al., 2024a; Elhoushi et al., 2024). Interestingly, as summarized in Table 5, the recursive TinyLlama model, uptrained on only 15 billion tokens, yields few-shot performance comparable to the original model pretrained on 105 billion tokens. This suggests that with sufficient training, even a recursive architecture can match the performance of a full-size pretrained model (Dehghani et al., 2019; Takase & Kiyono, 2023).

**Recursive Gemma 1B outperforms both pretrained TinyLlama 1.1B and Pythia 1B**  The looped Gemma 1B model, utilizing our proposed Stepwise method, outperformed reduced-size baselines with equivalent parameter counts by up to 13.5 percentage points (51.7% vs. 38.2%). Furthermore, it even outperformed the full-size TinyLlama 1.1B and Pythia 1B models (see Figure 5c). This is a noteworthy achievement given that Pythia was pretrained on 300 billion tokens, whereas the recursive Gemma was uptrained on only 15 billion tokens. Consequently, high-performing LLMs serve as a promising starting point, as their recursive counterparts readily outperform other ordinary vanilla models of similar size. Further details can be found in Appendix I.

> **Takeaways for the Recursive Transformer**
>
> We find that converting well-pretrained models into Recursive Transformers leads to high-performing models with minimal uptraining. Notably, initializing looped layers via the Stepwise method yields the best results. With just 15 billion tokens of uptraining, a recursive Gemma 1B model outperforms even the full-size pretrained TinyLlama and Pythia models.

## 3.5 Relaxation of Strict Parameter Sharing via LoRA Modules

**Average initialization for looped layers optimally suits Relaxed Recursive Transformer**  Figures 6a and 6b illustrate the effect of relaxing parameter sharing via layer-wise LoRA modules. Notably, initializing tied layers in relaxed models with Average method yielded substantial performance improvements, even outperforming the non-relaxed model initialized with Stepwise. Approximating residual matrices between averaged weights and their individual weights appears readily achievable using truncated SVD with low ranks. In contrast, we observed an intriguing phenomenon where our models initialized with Stepwise occasionally showed performance degradation after relaxation. This is likely because capturing the nuances between entirely distinct layer weights is challenging with an insufficient rank, leading to a suboptimal solution. Further details are provided in Appendix J.

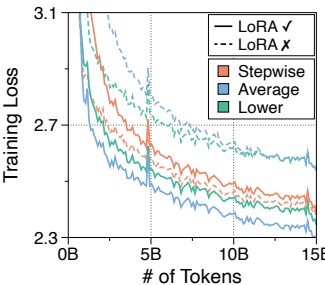 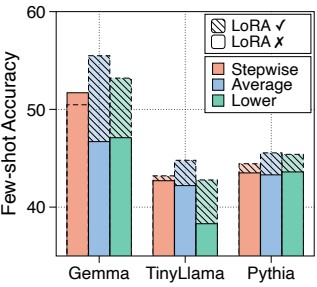 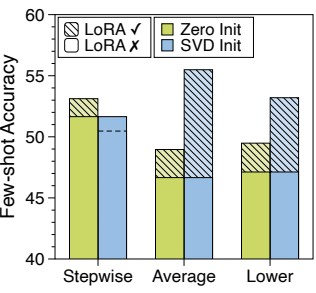

(a) Loss changes in Gemma model    (b) Accuracy gains from relaxation    (c) Effects of SVD initialization

Figure 6: The Relaxed Recursive Transformer, with its looped layer initialized using Average method, achieved the best performance in terms of both (**a**) training loss and (**b**) few-shot accuracy. The models utilize two blocks, with the LoRA modules initialized using the SVD method at a rank of 512. (**c**) SVD initialization method significantly enhanced performance compared to zero initialization.

**SVD initialization to approximate pretrained weights outperforms zero initialization**   LoRA modules initialized with zero values guarantee that the model begins training from the same point as the non-relaxed model. Conversely, SVD initialization positions the model closer to either the full-size model (with full-rank) or the non-relaxed model (with small rank). To emphasize the effectiveness of initializing near full-size model weights, we compared these two methods at a moderately large rank of 512, as shown in Figure 6c. Our proposed SVD strategy demonstrated an impressive performance boost of up to 6.5 points, facilitating faster convergence by updating the principal low-rank matrices (aligned with findings in Meng et al. (2024)). For results across other architectures, refer to Figure 15.

**Higher rank enhances recovery of original pretrained weights**   At full rank, relaxed models can perfectly match full-size pretrained models. Consequently, as illustrated in Figure 7a, performance generally improves with increasing rank, resulting in a clear Pareto frontier between model size and performance. However, only Stepwise initialization showed a U-shaped performance trend: a middle-range rank resulted in poor approximation, whereas very low ranks (akin to random initialization for LoRA modules) yielded better performance. The overall results are summarized in Table 9.

> **Takeaways for the Relaxed Recursive Transformer**
>
> Adjusting the LoRA rank in the Relaxed Recursive Transformer, together with our SVD-based initialization technique, allows for a smoother trade-off between a fully weight-tied recursive model and a vanilla model. Furthermore, we find that initializing the shared weights in the looped layers with the Average method leads to the best performance in this setting.

### 3.6 EXTENDED UPTRAINING AND KNOWLEDGE DISTILLATION

We further enhanced the performance of our low-rank models by introducing two techniques: uptraining on an extended corpus and knowledge distillation from the full-sized model. Specifically, we increased the number of uptraining tokens from 0.5% to 2% of the total 3 trillion tokens used for pretraining Gemma models, resulting in a total of 60 billion tokens. Additionally, we regularized the losses using a forward Kullback-Leibler divergence (Hinton et al., 2015; Kim & Rush, 2016), which exhibited the best performance gains among the examined distillation losses. Table 11 summarizes the results of various ablation studies conducted to investigate the impact of these two techniques.

The combined effect of these techniques is presented in Figure 7b, demonstrating an improvement of up to 4.1 percentage points in few-shot accuracy compared to the previous 15 billion token uptraining results. Notably, the relaxed Gemma model with a rank of 512 nearly matched the performance of the full-size model. We also expect that further performance gains can be achieved with a much lighter recursive model by utilizing a superior teacher model or conducting more extensive training on high-quality data. Figure 7c illustrates the Pareto frontier achieved by the final models. All models exhibit competitive performance compared to the full-size model. Moreover, the superior performance of the recursive Gemma model strongly highlights the advantages of converting high-performing LLMs to a recursive architecture. Additional details can be found in Appendix L.

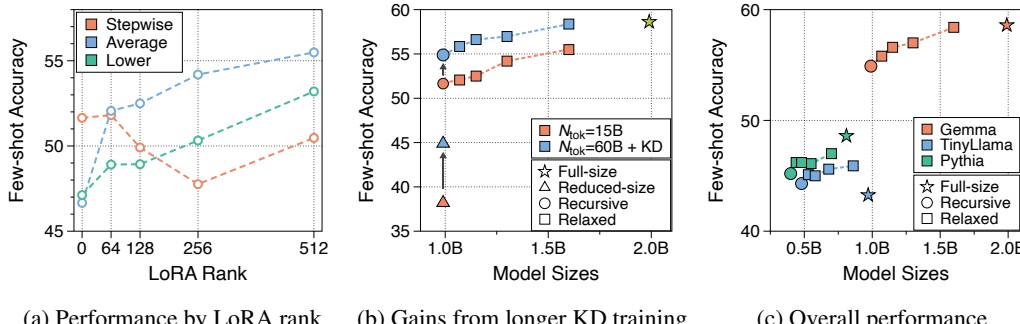

| (a) Performance by LoRA rank | (b) Gains from longer KD training | (c) Overall performance |

Figure 7: (**a**) Increasing the LoRA rank typically leads to improved performance in relaxed Gemma models, attributed to the use of SVD initialization. (**b**) Extended uptraining and knowledge distillation yielded substantial accuracy improvements for Gemma models. Note that the full-size model is a pretrained model that is further uptrained on 60 billion tokens. (**c**) Recursive and Relaxed Recursive Transformers achieve a compelling Pareto frontier with respect to model size and performance. Recursive and relaxed models used Stepwise and Average method to initialize looped layers, respectively.

Table 1: A small loss coefficient to the first loop output (intermediate output) can significantly improve intermediate performance without compromising the final performance. Performance was evaluated under a static-exiting scenario (Schuster et al., 2022), where all tokens exit at either first or second loop. We further trained the previously uptrained Gemma models on 15 billion tokens (post-training). Delta ($\Delta$) denotes the performance changes in the final outputs after early-exit training.

| | Uptrain | | Looping | | Early-Exit Train | | | Few-shot Accuracy $\uparrow$ | | | | | | | | |
|---|---|---|---|---|---|---|---|---|---|---|---|---|---|---|---|---|
| N-emb | PT | $N_{tok}$ | Block | Init | $N_{tok}$ | CE | KD | LD | HS | PQ | WG | ARC-e | ARC-c | OB | Avg | $\Delta$ |
| 0.99B | ✓ | 15B | 2 | Step | - | - | - | 53.0 | 57.3 | 73.2 | 56.2 | 56.1 | 29.2 | 36.6 | 51.7 | - |
| 0.99B | ✓ | 15B | 2 | Step | 15B | Weighted | ✗ | 48.9 | 55.5 | 72.7 | 55.3 | 54.9 | 30.1 | 36.0 | 50.5 | −1.2 |
| | | | | | | | | 49.5 | 54.8 | 72.0 | 53.4 | 54.1 | 29.1 | 35.6 | 49.8 | - |
| 0.99B | ✓ | 15B | 2 | Step | 15B | Agg (0.1) | ✗ | 53.0 | 59.1 | 73.9 | 55.4 | 57.4 | 30.6 | 37.8 | 52.5 | +0.8 |
| | | | | | | | | 45.9 | 51.2 | 71.4 | 54.5 | 48.1 | 26.8 | 32.0 | 47.1 | - |
| 0.99B | ✓ | 15B | 2 | Step | 15B | Weighted | ✓ | 47.7 | 55.1 | 73.2 | 55.6 | 54.5 | 29.1 | 37.2 | 50.4 | −1.3 |
| | | | | | | | | 48.3 | 54.9 | 72.1 | 55.9 | 54.3 | 28.4 | 35.4 | 49.9 | - |
| 0.99B | ✓ | 15B | 2 | Step | 15B | Agg (0.1) | ✓ | 52.9 | 58.9 | 73.7 | 55.7 | 57.5 | 31.1 | 38.2 | 52.6 | +0.9 |
| | | | | | | | | 46.3 | 52.1 | 71.6 | 55.3 | 49.2 | 28.5 | 32.6 | 48.0 | - |

## 3.7 EARLY-EXITING AND RECURSIVE TRANSFORMER

The throughput of Recursive Transformers can be amplified by an early-exiting framework. Hence, we further train intermediate representations from fewer looping iterations to enable token prediction. We conducted an ablation study on various strategies, as summarized in Table 1 (more detailed results are presented in Table 13). Directly applying the weighted CE loss ($\mathcal{L} = \sum_{i=1}^{B} \alpha_i \mathcal{L}_i$ where $\alpha_i = i / \sum_i i$) commonly used in prior works (Schuster et al., 2022; Bae et al., 2023) led to an overemphasis on the training of intermediate representations. To address this, we employ an aggressive coefficient strategy that aggressively reduces the loss coefficient for intermediate outputs while maintaining a coefficient of 1 for the final output. Our experiments demonstrated that an aggressive coefficient of 0.1, utilizing knowledge distillation from the detached final outputs (Bae et al., 2023), effectively preserves final performance while enhancing intermediate performance. Notably, the first loop output yielded only a difference of 4.6 percentage points in accuracy compared to the final output. This underscores the potential to maximize the benefits of early-exiting in parameter-shared LLMs.

We applied this post-training strategy for early-exiting to our final uptrained models (shown in §3.3), with all experimental results detailed in Appendix M. The aggressive coefficient strategy, combined with self-distillation, consistently achieved the best performance for intermediate outputs while maintaining strong performance for the final loop output across all models. While the optimal strategy from non-relaxed models was applied to relaxed models, a tailored training approach could further improve intermediate loop output performance in Relaxed Recursive Transformers.

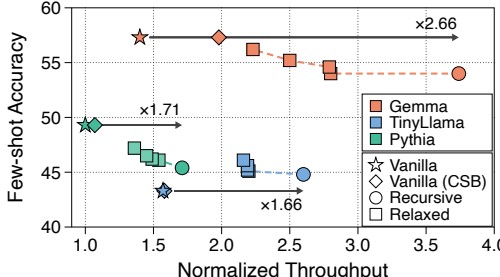

| N-emb | Loop | LoRA | Batch | Exit | Acc. | Thr. | $\Delta_V$ | $\Delta_{Seq}$ |
|---|---|---|---|---|---|---|---|---|
| 1.99B | - | - | - | ✗ | 57.3 | 1080 | ×1.00 | ×0.71 |
| 1.99B | - | - | CSB | ✗ | 57.3 | 1528 | ×1.41 | ×1.00 |
| 0.99B | 2 | - | CDB | ✓ | 54.0 | 2877 | ×2.66 | ×1.88 |
| 1.07B | 2 | 64 | CDB | ✓ | 54.0 | 2157 | ×2.00 | ×1.41 |
| 1.15B | 2 | 128 | CDB | ✓ | 54.6 | 2149 | ×1.99 | ×1.41 |
| 1.30B | 2 | 256 | CDB | ✓ | 55.2 | 1921 | ×1.78 | ×1.26 |
| 1.60B | 2 | 512 | CDB | ✓ | 56.2 | 1719 | ×1.59 | ×1.13 |

Figure 8: Continuous depth-wise batching (CDB) with early exiting enables Recursive Transformers to theoretically achieve significant throughput improvements. Throughput (tokens/sec) was averaged across SlimPajama, RedPajama, and PG19, and then normalized to the throughput of the vanilla Pythia model. The accompanying table gives detailed throughout and performance measurements for Gemma. $\Delta_V$ measures throughput relative to the vanilla Gemma model, while $\Delta_{Seq}$ measures throughput relative to the vanilla Gemma model with continuous sequence-wise batching (CSB).

## 3.8 HYPOTHETICAL GENERATION SPEEDUP VIA CONTINUOUS DEPTH-WISE BATCHING

**How we theoretically approximate actual throughput**  As developing practical early-exiting algorithms is beyond the scope of this work, we present hypothetical throughput improvements based on an oracle-exiting approach (Schuster et al., 2022; Bae et al., 2023). This assumes that tokens exit at the earliest looping block where their prediction aligns with the final loop's prediction. We simulated the generation of language modeling datasets as if they were generated by our models, to obtain the exit trajectory for each token. Then, we measured the average per-token generation time under specific constraints, such as different memory limit or context lengths. Using these measurements and the exit trajectory data, we conducted simulations to estimate theoretical throughput. Detailed explanations and limitations are discussed in Appendix N.

**Continuous depth-wise batching with early-exiting can substantially boost throughput**  Figure 8 illustrates the throughput of our proposed models and the vanilla Transformer across three architectures. We consistently achieve higher speeds than the vanilla models by combining continuous depth-wise batching with early-exiting, even surpassing those with continuous sequence-wise batching (Yu et al., 2022; Kwon et al., 2023). In particular, Recursive models demonstrate up to 2.66× speedup in generation compared to vanilla counterparts. Additionally, the recursive Gemma model significantly outperforms the vanilla pretrained Pythia model, with nearly 4× improvement in throughput. Relaxed recursive models show a clear trade-off between achievable performance and throughput, modulated by the degree of relaxation through LoRA ranks. This characteristic enables flexible model selection tailored to specific deployment scenarios. Comprehensive results are presented in Tables 17 and 19.

> **Takeaways for Continuous Depth-wise Batching**
>
> We analyze the potential for throughput improvement in the Recursive Transformer via continuous depth-wise batching, a novel inference paradigm. In theory, we find that we can achieve up to 2-3× speedup compared to a vanilla Transformer. This even outperforms the throughput gain achieved by existing continuous sequence-wise batching methods in vanilla models.

## 4 CONCLUSION

In this work, we introduced Recursive Transformers, in which we compress LLMs via parameter sharing across recursively looped blocks of layers. Additionally, we presented a novel relaxation strategy that allows for low-rank deltas between shared layers by integrating layer-specific LoRA modules into the fully-tied structure. Through novel initialization techniques for looped layers and LoRA modules, we achieved significant performance improvements that closely approximate the original pretrained model. Finally, by exploiting the recursive patterns and an early-exiting approach, we propose a continuous depth-wise batching paradigm tailored for efficient serving systems of Recursive Transformers. We theoretically demonstrated that an oracle-exiting strategy can yield substantial throughput gains, reaching up to 2-3× speedup. The limitations and future works are discussed in Appendix A.

## ACKNOWLEDGEMENTS

We thank Jacob Eisenstein for valuable feedback on an earlier version of the paper. We thank Jiyoun Ha, Alfred Piccioni, Dayeong Lee for the support with setting up the experimental environment. We also thank Donald Metzler, Ivan Korotkov, Jai Gupta, Sanket V. Mehta, Vinh Q. Tran, Brennan Saeta, Jean-François Kagy, Zhen Qin, and Jing Lu for helpful conversations. Finally, we thank the Google Cloud Platform for awarding Google Cloud credits for this project.

## REPRODUCIBILITY STATEMENT

To ensure the reproducibility of our work, we provide a comprehensive description of our model architectures in Appendix F, and details of experimental settings can be found in Appendix G. We utilized the open-source HuggingFace framework and followed established open-source frameworks for evaluation, further enhancing reproducibility. We plan to release the source codes upon publication to facilitate future research.

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

# A    LIMITATION AND FUTURE WORK

**Compatibility with sparse designs**   Sparsity-based approaches, such as pruning (Han et al., 2015), quantization (Jacob et al., 2018), or layer-skipping mechanisms (Raposo et al., 2024), recently also give promising model compression results. In fact, many of these techniques are complementary to our approach: for example, we can seamlessly have a recursive, *sparse* architecture. In this work, we rather choose to focus on recursive dense designs (a domain that remains relatively unexplored) that also have very promising, practical performance traits (i.e., allowing for continuous depth-wise batching for faster throughput). That said, while in this work we take the first step at studying Relaxed Recursive Transformer with dense Transformer layers, we do believe that incorporating Mixture-of-expert (Fedus et al., 2022), activation-skipping (Liu et al., 2023) and SSM components (Glorioso et al., 2024) within the looped blocks are promising directions for future research.

**Latent Reasoning via Recurrent Depth**   Beyond efficiency gains through down-scaling materialized parameters with recursive patterns, an alternative research direction lies in scaling-up recurrent depth to facilitate latent reasoning. Specifically, recurrent computation can manifest thinking vertically by processing internal hidden states at each depth. One promising approach involves leveraging contemplation tokens (Pfau et al., 2024; Goyal et al., 2024) or latent (continuous) space representations (Hao et al., 2024; Cheng & Van Durme, 2024) to enhance reasoning in mathematical and code generation tasks. Another valuable direction focuses on enhancing the efficiency and training stability of approaches that recursively scale-up depth, building upon concepts of deep thinking (Schwarzschild et al., 2021; Geiping et al., 2025).

**Scaling up Recursive Transformers**   Scaling our approach to larger LLMs (7B and beyond) is a promising avenue for future research. While our methodology is expected to remain effective, achieving comparable performance may require significantly higher uptraining costs. Increased model size offers the potential for a reduced memory footprint from recursive patterns; however, it is unclear whether this translates to larger batch sizes, given the corresponding increase in hidden dimensions. Nevertheless, our continuous depth-wise batching will yield considerable gains in serving efficiency.

**Beyond hypothetical generation speedup**   Our oracle-exiting approach assumes any intermediate prediction matching the final output can be exited. However, accurate throughput measurement requires confidence-based early-exiting algorithms (Schuster et al., 2022; Bae et al., 2023). Moreover, practical deployment needs to address decoding bottlenecks like key-value cache computation for exited tokens in remaining loops. Nevertheless, there are potential solutions: for example, the missing KV cache computations can be addressed by leveraging continuous depth-wise batching, allowing the KV cache for exited positions in subsequent loops to be performed in parallel with the computations for the next sequence sample. Moreover, we can explore key-value cache sharing strategies (Sun et al., 2024; Brandon et al., 2024) for future work.

**Efficient serving of multi-LoRA layers**   Relaxed models require the computation of distinct LoRA modules during batched inference, akin to multi-task learning (Feng et al., 2024; Wang et al., 2023), hindering parallel computation. We concatenated LoRA weights into a single weight to improve efficiency over sequential computation, yet it introduces redundancy. To mitigate this, we can explore optimized CUDA kernels for LoRA serving (Sheng et al., 2023; Chen et al., 2024a) and parallelization across accelerators, inspired by distributed training for Mixture of Experts (Fedus et al., 2022; Gale et al., 2023).

# B    RELATED WORK

Cross-layer parameter sharing has proven to be an effective method for achieving parameter efficiency in deep learning models such as RNNs (Sherstinsky, 2018; Graves, 2016b), CNNs (Eigen et al., 2014; Savarese & Maire, 2019; Guo et al., 2019; Shen et al., 2022), and the popular Transformer architecture. The Universal Transformer (Dehghani et al., 2019), a recurrent self-attentive model, demonstrated superior performance to non-recursive counterparts with significantly fewer parameters. This cross-layer parameter sharing approach has subsequently been explored in various tasks, including language understanding  (Lan et al., 2020), language modeling (Bai et al., 2019; Mohtashami et al., 2023; Liu et al., 2024b; Csordás et al., 2024; Glorioso et al., 2024), and machine translation (Dabre & Fujita, 2019; Milbauer et al., 2023; Xia et al., 2019; Takase & Kiyono, 2023; Ge et al., 2022). These methods often claim to achieve comparable performance with more compact models and increased

computational speed, while also setting the ground for effective adaptive compute solutions (Dehghani et al., 2019; Graves, 2016b; Schuster et al., 2021).

Concurrently, there has been growing interest in exploiting recurrent architectures for algorithmic or logical reasoning tasks (Saunshi et al., 2024). Prior research (Schwarzschild et al., 2021; McLeish & Tran-Thanh, 2022) has shown that recurrent networks can extrapolate reasoning strategies learned on simple problems to harder, larger problems through additional recurrences during inference. The looped Transformer structure has also been employed to emulate basic computing blocks for program simulation (Giannou et al., 2023), to learn iterative algorithms for data-fitting problems (Yang et al., 2024), to achieve length generalization in algorithmic tasks (Fan et al., 2024), and promising theoretical potential for few-shot learning (Gatmiry et al., 2024).

However, previous work has predominantly focused on relatively small Transformer models, trained from scratch without leveraging pretrained model weights. Our work distinguishes itself by investigating parameter sharing in the context of LLMs and proposing effective initialization strategies that leverage the knowledge embedded within existing LLMs. To the best of our knowledge, we are the first to propose a generalized framework for parameter-shared models, enabling relaxation in weight tying constraints through layer-specific modules.

In this paper, we also discuss how Recursive Transformers can be well suited for early-exiting techniques to accelerate decoding in LLMs. The inherent recursive structure readily enables early-exiting for individual responses within a large serving batch, which is often a practical limitation of such techniques. Vanilla Transformers encounter a synchronization issue with early-exiting, where the model must forward all layers if even a single token in a batch requires full processing (exited tokens must wait for them). Several approaches attempt to exploit this idle time by computing missing KV caches for exited tokens in later layers, which are essential for subsequent sequence generation. These techniques include state propagation (Schuster et al., 2022; Elbayad et al., 2020), SkipDecode (Del Corro et al., 2023), and parallel decoding (which can be combined with Speculative Decoding) (Bae et al., 2023; Elhoushi et al., 2024; Liu et al., 2024a; Chen et al., 2024b; Tang et al., 2024). Nevertheless, the heterogeneous parameters across varying model depths still hinder the efficient progression of exited tokens to subsequent sequences. In contrast, our Recursive Transformers enable parallel computation for tokens at different depths and sequences (in a continuous depth-wise batching paradigm)—also allow for parallel computation of missing KV caches with minimal overhead during the memory-bounded decoding phase.

## C  COMPONENTS IN TRANSFORMER ARCHITECTURE

The Transformer block consists of two core components: a multi-head attention (MHA) mechanism and a feed-forward network (FFN). MHA utilizes multiple attention heads to capture diverse relationships within the input sequence. The computation within each attention head is formulated as:

$$\text{Attention}(\mathbf{Q}, \mathbf{K}, \mathbf{V}) = \text{softmax}\left(\frac{\mathbf{Q}\mathbf{K}^T}{\sqrt{d_k}}\right)\mathbf{V},$$

where $\mathbf{Q}$, $\mathbf{K}$, and $\mathbf{V}$ are linear projections of the input, parameterized by learned weight matrices $\mathbf{W}_\ell^Q$, $\mathbf{W}_\ell^K$, and $\mathbf{W}_\ell^V$, respectively. The outputs from each head of the multi-head attention are concatenated and then projected back to the original hidden size using a learned weight matrix $\mathbf{W}_\ell^{out}$.

While the FFN structure typically consists of two linear transformations, in the Gemma model, it deviates from this standard architecture as follows:

$$\text{FFN}(\mathbf{x}) = \mathbf{W}_\ell^{down}(\text{GELU}(\mathbf{x}\mathbf{W}_\ell^{gate}) * \mathbf{x}\mathbf{W}_\ell^{up})$$

with three learned linear weight matrices and a GeGLU activation (Shazeer, 2020).

## D  PARAMETER SHARING STRATEGY

Takase & Kiyono (2023) discuss three strategies for partial layer tying in Transformer models, as depicted in Figure 9. The SEQUENE strategy is the simplest, assigning the same parameters to consecutive layers. The CYCLE strategy repeatedly stacks a single block of unique layers to achieve the desired depth. Meanwhile, the CYCLE (REV) strategy stacks the lower layers in reverse order for the remaining layers.

In the comparative analysis of SEQUENCE and CYCLE strategies (Liu et al., 2024b), CYCLE demonstrated marginally superior zero-shot performance. Although the SEQUENCE approach, which caches shared weights (the capacity of SRAM is typically sufficient to hold a single transformer block) and computes them iteratively, has the potential to mitigate the weight transfer bottleneck between SRAM and DRAM, we prioritized compatibility with early-exiting. Consequently, we specifically employed the CYCLE strategy, which enables continuous depth-wise batching and thereby maximizes the throughput of Recursive Transformers.

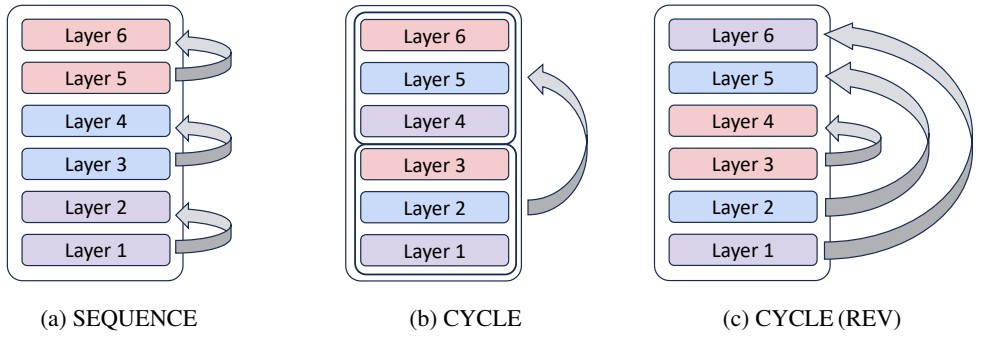

(a) SEQUENCE      (b) CYCLE      (c) CYCLE (REV)

Figure 9: Three strategies for parameter sharing (Takase & Kiyono, 2023). The examples utilize models with six layers, where identical colors represent shared weights.

# E  ILLUSTRATIVE EXAMPLES OF SVD INITIALIZATION IN RELAXED RECURSIVE TRANSFORMER

We propose an SVD initialization approach for LoRA modules within a Relaxed Recursive Transformer, effectively steering the summation of base and LoRA weights towards the pretrained weights of their corresponding depth. Figure 10 illustrates an overview of how the LoRA module is initialized under three different initialization techniques (Stepwise, Average, and Lower) for looped layers. One crucial point is that if the initialized looped layer's weights match those of the original pretrained model, its corresponding LoRA module undergoes standard zero initialization: random Gaussian for matrix $A$ and zero for $B$. For example, with the Stepwise method, the first loop's LoRA module receives standard zero initialization, while the second loop's LoRA is initialized using our proposed initialization.

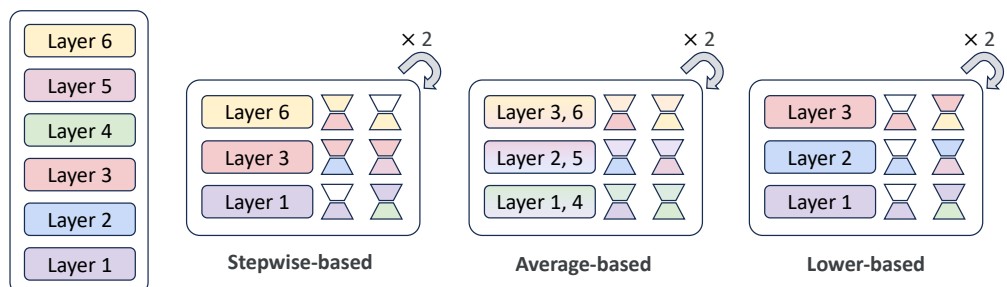

Figure 10: Overview of the proposed SVD initialization method for the Relaxed Recursive Transformer. We visualize how LoRA modules are initialized under three different looping initialization methods, assuming a full-size model with six layers and two looping blocks. $A$ matrices are colored according to the corresponding full-size model weights, while $B$ matrices are colored based on the looped layer weights. White $B$ matrices indicate cases where the full-size model and recursive model weights are identical, resulting in standard zero initialization.

# F OVERVIEW OF THREE PRETRAINED LLMS

We utilized three pretrained models—Gemma 2B (Team et al., 2024), TinyLlama 1.1B (Zhang et al., 2024b), and Pythia 1B (Biderman et al., 2023)—and converted them into Recursive Transformers. Detailed model configurations are summarized in Table 2, and their corresponding few-shot performance results are presented in Table 3.

Table 2: Key parameters and pretraining details of three models. The sizes of each model refer to the number of embedding parameters (embedding matrices and classifier heads), and all other non-embedding parameters. Gemma and TinyLlama utilize Multi-Query (Shazeer, 2019) and Grouped-Query (Ainslie et al., 2023) attention mechanisms, which leads to a reduced number of key-value heads. * Especially, we take an early TinyLlama checkpoint to study an under-trained model.

| Models | Model Architecture | | | | | | | | Pretraining | | |
|---|---|---|---|---|---|---|---|---|---|---|---|
| | N-emb | Emb | $N_L$ | $d_{model}$ | $N_{head}$ | $N_{KV}$ | $d_{head}$ | Vocab | Dataset | $N_{tok}$ | $L_{ctx}$ |
| Gemma 2B | 1.98B | 0.52B | 18 | 2048 | 8 | 1 | 256 | 256K | Unreleased | 3T | 8K |
| TinyLlama 1.1B | 0.97B | 0.13B | 22 | 2048 | 32 | 4 | 64 | 32K | SlimPajama + Starcoderdata | 73B* 32B | 2K |
| Pythia 1B | 0.81B | 0.21B | 16 | 2048 | 8 | 8 | 256 | 50K | Pile | 300B | 2K |

Table 3: Few-shot performance of pretrained models. Few-shot accuracy is measured on the LAM-BADA, HellaSwag, PIQA, WinoGrande, ARC-easy, ARC-challenge, and OpenBookQA benchmarks. We evaluated intermediate checkpoints up to the fully trained checkpoint for TinyLlama 1.1B. Among these, we utilized the 105B intermediate checkpoint to study an under-trained model.

| Models | N-emb | Dataset | $N_{token}$ | Few-shot Accuracy ↑ | | | | | | | |
|---|---|---|---|---|---|---|---|---|---|---|---|
| | | | | LD | HS | PQ | WG | ARC-e | ARC-c | OB | Avg |
| Gemma 2B | 1.99B | Unreleased | 3T | 63.13 | 71.38 | 78.13 | 65.04 | 72.26 | 41.89 | 40.20 | 61.72 |
| TinyLlama 1.1B | 0.97B | SlimPajama + Starcoderdata | 105B | 43.26 | 42.23 | 66.81 | 53.35 | 44.74 | 23.21 | 29.20 | 43.26 |
| | | | 503B | 48.92 | 49.56 | 69.42 | 55.80 | 48.32 | 26.54 | 31.40 | 47.14 |
| | | | 1T | 53.00 | 52.52 | 69.91 | 55.96 | 52.36 | 27.82 | 33.40 | 49.28 |
| | | | 2T | 53.33 | 54.63 | 70.67 | 56.83 | 54.67 | 28.07 | 33.40 | 50.23 |
| | | | 3T | 58.82 | 59.20 | 73.29 | 59.12 | 55.35 | 30.12 | 36.00 | 53.13 |
| Pythia 1B | 0.81B | Pile | 300B | 57.52 | 49.10 | 70.40 | 52.80 | 51.89 | 26.71 | 33.40 | 48.83 |

This diversity offers several benefits. First, with three versions of recursive models, we can compare their performance based on the number of trainable parameters. Notably, the comparison between the recursive Gemma and the pretrained TinyLlama and Pythia models highlights that leveraging well-trained model weights can lead to a superior Recursive Transformer of equivalent size, even with substantially lower uptraining costs. Second, by utilizing models ranging from under-trained (e.g., TinyLlama) to significantly over-trained (e.g., Gemma), we can gain insights into the uptraining costs required for Recursive Transformers to closely match the performance of pretrained models. Finally, the diversity in pretraining datasets allows us to observe how Recursive Transformers perform when faced with distribution shifts in the uptraining dataset. Table 4 in Section 3.2 presents the evaluation results obtained after uptraining each of the pretrained models. While TinyLlama readily improves its performance due to uptraining on the same dataset, Gemma and Pythia show a decline in few-shot performance with SlimPajama uptraining, which can be attributed to the differences in data distribution and the lower quality of the uptraining dataset.

## G  EXPERIMENTAL SETUP

**Uptraining setting**   To convert vanilla Transformers into Recursive Transformers, we conducted further uptraining on either 15 billion or 60 billion tokens from the SlimPajama dataset (Soboleva et al., 2023). SlimPajama is an open-source dataset designed for training large language models, which is created by cleaning and deduplicating the RedPajama dataset (Computer, 2023). The source data primarily consists of web-crawled data, along with data from Github, books, Arxiv, Wikipedia, and StackExchange. We employed the HuggingFace training framework (Wolf et al., 2020) and enhanced memory efficiency through the Zero Redundancy Optimizer (ZeRO) (Rajbhandari et al., 2020) from the DeepSpeed library (Rasley et al., 2020), along with mixed precision training. The context length was set to 2048, and the batch size was approximately 2 million tokens. We used the AdamW optimizer (Loshchilov & Hutter, 2019) with a learning rate of 2e-4, utilizing a cosine annealing learning rate scheduler (Loshchilov & Hutter, 2017). Additionally, we set warmup steps to 200 for 15 billion token training and 800 for 60 billion token training. Eight H100 GPUs were used for the training.

**Early-exit training setting**   Similar to the uptraining process, we used the SlimPajama dataset to enable models to predict next tokens at intermediate loops. Models with two looping blocks underwent additional training on a total of two exit points, whereas models with three blocks were trained on three exit points. We explored various strategies, but by default, we continued training on an additional 15 billion tokens, starting from the uptrained Recursive Transformers. We also utilized eight H100 GPUs and maintained consistent configurations with the uptraining settings, including batch size, context length, and learning rates.

**Evaluation setting**   We evaluated perplexity on test sets from three language modeling datasets: SlimPajama, RedPajama, and PG19 (Rae et al., 2019). Additionally, we used the Language Model Evaluation Harness framework (Gao et al., 2023) to evaluate accuracy on seven few-shot tasks: LAMBADA (Paperno et al., 2016), HellaSwag (Zellers et al., 2019), PIQA (Bisk et al., 2020), Wino-Grande (Sakaguchi et al., 2020), ARC-easy and ARC-challenge (Clark et al., 2018), and Open-BookQA (Mihaylov et al., 2018). We adhered to the standard number of shots specified by the evaluation framework for each dataset. For few-shot datasets, excluding LAMBADA and Wino-Grande, we normalized accuracy by the byte length of the target string. All evaluation performance measurements were conducted using a single H100 GPU.

**Throughput measurement settings**   To present the hypothetical generation speeds of our Recursive Transformers, we prepared two key elements: per-token generation time and exit trajectory datasets. Firstly, we measured the generation time under various model configurations using dummy weights and inputs. We measured the time for each component, such as embedding matrices, Transformer blocks, and the classifier head. Note that, for simplicity, throughput comparisons were based solely on the time spent within the Transformer block components. We tested two settings of prefix and decoding lengths (512 / 2048 and 64 / 256), calculating the per-token time by dividing the total elapsed time by the decoding length. Using a single A100 40GiB GPU, we measured these decoding times across different batch sizes, until an out-of-memory error occurred or under a specific memory constraint was reached. To obtain exit trajectory data, we assumed an oracle-exiting approach, where all tokens could exit at intermediate loops if intermediate predictions matched the final loop's prediction. Since our models are not finetuned on any specific downstream tasks, we simulated the generation of language modeling datasets as if they were generated by our models. For simplicity, we assumed a queue of 20K samples with varying context lengths, rather than considering their arrival in static or dynamic time intervals. With these two datasets, we present the hypothetical throughput of Recursive Transformers under various simulation scenarios.

## H  PERFORMANCE OF FULL-SIZE MODEL BASELINES

Our ultimate goal is for the Recursive Transformer to achieve performance comparable to the original, full-size pretrained model, but using the least amount of uptraining tokens possible. This is challenging because our recursive models have substantially fewer parameters, and model capacity is primarily determined by model size. However, prior works have suggested that FLOPs also play a role in influencing model performance (Dehghani et al., 2019; 2022; Goyal et al., 2024). Consequently, by recursively applying the function, we anticipate that with effective initialization techniques or training strategies, it might be possible to attain performance that closely approaches that of the full-size model.

However, the uptraining dataset itself can hinder this goal. Specifically, poor quality of the uptraining dataset or a significant distribution shift from the pretraining dataset can negatively impact performance. Indeed, as shown in Table 4, the Gemma model exhibited a performance decrease across all few-shot benchmarks after uptraining on SlimPajama. Conversely, TinyLlama, where the uptraining and pretraining datasets are both SlimPajama, consistently showed performance improvements.

Considering these results and our original goal, we adopted the following full-size model baselines: the original pretrained model for TinyLlama, and vanilla models uptrained with the same cost as their recursive counterparts for Gemma and Pythia.

Table 4: Uptraining pretrained models on datasets that differ significantly in quality or distribution from their pretraining data can lead to decreased performance. We evaluated models after uptraining on the SlimPajama dataset. We measured perplexity on test sets of SlimPajama, RedPajama, and PG19, and few-shot accuracy on LAMBADA, HellaSwag, PIQA, WinoGrande, ARC-easy, ARC-challenge, and OpenBookQA benchmarks.

| Models | N-emb | Uptrain | | Perplexity↓ | | | Few-shot Accuracy↑ | | | | | | | |
|---|---|---|---|---|---|---|---|---|---|---|---|---|---|---|
| | | PT | $N_{tok}$ | SlimP | RedP | PG19 | LD | HS | PQ | WG | ARC-e | ARC-c | OB | Avg |
| Gemma | 1.99B | ✓ | - | 11.46 | **8.18** | 13.52 | 63.1 | **71.4** | **78.1** | **65.0** | **72.3** | **41.9** | 40.2 | **61.7** |
| | 1.99B | ✓ | 15B | 10.76 | 8.47 | 13.08 | **63.5** | 68.5 | 77.0 | 63.5 | 67.6 | 38.1 | 42.6 | 60.1 |
| | 1.99B | ✓ | 60B | **10.58** | 8.44 | **12.71** | 60.3 | 67.9 | 76.9 | 63.5 | 64.9 | 37.2 | 39.6 | 58.6 |
| TinyLlama | 0.97B | ✓ | - | 12.26 | 9.37 | 11.94 | 43.3 | 42.2 | 66.8 | 53.4 | 44.7 | 23.2 | 29.2 | 43.3 |
| | 0.97B | ✓ | 15B | 9.87 | 8.24 | 10.73 | 49.2 | 46.3 | **68.8** | 54.0 | 48.2 | 26.0 | 32.2 | 46.4 |
| | 0.97B | ✓ | 60B | **9.59** | **8.12** | **10.42** | **51.6** | **48.8** | 68.6 | **54.1** | **49.9** | **26.2** | **32.8** | **47.4** |
| Pythia | 0.81B | ✓ | - | 15.68 | 9.90 | **12.05** | **57.5** | 49.1 | 70.4 | 52.8 | **51.9** | 26.7 | **33.4** | **48.8** |
| | 0.81B | ✓ | 15B | 13.46 | 9.95 | 13.38 | 55.0 | 49.0 | 71.0 | 53.6 | 51.8 | **28.2** | 32.8 | **48.8** |
| | 0.81B | ✓ | 60B | **12.83** | **9.76** | 13.57 | 53.0 | **50.2** | **71.1** | **54.8** | **51.9** | 27.7 | 31.6 | 48.6 |

## I  EXPANDED RESULTS OF INITIALIZATION METHODS FOR LOOPED LAYERS

**Ablation study of Stepwise method**  We initially hypothesized that the Stepwise method's performance could be significantly influenced by the specific rule used for layer selection from the pretrained model. To investigate this, we conducted a controlled experiment (illustrated in Figure 11a), where layers were selected at certain intervals starting from the first layer. We then varied whether the final layer of the pretrained model was included in the initialization or not. While a Pythia model showed no discernible differences in training loss or few-shot performance, other models like Gemma exhibited markedly superior results when both the first and last layers were preserved. This observation aligns well with prior work suggesting that maintaining the weights of the first and last layers during depth up-scaling for LLMs can yield performance benefits (Kim et al., 2024).

**Ablation study of Average method**  The Average initialization method exhibited notably poor performance, particularly when applied to the Gemma model. We hypothesized that this could be attributed to instability in the model's learned distribution, potentially arising from averaging of normalization layer weights. Relatedly, several studies (Csordás et al., 2024; Shim et al., 2024; Mohtashami et al., 2023) have explored the careful design of layer normalization in parameter-shared models. To investigate this further, we experimented with three different methods for initializing normalization weights, as outlined in Figure 11b: averaging weights (Norm-avg), selecting weights from a single layer (Norm-choice), and zero initialization (Norm-zero). The performance trend observed among these methods

varied across different model architectures. However, zero initialization of normalization layers resulted in a huge performance drop in certain architectures like TinyLlama and Pythia. Conversely, we observed no big difference between averaging and single-layer selection, suggesting that any form of distillation of the normalization weights appears to be sufficient for maintaining performance.

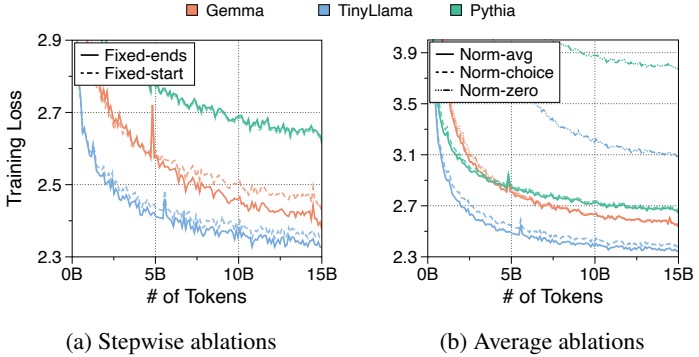

(a) Stepwise ablations  (b) Average ablations

Figure 11: Training loss curves of stepwise and average initialization variants across three models with two blocks. (a) "Fixed-start" indicates that the first layer of the pretrained model is selected initially, and subsequent layers are repeatedly chosen at a fixed interval. "Fixed-ends" means that the first and last layers are included, and intermediate layers are selected at specific step intervals. (b) When tying LayerNorm weights, we consider whether to average the weights (LN-avg), select a single weight (LN-choice), or use zero initialization (LN-zero).

**Overall comparison of training perplexity** Figure 12 presents a comparative analysis of training loss across three model architectures and varying looping blocks, incorporating our proposed initialization methodologies. To set an upper bound on performance, we utilized a full-size model further uptrained on SlimPajama, accounting for the distribution shift between uptraining and pretraining data. Additionally, we trained a Recursive Transformer from a random initialization, ensuring its exclusive reliance on the recursive architecture without leveraging any pretrained weights. While some variance was observed across architectures, all proposed methods utilizing pretrained model weights demonstrated significantly superior performance compared to random initialization. Notably, the Stepwise method consistently achieved the best performance across diverse settings. Although the full-size model's performance was considerably higher, bridging this gap with only 15 billion tokens of uptraining represents a remarkable achievement.

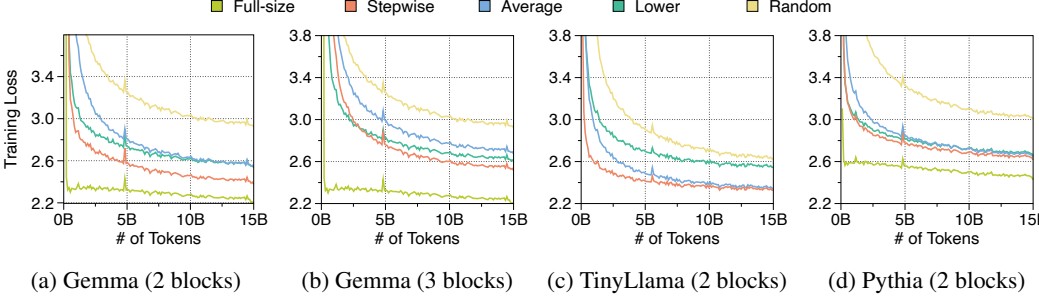

(a) Gemma (2 blocks)  (b) Gemma (3 blocks)  (c) TinyLlama (2 blocks)  (d) Pythia (2 blocks)

Figure 12: Training loss for Recursive Transformers using various initialization. We omitted a separate curve for the full-size TinyLlama model, as we used the original pretrained model as the full-size model since both pretraining and uptraining datasets are same as the SlimPajama dataset. Refer to Section 3.2 for more details.

**Overall comparison of few-shot performance** Few-shot performance exhibited a consistent trend with training perplexity. Table 5 provides a comparative summary of the proposed looping initialization methods against the full-size model, the reduced-size model, and Recursive Transformers utilizing random initialization. Moreover, Figure 13 visually illustrates the performance differences across different datasets. Notably, the Stepwise method consistently demonstrated the best performance, showing a performance improvement of up to 14.1%p compared to random initialization.

Table 5: Evaluation results of various initialization methods for looped layers. We indicate whether pretrained weights are used and the number of uptraining tokens. Perplexity is evaluated on test sets of three language modeling datasets, and accuracy is evaluated on seven few-shot benchmarks. Delta values ($\Delta$) show improvements over random initialization. We highlight the configurations that demonstrate the best performance.

| Models | N-emb | PT | $N_{tok}$ | Block | Init | SlimP | RedP | PG19 | LD | HS | PQ | WG | ARC-e | ARC-c | OB | Avg | $\Delta$ |
|---|---|---|---|---|---|---|---|---|---|---|---|---|---|---|---|---|---|
| | | | | **Uptrain** | | | **Perplexity**↓ | | | | | **Few-shot Accuracy**↑ | | | | | | |
| | 1.99B | ✓ | 15B | - | - | 10.76 | 8.47 | 13.08 | 63.5 | 68.5 | 77.0 | 63.5 | 67.6 | 38.1 | 42.6 | 60.1 | - |
| | 0.99B | ✗ | 15B | - | - | 22.63 | 20.03 | 32.60 | 28.9 | 31.6 | 63.1 | 52.3 | 41.2 | 22.5 | 27.8 | 38.2 | - |
| | 0.66B | ✗ | 15B | - | - | 24.44 | 21.69 | 36.03 | 27.2 | 30.6 | 63.8 | 50.5 | 40.6 | 22.0 | 27.0 | 37.4 | - |
| | 0.99B | ✓ | 15B | 2 | Step | **12.85** | **10.29** | **16.21** | 53.0 | 57.3 | 73.2 | 56.2 | 56.1 | 29.2 | 36.6 | 51.7 | +14.1 |
| Gemma | 0.99B | ✓ | 15B | 2 | Avg | 15.15 | 12.57 | 19.86 | 43.6 | 47.4 | 70.4 | 52.6 | 50.5 | 27.8 | 34.4 | 46.7 | +9.1 |
| | 0.99B | ✓ | 15B | 2 | Lower | 15.03 | 12.46 | 19.63 | 42.5 | 48.0 | 71.0 | 54.6 | 52.2 | 27.7 | 33.8 | 47.1 | +9.5 |
| | 0.99B | ✗ | 15B | 2 | Rand | 22.66 | 20.06 | 32.86 | 27.4 | 31.6 | 63.4 | 50.5 | 39.7 | 21.9 | 28.8 | 37.6 | - |
| | 0.66B | ✓ | 15B | 3 | Step | **14.75** | **12.10** | **19.32** | 45.0 | 49.9 | 69.8 | 55.8 | 52.7 | 27.9 | 33.6 | 47.8 | +9.9 |
| | 0.66B | ✓ | 15B | 3 | Avg | 17.45 | 14.65 | 23.63 | 39.4 | 39.0 | 66.6 | 48.7 | 46.5 | 24.7 | 31.8 | 42.4 | +4.5 |
| | 0.66B | ✓ | 15B | 3 | Lower | 15.96 | 13.24 | 20.90 | 41.9 | 43.2 | 70.0 | 52.6 | 49.5 | 26.6 | 31.6 | 45.0 | +7.1 |
| | 0.66B | ✗ | 15B | 3 | Rand | 22.67 | 20.09 | 32.77 | 28.1 | 31.4 | 63.8 | 51.1 | 41.0 | 23.0 | 26.6 | 37.9 | - |
| | 0.97B | ✓ | - | - | - | 12.26 | 9.37 | 11.94 | 43.3 | 42.2 | 66.8 | 53.4 | 44.7 | 23.2 | 29.2 | 43.3 | - |
| | 0.48B | ✗ | 15B | - | - | 16.61 | 15.66 | 20.27 | 22.3 | 30.0 | 60.9 | 50.6 | 37.0 | 23.0 | 28.0 | 36.0 | - |
| TinyLlama | 0.48B | ✓ | 15B | 2 | Step | **11.61** | **9.89** | **13.00** | 39.6 | 39.8 | 66.5 | 52.9 | 44.3 | 24.9 | 30.6 | 42.7 | +6.2 |
| | 0.48B | ✓ | 15B | 2 | Avg | 11.86 | 10.29 | 13.42 | 38.6 | 39.4 | 66.1 | 52.8 | 42.7 | 25.4 | 30.6 | 42.2 | +5.7 |
| | 0.48B | ✓ | 15B | 2 | Lower | 14.67 | 12.67 | 16.68 | 31.9 | 32.3 | 62.6 | 52.0 | 39.1 | 22.1 | 27.8 | 38.3 | +1.8 |
| | 0.48B | ✗ | 15B | 2 | Rand | 16.14 | 15.11 | 19.55 | 24.7 | 30.7 | 61.2 | 50.6 | 36.4 | 22.6 | 29.2 | 36.5 | - |
| | 0.81B | ✓ | 15B | - | - | 13.46 | 9.95 | 13.38 | 55.0 | 49.0 | 71.0 | 53.6 | 51.8 | 28.2 | 32.8 | 48.8 | - |
| | 0.40B | ✗ | 15B | - | - | 25.69 | 20.00 | 32.08 | 24.3 | 30.0 | 61.9 | 50.7 | 38.3 | 22.3 | 26.0 | 36.2 | - |
| Pythia | 0.40B | ✓ | 15B | 2 | Step | **16.38** | **12.37** | **17.74** | 43.4 | 40.5 | 67.4 | 50.8 | 46.3 | 25.7 | 30.0 | 43.5 | +7.3 |
| | 0.40B | ✓ | 15B | 2 | Avg | 16.76 | 12.76 | 18.63 | 43.6 | 39.1 | 68.2 | 51.9 | 45.4 | 25.1 | 29.8 | 43.3 | +7.1 |
| | 0.40B | ✓ | 15B | 2 | Lower | 17.04 | 12.62 | 18.44 | 43.9 | 39.2 | 66.3 | 53.4 | 45.4 | 25.8 | 31.2 | 43.6 | +7.4 |
| | 0.40B | ✗ | 15B | 2 | Rand | 24.45 | 18.93 | 29.63 | 25.2 | 30.2 | 62.1 | 51.1 | 39.2 | 22.4 | 23.6 | 36.2 | - |

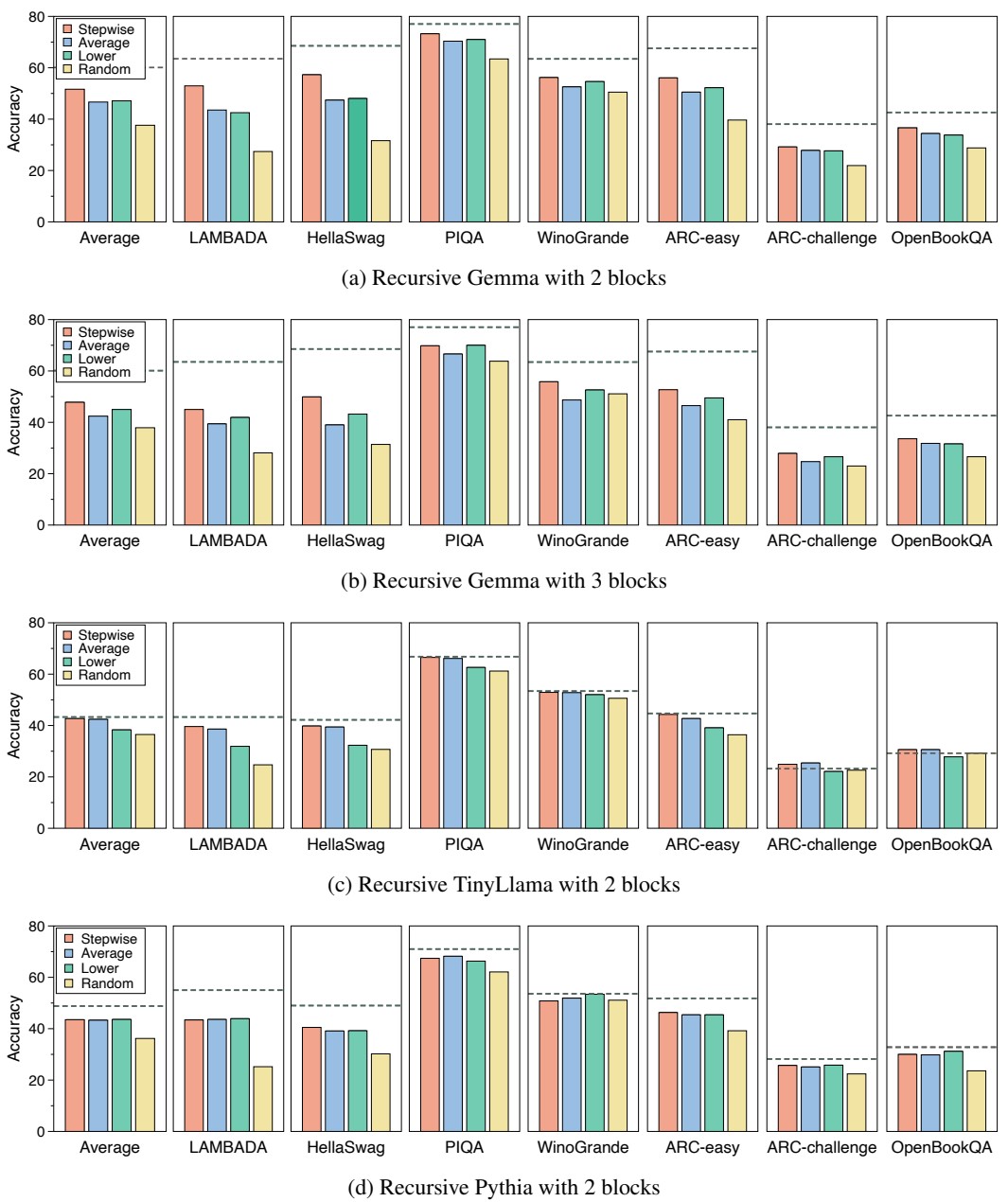

Figure 13: Few-shot performance on seven benchmarks and their average accuracy based on four looping initialization methods. Full-size model performance is represented by a gray dotted line.

**Comparison across various base model sizes** We observed a consistent superiority of Stepwise initialization strategy for recursive conversion across both 1B and 2B model scales. To further evaluate on a wide range of base model sizes, we additionally experimented with two smaller model sizes, Pythia 410M and 160M. Since we uptrained models on the Pile dataset (Gao et al., 2020), the pretraining corpus of the original Pythia model, we set the original model performance as the baseline for comparison. The results in Table 6 further validate the superior performance of the Stepwise method for looped layer initialization. These findings reinforce the robustness of our key observations regarding initialization methods for recursive conversion, complementing our original extensive experiments.

Table 6: Comparison between initialization methods for looped layers on Pythia 410M and 160M. Uptraining was performed using the Pile dataset, which was also used for pretraining the original Pythia model. In light of the inherent randomness in few-shot accuracy, a comparison based on the perplexity (PPL) would provide a more stable measure of performance.

|  | Uptrain | | Looping | | PPL↓ | Few-shot Accuracy↑ | | | | | | | |
| --- | --- | --- | --- | --- | --- | --- | --- | --- | --- | --- | --- | --- | --- |
| N-emb | PT | $N_{tok}$ | Block | Init | Pile | LD | HS | PQ | WG | ARC-e | ARC-c | OB | Avg |
| 300M | ✓ | - | - | - | - | 44.96 | 40.97 | 66.97 | 53.28 | 44.40 | 25.51 | 30.20 | 43.76 |
| 150M | ✓ | 15B | 2 | Step | **11.03** | **43.41** | **35.59** | **64.58** | **53.04** | 41.58 | 23.81 | **28.80** | **41.54** |
| 150M | ✓ | 15B | 2 | Lower | 11.47 | 42.98 | 34.32 | 63.93 | 52.41 | **42.34** | 24.15 | 25.00 | 40.73 |
| 150M | ✓ | 15B | 2 | Avg | 11.55 | 39.84 | 34.17 | 64.31 | 52.25 | 41.04 | **24.66** | 26.60 | 40.41 |
| 85M | ✓ | - | - | - | - | 13.53 | 30.67 | 58.22 | 48.62 | 36.62 | 25.00 | 28.60 | 34.47 |
| 43M | ✓ | 15B | 2 | Step | **15.93** | 21.02 | 29.28 | 60.01 | 48.93 | 37.92 | **23.98** | **28.00** | 35.59 |
| 43M | ✓ | 15B | 2 | Lower | 16.19 | 21.46 | **29.61** | 59.90 | **50.67** | **38.52** | 22.95 | **28.00** | **35.87** |
| 43M | ✓ | 15B | 2 | Avg | 16.12 | **22.36** | 29.07 | **60.17** | 49.96 | 37.24 | 23.29 | 26.60 | 35.53 |

**Individual contributions of leveraging pretrained weights and recursive patterns** To understand the performance of our Recursive Transformer, we established two non-recursive baselines: full-size model and reduced-size model. The reduced size model performance is meant to serve as a lower bound which we can use to better judge the efficacy of (1) unique looping and parameter sharing techniques that are made possible by our approach and (2) leveraging pretrained layers. To further ablate the effect of each of two components, we conducted experiments using the Pythia 410M model presented in Table 7. Intuitively, we observed significant performance gains by leveraging pretrained layers, with further improvement achieved through recursion. We believe this additional experiment provides valuable insight into the performance contributions of the two approaches proposed for constructing Recursive Transformers.

Table 7: Performance of recursive and baseline models with Pythia 410M to investigate the individual contributions of pretraining layers and looping strategy. Uptraining was performed using the Pile dataset (Gao et al., 2020), which was also used for pretraining the original Pythia model.

|  | Uptrain | | Looping | | PPL↓ | Few-shot Accuracy↑ | | | | | | | |
| --- | --- | --- | --- | --- | --- | --- | --- | --- | --- | --- | --- | --- | --- |
| N-emb | PT | $N_{tok}$ | Block | Init | Pile | LD | HS | PQ | WG | ARC-e | ARC-c | OB | Avg |
| 300M | ✓ | - | - | - | - | 44.96 | 40.97 | 66.97 | 53.28 | 44.40 | 25.51 | 30.20 | 43.76 |
| 150M | ✗ | 15B | - | - | 14.11 | 31.48 | 29.53 | 61.37 | **52.49** | 39.14 | 22.44 | 27.00 | 37.63 |
| 150M | ✗ | 15B | 2 | - | **13.81** | 31.55 | 29.94 | 62.30 | 50.88 | 40.28 | 23.98 | 28.20 | 38.02 |
| 150M | ✓ | 15B | - | Step | 11.48 | 40.48 | 34.19 | 63.42 | 50.99 | **41.84** | 23.12 | 28.40 | 40.35 |
| 150M | ✓ | 15B | 2 | Step | **11.03** | **43.41** | **35.59** | **64.58** | **53.04** | 41.58 | **23.81** | **28.80** | **41.54** |

## J    EXPANDED RESULTS OF RELAXED RECURSIVE TRANSFORMERS

**Training perplexity changes with LoRA modules**   Figure 14 illustrates the changes in training loss after incorporating the layer-wise LoRA modules. The Average and Lower initialization methods, when coupled with our proposed SVD-based initialization of the LoRA modules, demonstrated significantly enhanced benefits. In particular, the Relaxed Recursive Transformer employing the Average method consistently outperformed the others. This suggests that it is considerably easier to learn the difference between the original pretrained weights and the averaged looped weights using low-rank matrices.

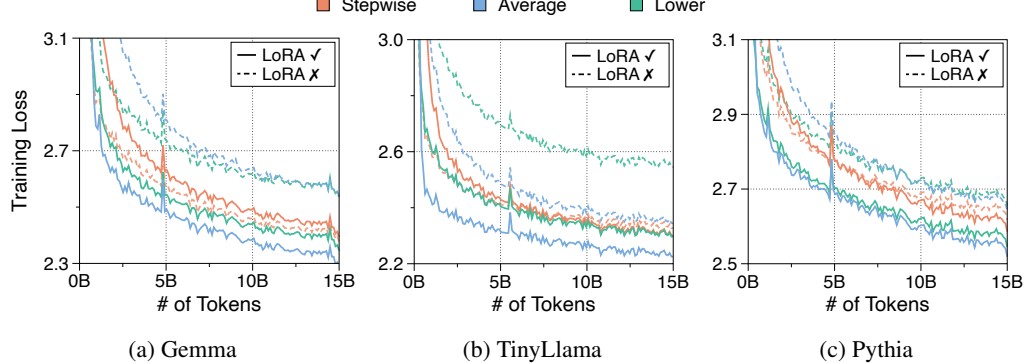

(a) Gemma                         (b) TinyLlama                         (c) Pythia

Figure 14: Comparison of training loss for recursive and relaxed recursive models. All recursive models utilize two looping blocks, and the LoRA rank is set to 512. The SVD initialization method is used for LoRA modules.

**Comparison between SVD and zero initialization**   The utilization of layer-wise LoRA modules enhances model capacity by introducing additional parameters and relaxation, thereby potentially improving performance. As depicted in Figure 15, SVD initialization significantly amplified these performance gains compared to standard zero initialization. However, an interesting exception was observed with the Stepwise method, where the SVD initialized LoRA module surprisingly led to a performance degradation in Gemma and TinyLlama. This appears to be attributed to the LoRA rank being insufficient to adequately approximate the low-rank deltas across layers, resulting in initialization at a sub-optimal point.

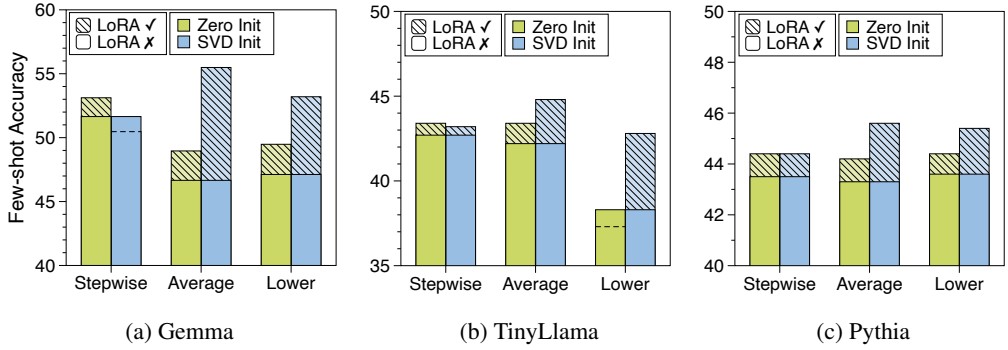

(a) Gemma                         (b) TinyLlama                         (c) Pythia

Figure 15: Comparison of average few-shot accuracy between zero and SVD initialization methods for layer-wise LoRA across three models. Performance gains due to LoRA relaxation are indicated by hatched bars, while cases where performance is lower than the recursive model without LoRAs are represented by dotted lines.

**Ablation study on the LoRA rank values**  Our proposed SVD initialization ensures that the Relaxed Recursive Transformer can function as an interpolation between vanilla and Recursive Transformers. The approximation accuracy of SVD is directly influenced by the LoRA rank value; a higher rank leads to improved restoration of the pretrained model weights. In Figure 16, we present a summary of the performance changes observed in the relaxed models by varying the LoRA ranks. As expected, for the Average and Lower looping initialization methods, a larger rank value results in enhanced performance. The Stepwise method, consistent with previous experimental findings, exhibited a U-shaped trend: with a lower rank, it behaves similarly to random initialization, resulting in a slight performance increase. However, with a higher, the approximation becomes more accurate, leading to a further increase in performance.

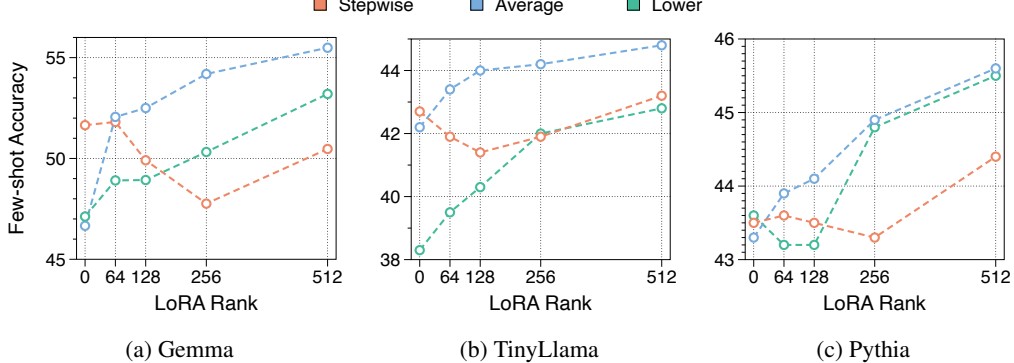

| (a) Gemma | (b) TinyLlama | (c) Pythia |

Figure 16: Performance comparison with varying LoRA ranks under different initialization methods for looped layers. All LoRA weights are initialized using our proposed SVD initialization method.

We further experimented with assigning different ranks to LoRA modules associated with each linear layer. Given the computational overhead inherent to LoRA modules, allocating varying ranks to each module can offer an optimal balance between performance and computational efficiency. The experimental results in Table 8 reveal a strong correlation between performance and overall model sizes. Due to the substantial hidden dimension of the linear weight within the FFN layer, reducing its rank led to the most significant performance drop. Conversely, the relatively smaller size of other attention weights resulted in less performance drops. An intriguing observation is the comparable performance maintained even with minimal relaxation of key-value weights (achieved through small ranks), despite the inherent strong sharing of key-value caches in the Multi-Query attention structure (Ainslie et al., 2023).

Table 8: Evaluation results of relaxed recursive Gemma models with varying LoRA ranks applied to Transformer components. We adjusted the LoRA ranks attached to query, key-value, out, and FFN linear weights. Non-embedding parameter sizes include both the base layer parameters and the attached LoRA weights.

| | Uptrain | | Looping | | LoRA | | | | | Perplexity ↓ | | | Few-shot Accuracy ↑ | | | | | | | |
|---|---|---|---|---|---|---|---|---|---|---|---|---|---|---|---|---|---|---|---|---|
| N-emb | PT | $N_{tok}$ | Block | Init | Q | KV | Out | FFN | Init | SlimP | RedP | PG19 | LD | HS | PQ | WG | ARC-e | ARC-c | OB | Avg |
| 1.99B | ✓ | 15B | - | - | - | - | - | - | - | 10.76 | 8.47 | 13.08 | 63.5 | 68.5 | 77.0 | 63.5 | 67.6 | 38.1 | 42.6 | 60.1 |
| 0.99B | ✗ | 15B | - | - | - | - | - | - | - | 22.63 | 20.03 | 32.60 | 28.9 | 31.6 | 63.1 | 52.3 | 41.2 | 22.5 | 27.8 | 38.2 |
| 1.30B | ✓ | 15B | 2 | Avg | 256 | 256 | 256 | 256 | SVD | 12.10 | 9.71 | 14.89 | 58.2 | 60.7 | 73.7 | 59.0 | 57.6 | 32.1 | 38.0 | 54.2 |
| 1.28B | ✓ | 15B | 2 | Avg | 128 | 256 | 128 | 256 | SVD | 12.27 | 9.81 | 15.10 | 57.4 | 60.2 | 72.5 | 58.9 | 58.1 | 32.6 | 37.8 | 53.9 |
| 1.29B | ✓ | 15B | 2 | Avg | 256 | 128 | 256 | 256 | SVD | 12.33 | 9.90 | 15.25 | 58.5 | 59.7 | 73.3 | 58.3 | 56.6 | 32.0 | 40.0 | 54.1 |
| 1.18B | ✓ | 15B | 2 | Avg | 256 | 256 | 256 | 128 | SVD | 12.56 | 10.12 | 15.59 | 57.0 | 58.7 | 73.0 | 57.4 | 57.0 | 31.6 | 38.2 | 53.3 |
| 1.27B | ✓ | 15B | 2 | Avg | 128 | 128 | 128 | 256 | SVD | 12.36 | 9.92 | 15.31 | 57.2 | 59.2 | 73.1 | 57.3 | 58.0 | 32.2 | 38.6 | 53.7 |
| 1.15B | ✓ | 15B | 2 | Avg | 128 | 128 | 128 | 128 | SVD | 12.52 | 10.07 | 15.51 | 56.1 | 58.2 | 72.3 | 55.8 | 57.1 | 30.7 | 37.2 | 52.5 |
| 1.14B | ✓ | 15B | 2 | Avg | 64 | 128 | 64 | 128 | SVD | 12.61 | 10.14 | 15.69 | 55.0 | 57.8 | 73.0 | 57.5 | 56.7 | 30.9 | 38.8 | 52.8 |
| 1.14B | ✓ | 15B | 2 | Avg | 128 | 64 | 128 | 128 | SVD | 12.72 | 10.18 | 15.76 | 55.5 | 57.7 | 72.7 | 57.0 | 56.9 | 30.1 | 38.2 | 52.6 |
| 1.08B | ✓ | 15B | 2 | Avg | 128 | 128 | 128 | 64 | SVD | 12.80 | 10.33 | 15.97 | 55.3 | 56.7 | 72.9 | 57.7 | 55.0 | 29.6 | 36.0 | 51.9 |
| 1.13B | ✓ | 15B | 2 | Avg | 64 | 64 | 64 | 128 | SVD | 12.77 | 10.29 | 15.95 | 55.2 | 57.4 | 73.0 | 56.7 | 56.5 | 30.5 | 37.2 | 52.3 |

**Overall performance comparison of Relaxed Recursive Transformers**  A comprehensive performance comparison is presented in Table 9. This encompasses an evaluation of the performance across three models and various looping initialization methods, considering the degree of relaxation induced by the layer-wise LoRA module. The configuration utilizing the Average method for looped layer initialization, paired with SVD initialization for the LoRA module, consistently outperformed all other baselines. Furthermore, performance clearly improved with increasing rank.

Table 9: Evaluation results of relaxed recursive models varying LoRA ranks. Delta ($\Delta$) represent the accuracy differences between relaxed and non-relaxed models using the same looping initialization.

| Models | N-emb | PT | $N_{tok}$ | Block | Init | Rank | Init | SlimP | RedP | PG19 | LD | HS | PQ | WG | ARC-e | ARC-c | OB | Avg | $\Delta$ |
|---|---|---|---|---|---|---|---|---|---|---|---|---|---|---|---|---|---|---|---|
| | | | Uptrain | Looping | | LoRA | | Perplexity↓ | | | Few-shot Accuracy↑ | | | | | | | | |
| | 1.99B | ✓ | 15B | - | - | - | - | 10.76 | 8.47 | 13.08 | 63.5 | 68.5 | 77.0 | 63.5 | 67.6 | 38.1 | 42.6 | 60.1 | - |
| | 0.99B | ✗ | 15B | - | - | - | - | 22.63 | 20.03 | 32.60 | 28.9 | 31.6 | 63.1 | 52.3 | 41.2 | 22.5 | 27.8 | 38.2 | - |
| | 0.99B | ✗ | 15B | 2 | Rand | - | - | 22.66 | 20.06 | 32.86 | 27.4 | 31.6 | 63.4 | 50.5 | 39.7 | 21.9 | 28.8 | 37.6 | - |
| | 0.99B | ✓ | 15B | 2 | Step | - | - | 12.85 | 10.29 | 16.21 | 53.0 | 57.3 | 73.2 | 56.2 | 56.1 | 29.2 | 36.6 | 51.7 | - |
| | 1.07B | ✓ | 15B | 2 | Step | 64 | SVD | 12.76 | 10.21 | 15.99 | 52.1 | 57.2 | 73.0 | 57.8 | 56.9 | 28.9 | 36.8 | 51.8 | +0.1 |
| | 1.15B | ✓ | 15B | 2 | Step | 128 | SVD | 13.44 | 10.80 | 16.98 | 50.5 | 53.0 | 71.5 | 54.4 | 55.9 | 29.3 | 34.8 | 49.9 | −1.8 |
| | 1.30B | ✓ | 15B | 2 | Step | 256 | SVD | 14.02 | 11.44 | 18.09 | 46.1 | 49.1 | 71.8 | 53.2 | 52.8 | 27.8 | 33.4 | 47.8 | −3.9 |
| | 1.60B | ✓ | 15B | 2 | Step | 512 | SVD | 13.13 | 10.66 | 16.63 | 53.0 | 54.3 | 72.1 | 54.9 | 54.8 | 28.8 | 35.4 | 50.5 | −1.2 |
| | 1.60B | ✓ | 15B | 2 | Step | 512 | Zero | 12.46 | 9.97 | 15.58 | 54.9 | 58.8 | **74.0** | 58.1 | 58.8 | 30.6 | 36.6 | 53.1 | +1.4 |
| Gemma | 0.99B | ✓ | 15B | 2 | Avg | - | - | 15.15 | 12.57 | 19.86 | 43.6 | 47.4 | 70.4 | 52.6 | 50.5 | 27.8 | 34.4 | 46.7 | - |
| | 1.07B | ✓ | 15B | 2 | Avg | 64 | SVD | 12.83 | 10.35 | 16.02 | 55.9 | 56.8 | 72.5 | 56.8 | 55.7 | 30.6 | 36.2 | 52.1 | +5.4 |
| | 1.15B | ✓ | 15B | 2 | Avg | 128 | SVD | 12.52 | 10.07 | 15.51 | 56.1 | 58.2 | 72.3 | 55.8 | 57.1 | 30.7 | 37.2 | 52.5 | +5.8 |
| | 1.30B | ✓ | 15B | 2 | Avg | 256 | SVD | 12.10 | 9.71 | 14.89 | 58.2 | 60.7 | 73.7 | 59.0 | 57.6 | 32.1 | **38.0** | 54.2 | +7.5 |
| | 1.60B | ✓ | 15B | 2 | Avg | 512 | SVD | **11.83** | **9.46** | **14.57** | 59.3 | 62.8 | 74.0 | 61.6 | 60.1 | 32.9 | 37.6 | **55.5** | +8.8 |
| | 1.60B | ✓ | 15B | 2 | Avg | 512 | Zero | 13.78 | 11.31 | 17.71 | 49.8 | 52.4 | 71.7 | 53.3 | 51.2 | 29.4 | 35.0 | 49.0 | +2.3 |
| | 0.99B | ✓ | 15B | 2 | Lower | - | - | 15.03 | 12.46 | 19.63 | 42.5 | 48.0 | 71.0 | 54.6 | 52.2 | 27.7 | 33.8 | 47.1 | - |
| | 1.07B | ✓ | 15B | 2 | Lower | 64 | SVD | 14.21 | 11.77 | 18.40 | 47.5 | 50.5 | 70.9 | 54.2 | 54.1 | 29.2 | 36.0 | 48.9 | +1.8 |
| | 1.15B | ✓ | 15B | 2 | Lower | 128 | SVD | 14.23 | 11.83 | 18.49 | 48.0 | 50.5 | 72.0 | 56.8 | 54.4 | 27.5 | 33.4 | 48.9 | +1.8 |
| | 1.30B | ✓ | 15B | 2 | Lower | 256 | SVD | 13.51 | 11.06 | 17.30 | 53.1 | 53.7 | 71.8 | 57.4 | 52.5 | 28.7 | 35.2 | 50.3 | +3.2 |
| | 1.60B | ✓ | 15B | 2 | Lower | 512 | SVD | 12.54 | 10.22 | 15.90 | 57.1 | 58.2 | 73.7 | 58.6 | 57.6 | 31.5 | 35.6 | 53.2 | +6.1 |
| | 1.60B | ✓ | 15B | 2 | Lower | 512 | Zero | 13.95 | 11.59 | 18.02 | 48.4 | 52.1 | 71.9 | 55.7 | 54.9 | 28.8 | 34.6 | 49.5 | +2.4 |
| | 0.97B | ✓ | - | - | - | - | - | 12.26 | 9.37 | 11.94 | 43.3 | 42.2 | 66.8 | 53.4 | 44.7 | 23.2 | 29.2 | 43.3 | - |
| | 0.48B | ✗ | 15B | - | - | - | - | 16.61 | 15.66 | 20.27 | 22.3 | 30.0 | 60.9 | 50.6 | 37.0 | 23.0 | 28.0 | 36.0 | - |
| | 0.48B | ✗ | 15B | 2 | Rand | - | - | 16.14 | 15.11 | 19.55 | 24.7 | 30.7 | 61.2 | 50.6 | 36.4 | 22.6 | 29.2 | 36.5 | - |
| | 0.48B | ✓ | 15B | 2 | Step | - | - | 11.61 | 9.89 | 13.00 | 39.6 | 39.8 | 66.5 | 52.9 | 44.3 | 24.9 | 30.6 | 42.7 | - |
| | 0.53B | ✓ | 15B | 2 | Step | 64 | SVD | 12.10 | 10.40 | 13.75 | 38.9 | 38.3 | 65.2 | 51.5 | 42.0 | **26.0** | 31.0 | 41.9 | −0.8 |
| | 0.58B | ✓ | 15B | 2 | Step | 128 | SVD | 12.41 | 10.72 | 14.10 | 36.8 | 37.4 | 64.7 | 53.4 | 42.2 | 24.7 | 30.4 | 41.4 | −1.3 |
| | 0.68B | ✓ | 15B | 2 | Step | 256 | SVD | 11.96 | 10.35 | 13.48 | 38.9 | 38.3 | 65.8 | 51.9 | 43.1 | 25.4 | 29.8 | 41.9 | −0.8 |
| | 0.86B | ✓ | 15B | 2 | Step | 512 | SVD | 11.33 | 9.79 | 12.61 | 42.2 | 40.9 | 67.7 | 51.1 | 45.0 | 25.3 | 30.2 | 43.2 | +0.5 |
| | 0.86B | ✓ | 15B | 2 | Step | 512 | Zero | 11.24 | 9.60 | 12.56 | 42.0 | 41.0 | 67.4 | 52.2 | 44.5 | 25.9 | **31.2** | 43.4 | +0.7 |
| TinyLlama | 0.48B | ✓ | 15B | 2 | Avg | - | - | 11.86 | 10.29 | 13.42 | 38.6 | 39.4 | 66.1 | 52.8 | 42.7 | 25.4 | 30.6 | 42.2 | - |
| | 0.53B | ✓ | 15B | 2 | Avg | 64 | SVD | 11.22 | 9.66 | 12.51 | 41.8 | 41.6 | 67.0 | 53.3 | 43.9 | 24.7 | **31.2** | 43.4 | +1.2 |
| | 0.58B | ✓ | 15B | 2 | Avg | 128 | SVD | 10.99 | 9.45 | 12.21 | 43.2 | 42.1 | **68.3** | 53.2 | 44.8 | 25.9 | 30.4 | 44.0 | +1.8 |
| | 0.68B | ✓ | 15B | 2 | Avg | 256 | SVD | 10.71 | 9.18 | 11.82 | 44.1 | 43.2 | 68.1 | **53.5** | 44.4 | 25.7 | 30.4 | 44.2 | +2.0 |
| | 0.86B | ✓ | 15B | 2 | Avg | 512 | SVD | **10.46** | **8.92** | **11.50** | 46.0 | 44.1 | 68.2 | 53.0 | 45.8 | 25.1 | **31.2** | 44.8 | +2.6 |
| | 0.86B | ✓ | 15B | 2 | Avg | 512 | Zero | 11.28 | 9.75 | 12.69 | 41.5 | 41.0 | 66.8 | 53.2 | 44.8 | 25.5 | **31.2** | 43.4 | +1.2 |
| | 0.48B | ✓ | 15B | 2 | Lower | - | - | 14.67 | 12.67 | 16.68 | 31.9 | 32.3 | 62.6 | 52.0 | 39.1 | 22.1 | 27.8 | 38.3 | - |
| | 0.53B | ✓ | 15B | 2 | Lower | 64 | SVD | 13.68 | 11.77 | 15.48 | 35.5 | 34.0 | 63.8 | 51.0 | 40.0 | 24.6 | 28.0 | 39.5 | +1.2 |
| | 0.58B | ✓ | 15B | 2 | Lower | 128 | SVD | 13.00 | 11.14 | 14.61 | 37.6 | 35.4 | 65.3 | 51.5 | 40.4 | 24.5 | 27.6 | 40.3 | +2.0 |
| | 0.68B | ✓ | 15B | 2 | Lower | 256 | SVD | 12.14 | 10.39 | 13.59 | 40.0 | 37.7 | 66.1 | 52.6 | 42.5 | 24.8 | 30.0 | 42.0 | +3.7 |
| | 0.86B | ✓ | 15B | 2 | Lower | 512 | SVD | 11.31 | 9.61 | 12.49 | 43.2 | 40.5 | 66.0 | 50.8 | 43.9 | 24.8 | 30.0 | 42.8 | +4.5 |
| | 0.86B | ✓ | 15B | 2 | Lower | 512 | Zero | 14.56 | 12.69 | 16.57 | 21.2 | 32.9 | 63.9 | 52.6 | 39.5 | 22.9 | 27.8 | 37.3 | −1.0 |
| | 0.81B | ✓ | 15B | - | - | - | - | 13.46 | 9.95 | 13.38 | 55.0 | 49.0 | 71.0 | 53.6 | 51.8 | 28.2 | 32.8 | 48.8 | - |
| | 0.40B | ✗ | 15B | - | - | - | - | 25.69 | 20.00 | 32.08 | 24.3 | 30.0 | 61.9 | 50.7 | 38.3 | 22.3 | 26.0 | 36.2 | - |
| | 0.40B | ✗ | 15B | 2 | Rand | - | - | 24.45 | 18.93 | 29.63 | 25.2 | 30.2 | 62.1 | 51.1 | 39.2 | 22.4 | 23.6 | 36.2 | - |
| | 0.40B | ✓ | 15B | 2 | Step | - | - | 16.38 | 12.37 | 17.74 | 43.4 | 40.5 | 67.4 | 50.8 | 46.3 | 25.7 | 30.0 | 43.5 | - |
| | 0.44B | ✓ | 15B | 2 | Step | 64 | SVD | 16.44 | 12.44 | 17.89 | 43.7 | 40.4 | 66.5 | 52.9 | 46.5 | 26.2 | 28.8 | 43.6 | +0.1 |
| | 0.48B | ✓ | 15B | 2 | Step | 128 | SVD | 16.63 | 12.61 | 18.35 | 42.4 | 39.3 | 68.0 | 51.5 | 46.3 | 26.7 | 30.6 | 43.5 | +0.0 |
| | 0.55B | ✓ | 15B | 2 | Step | 256 | SVD | 16.54 | 12.61 | 18.39 | 42.8 | 39.1 | 67.2 | 53.7 | 46.4 | 25.9 | 27.8 | 43.3 | −0.2 |
| | 0.70B | ✓ | 15B | 2 | Step | 512 | SVD | 15.68 | 11.88 | 17.25 | 45.4 | 41.3 | **68.5** | 52.6 | 46.7 | 25.4 | 31.2 | 44.4 | +0.9 |
| | 0.70B | ✓ | 15B | 2 | Step | 512 | Zero | 15.88 | 12.01 | 17.16 | 45.5 | 41.8 | 68.0 | 52.6 | 46.6 | **26.3** | 30.0 | 44.4 | +0.9 |
| Pythia | 0.40B | ✓ | 15B | 2 | Avg | - | - | 16.76 | 12.76 | 18.63 | 43.6 | 39.1 | 68.2 | 51.9 | 45.4 | 25.1 | 29.8 | 43.3 | - |
| | 0.44B | ✓ | 15B | 2 | Avg | 64 | SVD | 16.03 | 12.19 | 17.59 | 45.8 | 40.9 | 67.3 | 50.0 | 45.8 | 25.5 | 31.8 | 43.9 | +0.6 |
| | 0.48B | ✓ | 15B | 2 | Avg | 128 | SVD | 15.67 | 11.93 | 17.10 | 46.9 | 41.9 | 67.4 | 51.2 | 45.4 | 24.8 | 31.2 | 44.1 | +0.8 |
| | 0.55B | ✓ | 15B | 2 | Avg | 256 | SVD | 15.22 | 11.54 | 16.47 | 48.5 | 43.3 | 67.2 | 51.4 | 46.7 | 25.5 | 32.0 | 44.9 | +1.6 |
| | 0.70B | ✓ | 15B | 2 | Avg | 512 | SVD | **14.70** | **11.07** | **15.71** | 50.2 | **44.7** | 68.2 | 51.6 | 47.6 | 25.4 | 31.2 | **45.6** | +2.3 |
| | 0.70B | ✓ | 15B | 2 | Avg | 512 | Zero | 15.97 | 12.14 | 17.65 | 45.7 | 41.5 | 68.1 | 51.7 | 46.5 | 25.7 | 30.0 | 44.2 | +0.9 |
| | 0.40B | ✓ | 15B | 2 | Lower | - | - | 17.04 | 12.62 | 18.44 | 43.9 | 39.2 | 66.3 | 53.4 | 45.4 | 25.8 | 31.2 | 43.6 | - |
| | 0.44B | ✓ | 15B | 2 | Lower | 64 | SVD | 17.03 | 12.78 | 18.73 | 44.1 | 38.3 | 66.9 | 51.9 | 45.4 | 24.7 | 30.8 | 43.2 | −0.4 |
| | 0.48B | ✓ | 15B | 2 | Lower | 128 | SVD | 16.63 | 12.49 | 18.17 | 45.2 | 39.2 | 66.8 | 51.0 | 45.2 | 24.9 | 29.6 | 43.2 | −0.4 |
| | 0.55B | ✓ | 15B | 2 | Lower | 256 | SVD | 15.93 | 11.99 | 17.30 | 47.6 | 41.4 | 68.3 | 53.2 | 46.0 | 25.8 | 31.0 | 44.8 | +1.2 |
| | 0.70B | ✓ | 15B | 2 | Lower | 512 | SVD | 15.11 | 11.34 | 16.07 | **50.2** | 43.5 | 67.8 | 51.8 | 47.2 | 25.2 | 32.0 | 45.4 | +1.8 |
| | 0.70B | ✓ | 15B | 2 | Lower | 512 | Zero | 16.45 | 12.25 | 17.76 | 45.2 | 40.4 | 66.4 | **54.5** | 45.8 | 25.9 | **32.6** | 44.4 | +0.8 |

# K    ALTERNATIVE APPROACHES TO RELAXATION OF PARAMETER SHARING

To mitigate the restrictive weight-tying inherent in parameter sharing, we employed LoRA modules as discussed in §2.3, similar to prior work (Ge et al., 2022; Shim et al., 2024). However, efficiently computing multiple LoRA modules requires specialized kernels and sequential computations of the base layers and LoRA modules, incurring computational overhead. Consequently, we explored layer-specific prompts (Liu et al., 2021) as an alternative. This approach integrates prompts specific to each layer as prefix tokens, generating layer-wise key and value states for self-attention, and is significantly more amenable to parallel computation.

Table 10 summarizes performance of the prefix tuning method. While offering computational advantages, its reliance on small, learnable prompts resulted in limited performance gains. Additionally, without leveraging the original pretrained weights, performance was significantly lower (52.1% vs. 47.6% with the Average method in 1.07B model size). Future work will explore enhancing the effectiveness of this parallel approach, as well as other strategies such as bias term-based relaxation (Ge et al., 2022).

Table 10: Evaluation results of relaxation through prefix tuning methods. Prefix length denotes the sequence length of trainable vectors used to generate key-value prompts in each self-attention layer. Non-embedding parameter sizes include the sizes of these trainable prefixes. Delta ($\Delta$) represent the accuracy differences between non-relaxed models and their corresponding prefix-tuned models using the same looping initialization.

| Models | N-emb | PT | $N_{tok}$ | Block | Init | Len | Size | SlimP | RedP | PG19 | LD | HS | PQ | WG | ARC-e | ARC-c | OB | Avg | $\Delta$ |
|---|---|---|---|---|---|---|---|---|---|---|---|---|---|---|---|---|---|---|---|
| | | | | | | | | | Perplexity↓ | | | | | | Few-shot Accuracy↑ | | | | |
| | 1.99B | ✓ | 15B | - | - | - | - | 10.76 | 8.47 | 13.08 | 63.5 | 68.5 | 77.0 | 63.5 | 67.6 | 38.1 | 42.6 | 60.1 | - |
| | 0.99B | ✗ | 15B | - | - | - | - | 22.63 | 20.03 | 32.60 | 28.9 | 31.6 | 63.1 | 52.3 | 41.2 | 22.5 | 27.8 | 38.2 | - |
| | 0.99B | ✓ | 15B | 2 | Step | - | - | 12.85 | 10.29 | 16.21 | 53.0 | 57.3 | 73.2 | 56.2 | 56.1 | 29.2 | 36.6 | 51.7 | - |
| | 1.00B | ✓ | 15B | 2 | Step | 256 | 9.4M | 12.62 | 10.06 | 15.80 | 53.5 | 58.3 | 73.9 | 57.6 | 57.5 | 29.3 | 35.6 | 52.2 | +0.5 |
| | 1.01B | ✓ | 15B | 2 | Step | 512 | 18.9M | 12.67 | 10.10 | 15.85 | 54.1 | 57.8 | 73.8 | 58.4 | 57.2 | 28.7 | 35.8 | 52.3 | +0.6 |
| | 1.03B | ✓ | 15B | 2 | Step | 1024 | 37.7M | 12.89 | 10.34 | 16.22 | 53.5 | 57.1 | 72.4 | 57.2 | 56.9 | 28.6 | 36.8 | 51.8 | +0.1 |
| | 1.07B | ✓ | 15B | 2 | Step | 2048 | 75.5M | 12.75 | 10.21 | 16.09 | 55.0 | 57.3 | 73.3 | 58.2 | 56.8 | 29.2 | 37.8 | 52.5 | +0.8 |
| Gemma | 0.99B | ✓ | 15B | 2 | Avg | - | - | 15.15 | 12.57 | 19.86 | 43.6 | 47.4 | 70.4 | 52.6 | 50.5 | 27.8 | 34.4 | 46.7 | - |
| | 1.00B | ✓ | 15B | 2 | Avg | 256 | 9.4M | 14.85 | 12.31 | 19.41 | 46.9 | 48.3 | 70.4 | 52.7 | 51.4 | 27.2 | 34.0 | 47.3 | +0.6 |
| | 1.01B | ✓ | 15B | 2 | Avg | 512 | 18.9M | 15.23 | 12.64 | 19.98 | 44.5 | 47.2 | 70.7 | 54.5 | 49.5 | 28.0 | 33.2 | 46.8 | +0.1 |
| | 1.03B | ✓ | 15B | 2 | Avg | 1024 | 37.7M | 14.60 | 12.02 | 18.89 | 46.9 | 49.7 | 71.1 | 52.3 | 51.0 | 28.6 | 34.2 | 47.7 | +1.0 |
| | 1.07B | ✓ | 15B | 2 | Avg | 2048 | 75.5M | 14.63 | 12.07 | 19.03 | 47.3 | 49.5 | 70.8 | 53.1 | 50.7 | 28.2 | 33.4 | 47.6 | +0.9 |
| | 0.99B | ✓ | 15B | 2 | Lower | - | - | 15.03 | 12.46 | 19.63 | 42.5 | 48.0 | 71.0 | 54.6 | 52.2 | 27.7 | 33.8 | 47.1 | - |
| | 1.00B | ✓ | 15B | 2 | Lower | 256 | 9.4M | 14.59 | 12.12 | 19.02 | 46.3 | 49.7 | 71.5 | 55.1 | 52.9 | 29.0 | 34.0 | 48.4 | +1.3 |
| | 1.01B | ✓ | 15B | 2 | Lower | 512 | 18.9M | 14.53 | 12.03 | 18.88 | 45.7 | 49.8 | 71.9 | 56.4 | 53.6 | 29.4 | 33.2 | 48.6 | +1.5 |
| | 1.03B | ✓ | 15B | 2 | Lower | 1024 | 37.7M | 14.43 | 11.98 | 18.74 | 46.3 | 50.0 | 71.9 | 55.1 | 54.3 | 29.7 | 33.8 | 48.7 | +1.6 |
| | 1.07B | ✓ | 15B | 2 | Lower | 2048 | 75.5M | 14.79 | 12.26 | 19.23 | 46.1 | 48.7 | 71.4 | 55.4 | 51.3 | 28.2 | 34.0 | 47.9 | +0.8 |
| | 0.97B | ✓ | - | - | - | - | - | 12.26 | 9.37 | 11.94 | 43.3 | 42.2 | 66.8 | 53.4 | 44.7 | 23.2 | 29.2 | 43.3 | - |
| | 0.48B | ✗ | 15B | - | - | - | - | 16.61 | 15.66 | 20.27 | 22.3 | 30.0 | 60.9 | 50.6 | 37.0 | 23.0 | 28.0 | 36.0 | - |
| | 0.48B | ✓ | 15B | 2 | Step | - | - | 11.61 | 9.89 | 13.00 | 39.6 | 39.8 | 66.5 | 52.9 | 44.3 | 24.9 | 30.6 | 42.7 | - |
| | 0.49B | ✓ | 15B | 2 | Step | 256 | 11.5M | 11.61 | 9.89 | 13.00 | 39.6 | 39.9 | 66.5 | 53.9 | 44.4 | 25.3 | 30.6 | 42.9 | +0.2 |
| | 0.50B | ✓ | 15B | 2 | Step | 512 | 23.1M | 11.61 | 9.89 | 13.01 | 39.5 | 39.9 | 66.7 | 53.4 | 44.1 | 25.3 | 29.8 | 42.7 | +0.0 |
| | 0.53B | ✓ | 15B | 2 | Step | 1024 | 46.1M | 11.60 | 9.88 | 13.00 | 39.7 | 39.9 | 66.7 | 53.0 | 44.3 | 25.1 | 30.6 | 42.8 | +0.1 |
| | 0.57B | ✓ | 15B | 2 | Step | 2048 | 92.3M | 11.58 | 9.87 | 13.01 | 40.1 | 39.9 | 66.8 | 53.4 | 44.4 | 24.9 | 30.0 | 42.8 | +0.1 |
| TinyLlama | 0.48B | ✓ | 15B | 2 | Avg | - | - | 11.86 | 10.29 | 13.42 | 38.6 | 39.4 | 66.1 | 52.8 | 42.7 | 25.4 | 30.6 | 42.2 | - |
| | 0.49B | ✓ | 15B | 2 | Avg | 256 | 11.5M | 11.86 | 10.28 | 13.41 | 38.5 | 39.4 | 66.2 | 52.5 | 42.8 | 25.9 | 30.8 | 42.3 | +0.1 |
| | 0.50B | ✓ | 15B | 2 | Avg | 512 | 23.1M | 11.86 | 10.28 | 13.41 | 38.1 | 39.3 | 66.3 | 52.2 | 42.6 | 25.6 | 30.8 | 42.1 | −0.1 |
| | 0.53B | ✓ | 15B | 2 | Avg | 1024 | 46.1M | 11.86 | 10.28 | 13.42 | 38.4 | 39.2 | 65.7 | 52.7 | 42.5 | 25.5 | 31.0 | 42.1 | −0.1 |
| | 0.57B | ✓ | 15B | 2 | Avg | 2048 | 92.3M | 11.86 | 10.28 | 13.42 | 38.5 | 39.5 | 65.9 | 52.7 | 42.4 | 25.7 | 31.0 | 42.2 | +0.0 |
| | 0.48B | ✓ | 15B | 2 | Lower | - | - | 14.67 | 12.67 | 16.68 | 31.9 | 32.3 | 62.6 | 52.0 | 39.1 | 22.1 | 27.8 | 38.3 | - |
| | 0.49B | ✓ | 15B | 2 | Lower | 256 | 11.5M | 14.67 | 12.67 | 16.69 | 31.9 | 32.4 | 62.7 | 51.5 | 38.9 | 22.3 | 27.8 | 38.2 | −0.1 |
| | 0.50B | ✓ | 15B | 2 | Lower | 512 | 23.1M | 14.67 | 12.67 | 16.69 | 31.9 | 32.3 | 62.8 | 51.7 | 38.9 | 22.2 | 27.8 | 38.2 | −0.1 |
| | 0.53B | ✓ | 15B | 2 | Lower | 1024 | 46.1M | 14.67 | 12.67 | 16.68 | 31.6 | 32.3 | 63.0 | 51.9 | 38.9 | 22.1 | 28.0 | 38.3 | +0.0 |
| | 0.57B | ✓ | 15B | 2 | Lower | 2048 | 92.3M | 14.67 | 12.67 | 16.67 | 34.1 | 32.5 | 62.8 | 52.4 | 38.5 | 23.0 | 27.6 | 38.7 | +0.4 |

## L  EXPANDED RESULTS OF EXTENDED UPTRAINING AND DISTILLATION

**Ablation study on individual techniques**  To further enhance performance through uptraining, we increased the number of uptraining tokens and employed knowledge distillation loss (Hinton et al., 2015; Kim & Rush, 2016). Specifically, we expanded the token number from 15 billion to 60 billion. Furthermore, we designated the teacher model as the full-size model for each architecture, uptrained on 15 billion tokens from the SlimPajama dataset. Given the huge number of uptraining tokens, we adopted an online approach to extract logits from the teacher model. Four loss functions were utilized: forward KL (FKL; Kim & Rush (2016)), reverse KL (RKL; Gu et al. (2024)), Jensen–Shannon divergence (JSD; Agarwal et al. (2024)), and total variation distance (TVD; Wen et al. (2023)). Table 11 summarizes the controlled experimental results for each method. We observed a performance improvement of 1.7% attributed to the extended uptraining and up to 1.7% from the KD loss. We finally selected forward KL as the loss function due to its simplicity and superior performance. These significant gains suggest that combining both techniques could yield even greater gains.

Table 11: Evaluation results of ablation studies related to longer uptraining and knowledge distillation loss. Performance improvements, represented by Delta, were measured for each technique. For the knowledge distillation loss function, we experimented with four options: FKL, RKL, JSD, and TVD. Forward KL was selected as the final configuration due to its simplicity and superior performance.

| | | Uptrain | | | Looping | | LoRA | | Perplexity↓ | | | | Few-shot Accuracy↑ | | | | | | | | |
|---|---|---|---|---|---|---|---|---|---|---|---|---|---|---|---|---|---|---|---|---|---|
| N-emb | PT | $N_{tok}$ | KD | Func | Block | Init | Rank | Init | SlimP | RedP | PG19 | LD | HS | PQ | WG | ARC-e | ARC-c | OB | Avg | Δ |
| 0.99B | ✓ | 15B | ✗ | - | 2 | Step | - | - | 12.85 | 10.29 | 16.21 | 53.0 | 57.3 | 73.2 | 56.2 | 56.1 | 29.2 | 36.6 | 51.7 | - |
| 0.99B | ✓ | 60B | ✗ | - | 2 | Step | - | - | 12.00 | 9.70 | 14.84 | 52.5 | 59.9 | 74.7 | 58.5 | 58.0 | 30.3 | 40.2 | 53.4 | **+1.7** |
| 0.99B | ✓ | 15B | ✗ | - | 2 | Step | - | - | 12.85 | 10.29 | 16.21 | 53.0 | 57.3 | 73.2 | 56.2 | 56.1 | 29.2 | 36.6 | 51.7 | - |
| 0.99B | ✓ | 15B | ✓ | FKL | 2 | Step | - | - | 12.36 | 9.85 | 15.45 | 56.8 | 58.6 | 74.8 | 58.6 | 59.1 | 29.2 | 36.6 | 53.4 | **+1.7** |
| 0.99B | ✓ | 15B | ✓ | RKL | 2 | Step | - | - | 12.56 | 10.09 | 15.80 | 55.6 | 58.3 | 74.3 | 58.6 | 58.3 | 30.4 | 37.4 | 53.3 | **+1.6** |
| 0.99B | ✓ | 15B | ✓ | JSD | 2 | Step | - | - | 12.60 | 10.06 | 15.77 | 56.1 | 58.4 | 73.4 | 57.0 | 58.4 | 29.8 | 37.2 | 52.9 | **+1.2** |
| 0.99B | ✓ | 15B | ✓ | TVD | 2 | Step | - | - | 12.47 | 9.92 | 15.52 | 55.1 | 58.5 | 74.0 | 58.2 | 58.9 | 29.5 | 36.8 | 53.0 | **+1.3** |
| 1.30B | ✓ | 15B | ✗ | - | 2 | Avg | 256 | SVD | 12.10 | 9.71 | 14.89 | 58.2 | 60.7 | 73.7 | 59.0 | 57.6 | 32.1 | 38.0 | 54.2 | - |
| 1.30B | ✓ | 15B | ✓ | FKL | 2 | Avg | 256 | SVD | 11.90 | 9.52 | 14.63 | 59.9 | 62.0 | 74.1 | 60.0 | 58.6 | 33.2 | 38.0 | 55.1 | **+0.9** |
| 1.30B | ✓ | 15B | ✓ | RKL | 2 | Avg | 256 | SVD | 11.95 | 9.62 | 14.79 | 60.0 | 61.6 | 74.5 | 60.0 | 58.1 | 32.9 | 37.8 | 55.0 | **+0.8** |
| 1.30B | ✓ | 15B | ✓ | JSD | 2 | Avg | 256 | SVD | 12.09 | 9.65 | 14.81 | 58.1 | 61.1 | 73.1 | 60.8 | 59.0 | 33.2 | 38.6 | 54.8 | **+0.6** |
| 1.30B | ✓ | 15B | ✓ | TVD | 2 | Avg | 256 | SVD | 12.05 | 9.62 | 14.78 | 59.3 | 61.5 | 73.9 | 60.5 | 59.0 | 33.0 | 38.2 | 55.1 | **+0.9** |

**Overall performance after longer training with distillation loss**  Figure 17 and Table 12 summarize the performance gains achieved by incorporating advanced training techniques: extended uptraining and knowledge distillation loss. We consistently observed substantial improvements in few-shot performance across all architectures and with varying numbers of looping blocks. We anticipate that further performance enhancements can be achieved by utilizing a superior teacher model and increasing the uptraining cost.

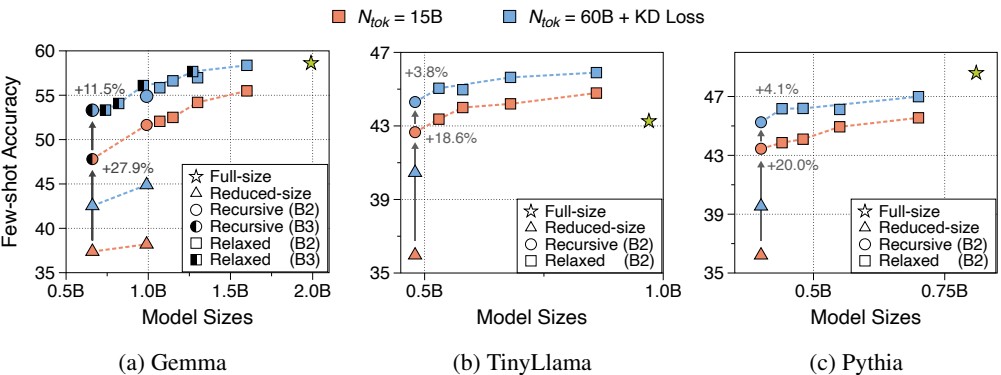

Figure 17: Few-shot performance of three models with extended uptraining and knowledge distillation. Optimal configurations are used for each model size. The full-size model is the pretrained model itself for Tinyllama, but for other models, it is further uptrained on 60 billion tokens. Reduced-size models are non-recursive and pretrained from scratch at their corresponding sizes. Dotted lines represent the Pareto frontier, showing the optimal trade-offs between model size and performance for each setting.

Table 12: Evaluation results of our Recursive Transformers with 60 billion token uptraining and knowledge distillation loss. We utilized the forward KL loss as the knowledge distillation loss function. Full-size model baselines for Gemma and Pythia are the pretrained models further uptrained on 60 billion tokens, accounting for distribution shifts between Slimapajama and their pretraining datasets. Delta ($\Delta$) represents the accuracy differences between the longer uptrained models with KD and their 15 billion uptrained counterparts. We omit the Delta values for relaxed recursive Gemma models with three blocks as they lack 15 billion uptrained counterparts. Results with extended uptraining and knowledge distillation are highlighted.

| Models | N-emb | PT | $N_{tok}$ | KD | Block | Init | Rank | Init | SlimP | RedP | PG19 | LD | HS | PQ | WG | ARC-e | ARC-c | OB | Avg | $\Delta$ |
|---|---|---|---|---|---|---|---|---|---|---|---|---|---|---|---|---|---|---|---|---|
| | | | **Uptrain** | | **Looping** | | **LoRA** | | **Perplexity↓** | | | **Few-shot Accuracy↑** | | | | | | | | |
| Gemma | 1.99B | ✓ | 60B | ✗ | - | - | - | - | 10.58 | 8.44 | 12.71 | 60.3 | 67.9 | 76.9 | 63.5 | 64.9 | 37.2 | 39.6 | 58.6 | - |
| | 0.99B | ✗ | 60B | ✓ | - | - | - | - | 15.33 | 13.04 | 20.37 | 42.3 | 43.0 | 68.8 | 53.4 | 49.4 | 26.3 | 31.0 | 44.9 | - |
| | 0.99B | ✗ | 15B | ✗ | - | - | - | - | 22.63 | 20.03 | 32.60 | 28.9 | 31.6 | 63.1 | 52.3 | 41.2 | 22.5 | 27.8 | 38.2 | - |
| | 0.66B | ✗ | 60B | ✓ | - | - | - | - | 16.79 | 14.39 | 22.85 | 37.5 | 38.4 | 68.7 | 50.4 | 46.5 | 24.6 | 31.6 | 42.5 | - |
| | 0.66B | ✗ | 15B | ✗ | - | - | - | - | 24.44 | 21.69 | 36.03 | 27.2 | 30.6 | 63.8 | 50.5 | 40.6 | 22.0 | 27.0 | 37.4 | - |
| | 0.99B | ✓ | 15B | ✗ | 2 | Step | - | - | 12.85 | 10.29 | 16.21 | 53.0 | 57.3 | 73.2 | 56.2 | 56.1 | 29.2 | 36.6 | 51.7 | - |
| | 0.66B | ✓ | 15B | ✗ | 3 | Step | - | - | 14.75 | 12.10 | 19.32 | 45.0 | 49.9 | 69.8 | 55.8 | 52.7 | 27.9 | 33.6 | 47.8 | - |
| | 1.07B | ✓ | 15B | ✗ | 2 | Avg | 64 | SVD | 12.83 | 10.35 | 16.02 | 55.9 | 56.8 | 72.5 | 56.8 | 55.7 | 30.6 | 36.2 | 52.1 | - |
| | 1.15B | ✓ | 15B | ✗ | 2 | Avg | 128 | SVD | 12.52 | 10.07 | 15.51 | 56.1 | 58.2 | 72.3 | 55.8 | 57.1 | 30.7 | 37.2 | 52.5 | - |
| | 1.30B | ✓ | 15B | ✗ | 2 | Avg | 256 | SVD | 12.10 | 9.71 | 14.89 | 58.2 | 60.7 | 73.7 | 59.0 | 57.6 | 32.1 | 38.0 | 54.2 | - |
| | 1.60B | ✓ | 15B | ✗ | 2 | Avg | 512 | SVD | 11.83 | 9.46 | 14.57 | 59.3 | 62.8 | 74.0 | 61.6 | 60.1 | 32.9 | 37.6 | 55.5 | - |
| | 0.99B | ✓ | 60B | ✓ | 2 | Step | - | - | 11.44 | 9.14 | 13.98 | 56.5 | 62.1 | 75.2 | 59.4 | 59.8 | 32.5 | 38.6 | 54.9 | **+3.2** |
| | 1.07B | ✓ | 60B | ✓ | 2 | Avg | 64 | SVD | 11.36 | 9.14 | 13.82 | 58.9 | 62.8 | 75.1 | 61.5 | 61.2 | 33.7 | 37.6 | 55.8 | **+3.7** |
| | 1.15B | ✓ | 60B | ✓ | 2 | Avg | 128 | SVD | 11.25 | 9.04 | 13.64 | 58.7 | 63.6 | 76.5 | 61.2 | 62.6 | 34.6 | 39.0 | 56.6 | **+4.1** |
| | 1.30B | ✓ | 60B | ✓ | 2 | Avg | 256 | SVD | 11.05 | 8.88 | 13.35 | 60.6 | 64.7 | 75.3 | 62.5 | 61.6 | 35.3 | 38.8 | 57.0 | **+2.8** |
| | 1.60B | ✓ | 60B | ✓ | 2 | Avg | 512 | SVD | **10.81** | **8.63** | **12.94** | 61.4 | **65.8** | **76.3** | **63.5** | **65.1** | **37.1** | 39.4 | **58.4** | **+2.9** |
| | 0.66B | ✓ | 60B | ✓ | 3 | Step | - | - | 12.27 | 9.90 | 15.24 | 55.6 | 58.1 | 73.1 | 60.2 | 58.8 | 30.2 | 37.2 | 53.3 | **+5.5** |
| | 0.74B | ✓ | 60B | ✓ | 3 | Avg | 64 | SVD | 12.13 | 9.80 | 14.95 | 55.5 | 58.3 | 73.5 | 60.1 | 58.0 | 31.1 | 36.8 | 53.3 | - |
| | 0.82B | ✓ | 60B | ✓ | 3 | Avg | 128 | SVD | 11.83 | 9.53 | 14.51 | 56.7 | 60.2 | 74.2 | 59.8 | 59.1 | 33.0 | 35.4 | 54.1 | - |
| | 0.97B | ✓ | 60B | ✓ | 3 | Avg | 256 | SVD | 11.43 | 9.17 | 13.87 | 59.3 | 62.6 | 74.7 | 61.2 | 61.6 | 32.9 | **40.2** | 56.1 | - |
| | 1.27B | ✓ | 60B | ✓ | 3 | Avg | 512 | SVD | 11.01 | 8.80 | 13.25 | **61.5** | 64.9 | 76.2 | 62.0 | 64.3 | 35.6 | 39.2 | 57.7 | - |
| TinyLlama | 0.97B | ✓ | - | - | - | - | - | - | 12.26 | 9.37 | 11.94 | 43.3 | 42.2 | 66.8 | 53.4 | 44.7 | 23.2 | 29.2 | 43.3 | - |
| | 0.48B | ✗ | 60B | ✓ | - | - | - | - | 11.93 | 10.86 | 13.93 | 33.3 | 37.3 | 66.8 | 50.1 | 41.7 | 23.9 | 30.2 | 40.5 | - |
| | 0.48B | ✗ | 15B | ✗ | - | - | - | - | 16.61 | 15.66 | 20.27 | 22.3 | 30.0 | 60.9 | 50.6 | 37.0 | 23.0 | 28.0 | 36.0 | - |
| | 0.48B | ✓ | 15B | ✗ | 2 | Step | - | - | 11.61 | 9.89 | 13.00 | 39.6 | 39.8 | 66.5 | 52.9 | 44.3 | 24.9 | 30.6 | 42.7 | - |
| | 0.53B | ✓ | 15B | ✗ | 2 | Avg | 64 | SVD | 11.22 | 9.66 | 12.51 | 41.8 | 41.6 | 67.0 | 53.3 | 43.9 | 24.7 | 31.2 | 43.4 | - |
| | 0.58B | ✓ | 15B | ✗ | 2 | Avg | 128 | SVD | 10.99 | 9.45 | 12.21 | 43.2 | 42.1 | 68.3 | 53.2 | 44.8 | 25.9 | 30.4 | 44.0 | - |
| | 0.68B | ✓ | 15B | ✗ | 2 | Avg | 256 | SVD | 10.71 | 9.18 | 11.82 | 44.1 | 43.2 | 68.1 | 53.5 | 44.4 | 25.7 | 31.2 | 44.2 | - |
| | 0.86B | ✓ | 15B | ✗ | 2 | Avg | 512 | SVD | 10.46 | 8.92 | 11.50 | 46.0 | 44.1 | 68.2 | 53.0 | 45.8 | 25.1 | 31.2 | 44.8 | - |
| | 0.48B | ✓ | 60B | ✓ | 2 | Step | - | - | 10.51 | 9.01 | 11.60 | 44.2 | 43.1 | 68.2 | 52.4 | 44.7 | 25.3 | 32.2 | 44.3 | **+1.6** |
| | 0.53B | ✓ | 60B | ✓ | 2 | Avg | 64 | SVD | 10.14 | 8.77 | 11.19 | 44.3 | 44.9 | 69.5 | 52.5 | 46.5 | 26.1 | **31.6** | 45.1 | **+1.6** |
| | 0.58B | ✓ | 60B | ✓ | 2 | Avg | 128 | SVD | 10.07 | 8.68 | 11.07 | 45.9 | 45.1 | 69.4 | 50.5 | 46.8 | 25.4 | **31.6** | 45.0 | **+1.0** |
| | 0.68B | ✓ | 60B | ✓ | 2 | Avg | 256 | SVD | 9.96 | 8.56 | 10.93 | 46.2 | 45.7 | 69.0 | 53.2 | **47.9** | 25.9 | **31.6** | 45.6 | **+1.4** |
| | 0.86B | ✓ | 60B | ✓ | 2 | Avg | 512 | SVD | **9.85** | **8.44** | **10.76** | **47.4** | **46.3** | **69.7** | 52.8 | 47.5 | **26.3** | 31.4 | **45.9** | **+1.1** |
| Pythia | 0.81B | ✓ | 60B | ✗ | - | - | - | - | 12.83 | 9.76 | 13.57 | 53.0 | 50.2 | 71.1 | 54.8 | 51.9 | 27.7 | 31.6 | 48.6 | - |
| | 0.40B | ✗ | 60B | ✗ | - | - | - | - | 18.27 | 14.39 | 21.93 | 32.1 | 35.0 | 66.1 | 49.6 | 42.9 | 24.2 | 27.0 | 39.5 | - |
| | 0.40B | ✗ | 15B | ✗ | - | - | - | - | 25.69 | 20.00 | 32.08 | 24.3 | 30.0 | 61.9 | 50.7 | 38.3 | 22.3 | 26.0 | 36.2 | - |
| | 0.40B | ✓ | 15B | ✗ | 2 | Step | - | - | 16.38 | 12.37 | 17.74 | 43.4 | 40.5 | 67.4 | 50.8 | 46.3 | 25.7 | 30.0 | 43.5 | - |
| | 0.44B | ✓ | 15B | ✗ | 2 | Avg | 64 | SVD | 16.03 | 12.19 | 17.59 | 45.8 | 40.9 | 67.3 | 50.0 | 45.8 | 25.5 | 31.8 | 43.9 | - |
| | 0.48B | ✓ | 15B | ✗ | 2 | Avg | 128 | SVD | 15.67 | 11.93 | 17.10 | 46.9 | 41.9 | 67.4 | 51.2 | 45.4 | 24.8 | 31.2 | 44.1 | - |
| | 0.55B | ✓ | 15B | ✗ | 2 | Avg | 256 | SVD | 15.22 | 11.54 | 16.47 | 48.5 | 43.3 | 67.2 | 51.4 | 46.7 | 25.5 | 32.0 | 44.9 | - |
| | 0.70B | ✓ | 15B | ✗ | 2 | Avg | 512 | SVD | 14.70 | 11.07 | 15.71 | 50.2 | 44.7 | 68.2 | 51.6 | 47.6 | 25.4 | 31.2 | 45.6 | - |
| | 0.40B | ✓ | 60B | ✓ | 2 | Step | - | - | 14.59 | 11.13 | 15.79 | 47.8 | 43.8 | 69.3 | 52.0 | 48.1 | 25.4 | 30.4 | 45.2 | **+1.7** |
| | 0.44B | ✓ | 60B | ✓ | 2 | Avg | 64 | SVD | 14.24 | 10.89 | 15.52 | 50.0 | 44.5 | 68.9 | **54.1** | 48.0 | 26.5 | 31.2 | 46.2 | **+2.3** |
| | 0.48B | ✓ | 60B | ✓ | 2 | Avg | 128 | SVD | 14.10 | 10.79 | 15.27 | 50.1 | 45.5 | 69.0 | 52.6 | 48.3 | 25.8 | 32.0 | 46.2 | **+2.1** |
| | 0.55B | ✓ | 60B | ✓ | 2 | Avg | 256 | SVD | 13.91 | 10.61 | 14.91 | 50.5 | 45.6 | 68.7 | 51.2 | 48.4 | 25.7 | **32.8** | 46.1 | **+1.2** |
| | 0.70B | ✓ | 60B | ✓ | 2 | Avg | 512 | SVD | **13.59** | **10.38** | **14.43** | **52.0** | **47.0** | **69.6** | 53.4 | **48.9** | **26.9** | 31.2 | **47.0** | **+1.4** |

# M    EARLY-EXIT TRAINING

**Ablation study on early-exit training strategy**   To enable early-exiting capabilities, all models require additional training to align intermediate representations with classifier heads. In this study, we conduct ablation studies on various strategies, demonstrating Recursive Transformers can be transformed into early-exiting models without compromising final loop output's performance. Table 13 presents a comprehensive summary of our findings across various categories, including training procedures, loss functions, and early-exit training data. Our key findings are as follows:

- Post-training after uptraining is essential for preserving final loop performance. Jointly training intermediate loop output during the uptraining phase, even with an aggressive loss coefficient strategy, significantly degraded the final output performance.

- Training solely the early loops with learnable LoRA modules, while freezing other parameters, hindered effective intermediate representation learning. We attempted to fine-tune intermediate outputs by attaching LoRA modules to classifier heads, but this proved ineffective.

- The aggressive coefficient strategy successfully maintained final loop output performance while enhancing intermediate layer performance. Moreover, incorporating knowledge distillation from detached final outputs further enhanced intermediate layer performance.

- No significant performance differences were observed when using the same uptraining data versus new SlimPajama tokens for post-training.

Finally, we opted to utilize the uptrained model and perform post-training with new tokens sourced from the same SlimPajama dataset. Moreover, we incorporated a distillation loss from the final loop output, while using a strategy that aggressively reduces the loss coefficients of intermediate outputs.

Table 13: Ablation studies on early-exit training for recursive Gemma models. We evaluated performance in a static-exiting scenario (Schuster et al., 2022; Bae et al., 2023), where all tokens exit at either 9th or 18th layers. We explored post-training (after uptraining) and co-training (during uptraining) approaches. We experimented with freezing uptrained weights and adding LoRA with the rank of 128 to the classifier head, and we used weighted CE and aggressive CE loss functions. Early-exit training utilized 15 billion tokens, either overlapping with uptraining data or entirely new. Delta ($\Delta$) indicates the performance changes of the final layer. We highlight the final configuration: post-training with aggressive CE and KD loss on 15 billion new tokens.

| | Uptrain | | Looping | | Early-Exit Train | | | | | | Perplexity ↓ | | | Few-shot Accuracy ↑ | | | | | | | | |
|---|---|---|---|---|---|---|---|---|---|---|---|---|---|---|---|---|---|---|---|---|---|---|
| N-emb | PT | $N_{tok}$ | Block | Init | Train | Freeze | $N_{tok}$ | CE | KD | Data | SlimP | RedP | PG19 | LD | HS | PQ | WG | ARC-e | ARC-c | OB | Avg | $\Delta$ |
| 1.99B | ✓ | 15B | - | - | - | - | - | - | - | - | 10.76 | 8.47 | 13.08 | 63.5 | 68.5 | 77.0 | 63.5 | 67.6 | 38.1 | 42.6 | 60.1 | - |
| 0.99B | ✗ | 15B | - | - | - | - | - | - | - | - | 22.63 | 20.03 | 32.60 | 28.9 | 31.6 | 63.1 | 52.3 | 41.2 | 22.5 | 27.8 | 38.2 | - |
| 0.99B | ✓ | 15B | 2 | Step | - | - | - | - | - | - | 12.85 | 10.29 | 16.21 | 53.0 | 57.3 | 73.2 | 56.2 | 56.1 | 29.2 | 36.6 | 51.7 | - |
| 0.99B | ✓ | 15B | 2 | Step | Post- | ✗ | 15B | Weighted | ✗ | Ovlp | 12.97 | 10.51 | 16.55 | 48.9 | 55.5 | 72.7 | 55.3 | 54.9 | 30.1 | 36.0 | 50.5 | −1.2 |
| | | | | | | | | | | | 13.11 | 10.59 | 16.71 | 49.5 | 54.8 | 72.0 | 53.4 | 54.1 | 29.1 | 35.6 | 49.8 | - |
| 0.99B | ✓ | 15B | 2 | Step | Post- | ✗ | 15B | Agg (0.3) | ✗ | Ovlp | 12.60 | 10.21 | 15.75 | 51.8 | 58.2 | 73.7 | 56.8 | 57.0 | 29.9 | 37.8 | 52.2 | +0.5 |
| | | | | | | | | | | | 13.63 | 11.02 | 17.55 | 47.5 | 53.0 | 71.2 | 54.9 | 50.2 | 28.2 | 34.8 | 48.5 | - |
| 0.99B | ✓ | 15B | 2 | Step | Post- | ✗ | 15B | Agg (0.1) | ✗ | Ovlp | 12.37 | 9.94 | 15.37 | 53.0 | 59.1 | 73.9 | 55.4 | 57.4 | 30.6 | 37.8 | 52.5 | +0.8 |
| | | | | | | | | | | | 14.55 | 11.87 | 19.00 | 45.9 | 51.2 | 71.4 | 54.5 | 48.1 | 26.8 | 32.0 | 47.1 | - |
| 0.99B | ✓ | 15B | 2 | Step | Post- | ✗ | 15B | Agg (0.05) | ✗ | Ovlp | 12.33 | 9.90 | 15.31 | 52.8 | 59.2 | 73.6 | 57.5 | 57.7 | 30.5 | 37.2 | 52.6 | +0.9 |
| | | | | | | | | | | | 15.70 | 12.93 | 20.69 | 43.1 | 49.8 | 69.8 | 55.2 | 46.0 | 26.9 | 31.2 | 46.0 | - |
| 0.99B | ✓ | 15B | 2 | Step | Post- | ✗ | 15B | Agg (0.01) | ✗ | Ovlp | 12.28 | 9.80 | 15.23 | 52.9 | 59.5 | 73.3 | 56.5 | 57.2 | 30.1 | 37.2 | 52.4 | +0.7 |
| | | | | | | | | | | | 22.76 | 20.37 | 30.39 | 32.2 | 45.2 | 67.5 | 53.9 | 40.3 | 26.3 | 29.2 | 42.1 | - |
| 0.99B | ✓ | 15B | 2 | Step | Post- | ✗ | 15B | Weighted | ✓ | Ovlp | 13.04 | 10.57 | 16.66 | 47.7 | 55.1 | 73.2 | 55.6 | 54.5 | 29.1 | 37.2 | 50.4 | −1.3 |
| | | | | | | | | | | | 13.04 | 10.54 | 16.66 | 48.3 | 54.9 | 72.1 | 55.9 | 54.3 | 28.4 | 35.4 | 49.9 | - |
| 0.99B | ✓ | 15B | 2 | Step | Post- | ✗ | 15B | Agg (0.1) | ✓ | Ovlp | 12.40 | 9.97 | 15.42 | 52.9 | 58.9 | 73.7 | 55.7 | 57.5 | 31.1 | 38.2 | 52.6 | +0.9 |
| | | | | | | | | | | | 14.11 | 11.47 | 18.32 | 46.3 | 52.1 | 71.6 | 55.3 | 49.2 | 28.5 | 32.6 | 48.0 | - |
| 0.99B | ✓ | 15B | 2 | Step | Post- | ✓ | 15B | Standard | ✗ | Ovlp | 12.85 | 10.29 | 16.21 | 53.0 | 57.3 | 73.2 | 56.2 | 56.1 | 29.2 | 36.6 | 51.7 | +0.0 |
| | | | | | | | | | | | 43.74 | 41.63 | 56.78 | 5.3 | 37.9 | 61.4 | 52.6 | 35.3 | 24.0 | 29.0 | 35.0 | - |
| 0.99B | ✓ | 15B | 2 | Step | Post- | ✓ | 15B | Standard | ✓ | Ovlp | 12.85 | 10.29 | 16.21 | 53.0 | 57.3 | 73.2 | 56.2 | 56.1 | 29.2 | 36.6 | 51.7 | +0.0 |
| | | | | | | | | | | | 43.09 | 39.97 | 55.37 | 5.6 | 37.7 | 62.5 | 52.7 | 34.5 | 24.7 | 29.2 | 35.3 | - |
| 0.99B | ✓ | 15B | 2 | Step | Co- | ✗ | 15B | Agg (0.1) | ✗ | Ovlp | 13.24 | 10.67 | 16.98 | 50.1 | 54.2 | 72.2 | 53.7 | 54.7 | 28.9 | 37.4 | 50.2 | −1.5 |
| | | | | | | | | | | | 13.59 | 10.89 | 17.42 | 50.6 | 52.7 | 71.2 | 54.4 | 53.0 | 27.5 | 35.0 | 49.2 | - |
| 0.99B | ✓ | 15B | 2 | Step | Post- | ✗ | 15B | Agg (0.1) | ✗ | New | 12.34 | 9.92 | 15.31 | 52.3 | 59.0 | 73.8 | 57.6 | 55.5 | 30.4 | 37.2 | 52.3 | +0.6 |
| | | | | | | | | | | | 14.49 | 11.86 | 18.89 | 43.9 | 51.3 | 71.0 | 54.9 | 48.1 | 27.5 | 31.4 | 46.9 | - |

**Early-exit training results on final models** We applied the aggressive coefficient strategy with distillation loss to the models uptrained on 60 billion tokens. Tables 14 and 15 summarize the performance of intermediate loops and the final loop across three models. For fair comparison, the full-size models (Gemma and Pythia) were also uptrained with 60 billion tokens and then post-trained with 15 billion tokens. As the optimal strategy derived from the non-relaxed models was directly applied to the relaxed models, further exploration of optimal strategies specifically for relaxed models is left for future work.

Table 14: Evaluation results of Gemma models after early-exit training. For relaxed models, we also experimented with increasing the coefficient to 0.3 because of the lower performance of the intermediate layer. The relaxed model with three blocks shows a more significant performance drop because KD loss could not be utilized due to out-of-memory issues. Delta ($\Delta$) represent the accuracy changes of the original last layer after early-exit post-training. These changes should be compared in reference to the performance drops observed in 75B and 60B uptraining for the full-size model.

| N-emb | PT | $N_{tok}$ | KD | Block | Init | Rank | Init | $N_{tok}$ | CE | KD | SlimP | RedP | PG19 | LD | HS | PQ | WG | ARC-e | ARC-c | OB | Avg | $\Delta$ |
|---|---|---|---|---|---|---|---|---|---|---|---|---|---|---|---|---|---|---|---|---|---|---|
| | | Uptrain | | Looping | | LoRA | | Early-Exit Train | | | Perplexity ↓ | | | Few-shot Accuracy ↑ | | | | | | | | |
| 1.99B | ✓ | 60B | ✗ | - | - | - | - | - | - | - | 10.58 | 8.44 | 12.71 | 60.3 | 67.9 | 76.9 | 63.5 | 64.9 | 37.2 | 39.6 | 58.6 | - |
| 1.99B | ✓ | 75B | ✗ | - | - | - | - | - | - | - | 11.03 | 8.88 | 13.33 | 57.0 | 65.9 | 76.2 | 63.9 | 63.0 | 35.9 | 38.8 | 57.3 | −1.3 |
| 0.99B | ✓ | 60B | ✓ | 2 | Step | - | - | - | - | - | 11.44 | 9.14 | 13.98 | 56.5 | 62.1 | 75.2 | 59.4 | 59.8 | 32.5 | 38.6 | 54.9 | - |
| 1.07B | ✓ | 60B | ✓ | 2 | Avg | 64 | SVD | - | - | - | 11.36 | 9.14 | 13.82 | 58.9 | 62.8 | 75.1 | 61.5 | 61.2 | 33.7 | 37.6 | 55.8 | - |
| 1.15B | ✓ | 60B | ✓ | 2 | Avg | 128 | SVD | - | - | - | 11.25 | 9.04 | 13.64 | 58.7 | 63.6 | 76.5 | 61.2 | 62.6 | 34.6 | 39.0 | 56.6 | - |
| 1.30B | ✓ | 60B | ✓ | 2 | Avg | 256 | SVD | - | - | - | 11.05 | 8.88 | 13.35 | 60.6 | 64.7 | 75.3 | 62.5 | 61.6 | 35.3 | 38.8 | 57.0 | - |
| 1.60B | ✓ | 60B | ✓ | 2 | Avg | 512 | SVD | - | - | - | 10.81 | 8.63 | 12.94 | 61.4 | 65.8 | 76.3 | 63.5 | 65.1 | 37.1 | 39.4 | 58.4 | - |
| 0.99B | ✓ | 60B | ✓ | 2 | Step | - | - | 15B | Agg (0.1) | ✓ | 11.71 | 9.56 | 14.46 | 54.0 | 61.7 | 75.1 | 58.9 | 58.6 | 31.9 | 37.6 | 54.0 | −0.9 |
| | | | | | | | | | | | 13.68 | 11.39 | 17.60 | 45.0 | 54.1 | 71.9 | 58.5 | 49.8 | 28.8 | 33.4 | 48.8 | - |
| 1.07B | ✓ | 60B | ✓ | 2 | Avg | 64 | SVD | 15B | Agg (0.1) | ✓ | 11.79 | 9.70 | 14.52 | 53.7 | 60.8 | 73.6 | 61.1 | 58.7 | 32.9 | 37.2 | 54.0 | −1.8 |
| | | | | | | | | | | | 19.45 | 16.46 | 26.10 | 30.7 | 37.9 | 66.5 | 55.3 | 42.2 | 25.3 | 27.6 | 40.8 | - |
| 1.15B | ✓ | 60B | ✓ | 2 | Avg | 128 | SVD | 15B | Agg (0.1) | ✓ | 11.66 | 9.59 | 14.32 | 53.3 | 62.1 | 74.9 | 60.0 | 59.9 | 33.4 | 38.8 | 54.6 | −2.0 |
| | | | | | | | | | | | 19.65 | 16.77 | 26.44 | 29.7 | 37.7 | 66.8 | 52.6 | 41.4 | 25.3 | 28.0 | 40.2 | - |
| 1.30B | ✓ | 60B | ✓ | 2 | Avg | 256 | SVD | 15B | Agg (0.1) | ✓ | 11.47 | 9.39 | 14.03 | 54.9 | 63.0 | 74.5 | 61.7 | 60.5 | 33.1 | 38.4 | 55.2 | −1.8 |
| | | | | | | | | | | | 19.67 | 16.82 | 26.40 | 29.7 | 38.3 | 66.4 | 53.1 | 43.8 | 24.7 | 27.6 | 40.5 | - |
| 1.60B | ✓ | 60B | ✓ | 2 | Avg | 512 | SVD | 15B | Agg (0.1) | ✓ | 11.20 | 9.14 | 13.58 | 57.2 | 64.1 | 75.2 | 61.7 | 62.1 | 34.6 | 38.2 | 56.2 | −2.2 |
| | | | | | | | | | | | 19.29 | 16.47 | 25.73 | 32.0 | 39.6 | 67.6 | 53.3 | 43.2 | 25.8 | 30.2 | 41.7 | - |
| 1.07B | ✓ | 60B | ✓ | 2 | Avg | 64 | SVD | 15B | Agg (0.3) | ✓ | 12.11 | 9.98 | 14.97 | 52.6 | 59.8 | 74.4 | 59.4 | 57.6 | 31.1 | 37.0 | 53.1 | −2.7 |
| | | | | | | | | | | | 16.09 | 13.54 | 21.19 | 35.4 | 42.8 | 69.8 | 52.8 | 45.8 | 25.8 | 31.0 | 43.3 | - |
| 1.15B | ✓ | 60B | ✓ | 2 | Avg | 128 | SVD | 15B | Agg (0.3) | ✓ | 11.96 | 9.87 | 14.76 | 52.3 | 60.5 | 74.2 | 59.1 | 58.9 | 33.0 | 37.2 | 53.6 | −3.0 |
| | | | | | | | | | | | 16.28 | 13.77 | 21.45 | 35.2 | 42.1 | 69.8 | 53.5 | 46.5 | 25.8 | 31.2 | 43.4 | - |
| 1.30B | ✓ | 60B | ✓ | 2 | Avg | 256 | SVD | 15B | Agg (0.3) | ✓ | 11.73 | 9.63 | 14.43 | 54.3 | 61.4 | 75.0 | 60.7 | 58.8 | 33.1 | 38.6 | 54.6 | −2.4 |
| | | | | | | | | | | | 16.41 | 13.89 | 21.68 | 35.6 | 42.3 | 69.0 | 52.7 | 46.8 | 26.4 | 29.8 | 43.2 | - |
| 1.60B | ✓ | 60B | ✓ | 2 | Avg | 512 | SVD | 15B | Agg (0.3) | ✓ | 11.47 | 9.36 | 13.93 | 56.2 | 62.7 | 75.4 | 60.9 | 60.4 | 34.0 | 37.0 | 55.2 | −3.2 |
| | | | | | | | | | | | 16.24 | 13.72 | 21.42 | 37.8 | 43.6 | 69.0 | 54.4 | 45.5 | 26.4 | 31.2 | 44.0 | - |
| 0.66B | ✓ | 60B | ✓ | 3 | Step | - | - | - | - | - | 12.27 | 9.90 | 15.24 | 55.6 | 58.1 | 73.1 | 60.2 | 58.8 | 30.2 | 37.2 | 53.3 | - |
| 0.74B | ✓ | 60B | ✓ | 3 | Avg | 64 | SVD | - | - | - | 12.13 | 9.80 | 14.95 | 55.5 | 58.3 | 73.5 | 60.1 | 58.0 | 31.1 | 36.8 | 53.3 | - |
| 0.82B | ✓ | 60B | ✓ | 3 | Avg | 128 | SVD | - | - | - | 11.83 | 9.53 | 14.51 | 56.7 | 60.2 | 74.2 | 59.8 | 59.1 | 33.0 | 35.4 | 54.1 | - |
| 0.97B | ✓ | 60B | ✓ | 3 | Avg | 256 | SVD | - | - | - | 11.43 | 9.17 | 13.87 | 59.3 | 62.6 | 74.7 | 61.2 | 61.6 | 32.9 | 40.2 | 56.1 | - |
| 1.27B | ✓ | 60B | ✓ | 3 | Avg | 512 | SVD | - | - | - | 11.01 | 8.80 | 13.25 | 61.5 | 64.9 | 76.2 | 62.0 | 64.3 | 35.6 | 39.2 | 57.7 | - |
| 0.66B | ✓ | 60B | ✓ | 3 | Step | - | - | 15B | Agg (0.1) | ✓ | 12.75 | 10.48 | 16.01 | 50.2 | 57.0 | 72.7 | 58.6 | 56.7 | 30.0 | 38.2 | 51.9 | −1.4 |
| | | | | | | | | | | | 13.81 | 11.47 | 17.80 | 48.4 | 53.0 | 72.4 | 55.6 | 51.6 | 27.2 | 35.2 | 49.0 | - |
| | | | | | | | | | | | 16.72 | 14.23 | 22.97 | 37.7 | 44.2 | 69.8 | 53.6 | 44.2 | 24.6 | 30.2 | 43.5 | - |
| 0.74B | ✓ | 60B | ✓ | 3 | Avg | 64 | SVD | 15B | Agg (0.1) | ✗ | 12.64 | 10.43 | 15.81 | 51.4 | 56.3 | 72.2 | 57.9 | 56.7 | 30.4 | 35.0 | 51.4 | −1.9 |
| | | | | | | | | | | | 19.90 | 16.88 | 26.26 | 30.4 | 39.3 | 66.3 | 54.1 | 41.2 | 24.8 | 29.2 | 40.8 | - |
| | | | | | | | | | | | 26.31 | 22.49 | 36.10 | 20.9 | 31.2 | 62.6 | 50.8 | 37.2 | 22.0 | 28.0 | 36.1 | - |
| 0.82B | ✓ | 60B | ✓ | 3 | Avg | 128 | SVD | 15B | Agg (0.1) | ✗ | 12.37 | 10.21 | 15.38 | 52.0 | 58.0 | 72.0 | 56.5 | 58.4 | 30.0 | 35.2 | 51.7 | −2.4 |
| | | | | | | | | | | | 20.07 | 17.09 | 26.47 | 30.9 | 40.5 | 66.3 | 55.4 | 40.8 | 24.4 | 29.6 | 41.1 | - |
| | | | | | | | | | | | 26.15 | 22.46 | 35.98 | 21.3 | 31.2 | 62.7 | 51.8 | 36.4 | 22.9 | 26.2 | 36.1 | - |
| 0.97B | ✓ | 60B | ✓ | 3 | Avg | 256 | SVD | 15B | Agg (0.1) | ✗ | 11.92 | 9.78 | 14.71 | 54.8 | 60.6 | 74.6 | 60.1 | 60.1 | 31.8 | 36.6 | 54.1 | −2.0 |
| | | | | | | | | | | | 19.29 | 16.49 | 25.51 | 35.2 | 42.5 | 65.8 | 55.6 | 41.5 | 25.6 | 29.4 | 42.2 | - |
| | | | | | | | | | | | 25.12 | 21.53 | 34.53 | 23.1 | 32.0 | 63.2 | 49.7 | 36.1 | 23.0 | 25.2 | 36.1 | - |
| 1.27B | ✓ | 60B | ✓ | 3 | Avg | 512 | SVD | 15B | Agg (0.1) | ✗ | 11.49 | 9.38 | 14.00 | 56.1 | 62.7 | 74.4 | 60.5 | 62.1 | 34.9 | 38.8 | 55.7 | −3.0 |
| | | | | | | | | | | | 18.52 | 15.79 | 24.34 | 36.7 | 44.9 | 67.2 | 55.3 | 43.8 | 26.0 | 30.4 | 43.5 | - |
| | | | | | | | | | | | 24.19 | 20.70 | 33.20 | 24.4 | 32.4 | 63.9 | 50.8 | 37.9 | 21.9 | 27.4 | 37.0 | - |
| 0.74B | ✓ | 60B | ✓ | 3 | Avg | 64 | SVD | 15B | Agg (0.3) | ✗ | 13.07 | 10.84 | 16.49 | 47.7 | 54.4 | 71.7 | 56.1 | 55.9 | 29.4 | 35.2 | 50.1 | −3.2 |
| | | | | | | | | | | | 16.68 | 14.08 | 21.86 | 35.4 | 42.4 | 68.2 | 53.8 | 44.6 | 26.3 | 29.4 | 42.9 | - |
| | | | | | | | | | | | 21.43 | 18.26 | 29.12 | 24.4 | 34.1 | 64.3 | 50.5 | 40.7 | 22.3 | 27.8 | 37.7 | - |
| 0.82B | ✓ | 60B | ✓ | 3 | Avg | 128 | SVD | 15B | Agg (0.3) | ✗ | 12.71 | 10.54 | 15.92 | 50.4 | 55.9 | 73.1 | 57.5 | 56.8 | 30.1 | 34.8 | 51.2 | −2.9 |
| | | | | | | | | | | | 16.90 | 14.32 | 22.18 | 37.6 | 43.5 | 67.6 | 54.5 | 45.0 | 25.3 | 29.0 | 43.2 | - |
| | | | | | | | | | | | 21.23 | 18.13 | 28.88 | 25.3 | 34.0 | 64.6 | 51.7 | 40.7 | 23.0 | 26.4 | 38.0 | - |
| 0.97B | ✓ | 60B | ✓ | 3 | Avg | 256 | SVD | 15B | Agg (0.3) | ✗ | 12.26 | 10.15 | 15.23 | 53.5 | 58.5 | 73.5 | 58.8 | 58.3 | 30.6 | 37.6 | 53.0 | −3.1 |
| | | | | | | | | | | | 16.56 | 14.09 | 21.68 | 42.6 | 45.1 | 68.2 | 57.7 | 45.7 | 25.9 | 28.8 | 44.8 | - |
| | | | | | | | | | | | 20.78 | 17.72 | 28.29 | 27.9 | 34.3 | 66.3 | 52.2 | 39.7 | 23.6 | 26.8 | 38.7 | - |
| 1.27B | ✓ | 60B | ✓ | 3 | Avg | 512 | SVD | 15B | Agg (0.3) | ✗ | 11.80 | 9.68 | 14.45 | 54.1 | 61.2 | 74.0 | 59.0 | 59.9 | 32.9 | 38.0 | 54.1 | −3.6 |
| | | | | | | | | | | | 16.02 | 13.53 | 20.86 | 43.5 | 47.5 | 68.3 | 56.2 | 47.1 | 27.0 | 30.4 | 45.7 | - |
| | | | | | | | | | | | 20.20 | 17.21 | 27.50 | 28.9 | 35.2 | 65.6 | 52.9 | 41.9 | 23.2 | 26.6 | 39.2 | - |

Table 15: Evaluation results of TinyLlama and Pythia models after early-exit training. Delta (Δ) represents the accuracy change in the original last layer following early-exit post-training. In case of Pythia, these changes should be compared in reference to the performance drops observed in 75B and 60B uptraining for the full-size model.

| Models | N-emb | Uptrain PT | $N_{tok}$ | KD | Looping Block | Init | LoRA Rank | Init | Early-Exit Train $N_{tok}$ | CE | KD | Perplexity↓ SlimP | RedP | PG19 | Few-shot Accuracy↑ LD | HS | PQ | WG | ARC-e | ARC-c | OB | Avg | Δ |
|---|---|---|---|---|---|---|---|---|---|---|---|---|---|---|---|---|---|---|---|---|---|---|---|
| | 0.97B | ✓ | - | ✗ | - | - | - | - | - | - | - | 12.26 | 9.37 | 11.94 | 43.3 | 42.2 | 66.8 | 53.4 | 44.7 | 23.2 | 29.2 | 43.3 | - |
| | 0.48B | ✓ | 60B | ✓ | 2 | Step | - | - | - | - | - | 10.51 | 9.01 | 11.60 | 44.2 | 43.1 | 68.2 | 52.4 | 44.7 | 25.3 | 32.2 | 44.3 | - |
| | 0.53B | ✓ | 60B | ✓ | 2 | Avg | 64 | SVD | - | - | - | 10.14 | 8.77 | 11.19 | 44.3 | 44.9 | 69.5 | 52.5 | 46.5 | 26.1 | 31.6 | 45.0 | - |
| | 0.58B | ✓ | 60B | ✓ | 2 | Avg | 128 | SVD | - | - | - | 10.07 | 8.68 | 11.07 | 45.9 | 45.1 | 69.4 | 50.5 | 46.8 | 25.4 | 31.6 | 45.0 | - |
| | 0.68B | ✓ | 60B | ✓ | 2 | Avg | 256 | SVD | - | - | - | 9.96 | 8.56 | 10.93 | 46.2 | 45.7 | 69.0 | 53.2 | 47.9 | 25.9 | 31.6 | 45.6 | - |
| | 0.86B | ✓ | 60B | ✓ | 2 | Avg | 512 | SVD | - | - | - | 9.85 | 8.44 | 10.76 | 47.4 | 46.3 | 69.7 | 52.8 | 47.5 | 26.3 | 31.4 | 45.9 | - |
| | 0.48B | ✓ | 60B | ✓ | 2 | Step | - | - | 15B | Agg(0.1) | ✓ | 10.55 | 9.16 | 11.68 | 45.0 | 43.7 | 68.9 | 53.4 | 44.8 | 25.3 | 32.2 | 44.8 | +0.5 |
| | | | | | | | | | | | | 12.28 | 10.62 | 13.83 | 38.2 | 39.4 | 65.8 | 52.3 | 41.5 | 24.7 | 30.6 | 41.8 | - |
| | 0.53B | ✓ | 60B | ✓ | 2 | Avg | 64 | SVD | 15B | Agg(0.1) | ✓ | 10.34 | 9.08 | 11.50 | 43.4 | 44.8 | 69.5 | 53.4 | 46.9 | 25.6 | 32.0 | 45.1 | +0.1 |
| | | | | | | | | | | | | 21.23 | 18.63 | 24.85 | 16.8 | 29.0 | 57.6 | 48.9 | 33.2 | 23.1 | 27.0 | 33.7 | - |
| | 0.58B | ✓ | 60B | ✓ | 2 | Avg | 128 | SVD | 15B | Agg(0.1) | ✓ | 10.25 | 8.97 | 11.36 | 45.2 | 45.5 | 68.8 | 54.0 | 46.5 | 25.0 | 31.6 | 45.2 | +0.2 |
| | | | | | | | | | | | | 21.30 | 18.56 | 24.75 | 18.5 | 28.9 | 58.4 | 48.0 | 34.1 | 21.8 | 27.4 | 33.9 | - |
| TinyLlama | 0.68B | ✓ | 60B | ✓ | 2 | Avg | 256 | SVD | 15B | Agg(0.1) | ✓ | 10.13 | 8.84 | 11.23 | 45.2 | 45.9 | 69.6 | 53.6 | 46.9 | 25.9 | 32.0 | 45.6 | +0.0 |
| | | | | | | | | | | | | 20.95 | 18.16 | 24.22 | 20.1 | 28.8 | 57.8 | 48.9 | 33.8 | 22.5 | 25.0 | 33.9 | - |
| | 0.86B | ✓ | 60B | ✓ | 2 | Avg | 512 | SVD | 15B | Agg(0.1) | ✓ | 10.02 | 8.74 | 11.04 | 46.6 | 46.5 | 68.6 | 54.5 | 47.9 | 26.3 | 32.2 | 46.1 | +0.2 |
| | | | | | | | | | | | | 20.38 | 17.70 | 23.57 | 19.9 | 28.8 | 58.2 | 49.0 | 34.7 | 22.8 | 25.8 | 34.2 | - |
| | 0.53B | ✓ | 60B | ✓ | 2 | Avg | 64 | SVD | 15B | Agg(0.3) | ✓ | 10.61 | 9.36 | 11.87 | 42.1 | 43.7 | 68.6 | 54.1 | 46.1 | 26.0 | 31.2 | 44.6 | −0.4 |
| | | | | | | | | | | | | 16.83 | 14.88 | 19.77 | 22.0 | 30.3 | 60.7 | 50.7 | 36.9 | 24.1 | 27.8 | 36.1 | - |
| | 0.58B | ✓ | 60B | ✓ | 2 | Avg | 128 | SVD | 15B | Agg(0.3) | ✓ | 10.50 | 9.22 | 11.71 | 44.2 | 44.2 | 69.2 | 53.0 | 46.0 | 25.5 | 31.2 | 44.8 | −0.2 |
| | | | | | | | | | | | | 17.10 | 15.03 | 19.99 | 23.5 | 30.1 | 60.8 | 51.3 | 36.5 | 23.8 | 26.4 | 36.0 | - |
| | 0.68B | ✓ | 60B | ✓ | 2 | Avg | 256 | SVD | 15B | Agg(0.3) | ✓ | 10.34 | 9.07 | 11.51 | 44.0 | 45.0 | 68.4 | 53.0 | 45.8 | 26.0 | 31.2 | 44.8 | −0.8 |
| | | | | | | | | | | | | 17.06 | 14.92 | 19.82 | 24.2 | 30.4 | 59.9 | 51.7 | 36.2 | 23.9 | 27.2 | 36.2 | - |
| | 0.86B | ✓ | 60B | ✓ | 2 | Avg | 512 | SVD | 15B | Agg(0.3) | ✓ | 10.21 | 8.94 | 11.28 | 45.1 | 45.8 | 69.3 | 54.5 | 46.7 | 25.9 | 33.4 | 45.8 | −0.1 |
| | | | | | | | | | | | | 16.76 | 14.68 | 19.43 | 24.4 | 30.0 | 61.1 | 51.9 | 37.1 | 22.9 | 28.2 | 36.5 | - |
| | 0.81B | ✓ | 60B | ✗ | - | - | - | - | - | - | - | 12.83 | 9.76 | 13.57 | 53.0 | 50.2 | 71.1 | 54.8 | 51.9 | 27.7 | 31.6 | 48.6 | - |
| | 0.81B | ✓ | 75B | ✗ | - | - | - | - | - | - | - | 12.86 | 9.86 | 13.74 | 54.8 | 50.3 | 70.5 | 55.3 | 52.2 | 28.8 | 33.0 | 49.3 | +0.7 |
| | 0.40B | ✓ | 60B | ✓ | 2 | Step | - | - | - | - | - | 14.59 | 11.13 | 15.79 | 47.8 | 43.8 | 69.3 | 52.0 | 48.1 | 25.4 | 30.4 | 45.2 | - |
| | 0.44B | ✓ | 60B | ✓ | 2 | Avg | 64 | SVD | - | - | - | 14.24 | 10.89 | 15.52 | 50.0 | 44.5 | 68.9 | 54.1 | 48.0 | 26.5 | 31.2 | 46.2 | - |
| | 0.48B | ✓ | 60B | ✓ | 2 | Avg | 128 | SVD | - | - | - | 14.10 | 10.79 | 15.27 | 50.1 | 45.5 | 69.0 | 52.6 | 48.3 | 25.8 | 32.0 | 46.2 | - |
| | 0.55B | ✓ | 60B | ✓ | 2 | Avg | 256 | SVD | - | - | - | 13.91 | 10.61 | 14.91 | 50.5 | 45.6 | 68.7 | 51.2 | 48.4 | 25.7 | 32.8 | 46.1 | - |
| | 0.70B | ✓ | 60B | ✓ | 2 | Avg | 512 | SVD | - | - | - | 13.59 | 10.38 | 14.43 | 52.0 | 47.0 | 69.6 | 53.4 | 48.9 | 26.9 | 31.2 | 47.0 | - |
| | 0.40B | ✓ | 60B | ✓ | 2 | Step | - | - | 15B | Agg(0.1) | ✓ | 14.72 | 11.38 | 16.31 | 47.0 | 44.2 | 69.2 | 53.4 | 48.6 | 24.7 | 30.4 | 45.4 | +0.2 |
| | | | | | | | | | | | | 18.61 | 14.11 | 20.96 | 38.4 | 38.1 | 67.0 | 53.7 | 43.3 | 24.4 | 29.0 | 42.0 | - |
| | 0.44B | ✓ | 60B | ✓ | 2 | Avg | 64 | SVD | 15B | Agg(0.1) | ✓ | 14.49 | 11.22 | 16.12 | 49.1 | 43.9 | 69.8 | 53.8 | 48.6 | 26.1 | 31.2 | 46.1 | −0.1 |
| | | | | | | | | | | | | 24.43 | 18.19 | 27.89 | 26.7 | 31.6 | 61.6 | 50.8 | 38.2 | 22.9 | 27.6 | 37.1 | - |
| | 0.48B | ✓ | 60B | ✓ | 2 | Avg | 128 | SVD | 15B | Agg(0.1) | ✓ | 14.35 | 11.17 | 15.93 | 50.1 | 44.7 | 69.0 | 52.1 | 49.9 | 25.3 | 32.6 | 46.2 | +0.0 |
| | | | | | | | | | | | | 24.33 | 18.09 | 27.96 | 28.2 | 32.3 | 61.1 | 53.0 | 38.8 | 23.7 | 27.4 | 37.8 | - |
| Pythia | 0.55B | ✓ | 60B | ✓ | 2 | Avg | 256 | SVD | 15B | Agg(0.1) | ✓ | 14.14 | 10.96 | 15.54 | 50.8 | 45.5 | 68.2 | 53.9 | 48.8 | 25.3 | 32.8 | 46.5 | +0.4 |
| | | | | | | | | | | | | 24.18 | 17.87 | 27.48 | 28.1 | 32.3 | 61.9 | 54.1 | 38.1 | 22.9 | 28.6 | 38.0 | - |
| | 0.70B | ✓ | 60B | ✓ | 2 | Avg | 512 | SVD | 15B | Agg(0.1) | ✓ | 13.81 | 10.72 | 15.11 | 52.4 | 47.0 | 69.3 | 52.7 | 50.1 | 26.9 | 32.0 | 47.2 | +0.2 |
| | | | | | | | | | | | | 23.50 | 17.49 | 26.72 | 29.5 | 32.8 | 63.2 | 52.3 | 38.8 | 22.8 | 27.8 | 38.2 | - |
| | 0.44B | ✓ | 60B | ✓ | 2 | Avg | 64 | SVD | 15B | Agg(0.3) | ✓ | 14.87 | 11.53 | 16.61 | 47.0 | 43.1 | 68.7 | 53.0 | 47.4 | 25.7 | 31.0 | 45.1 | −0.9 |
| | | | | | | | | | | | | 20.62 | 15.60 | 23.57 | 32.6 | 33.6 | 63.4 | 51.2 | 40.7 | 23.3 | 28.0 | 39.0 | - |
| | 0.48B | ✓ | 60B | ✓ | 2 | Avg | 128 | SVD | 15B | Agg(0.3) | ✓ | 14.69 | 11.46 | 16.36 | 48.9 | 43.8 | 68.4 | 53.0 | 49.1 | 26.2 | 31.6 | 45.9 | −0.3 |
| | | | | | | | | | | | | 20.60 | 15.56 | 23.63 | 33.2 | 33.6 | 62.7 | 51.1 | 41.3 | 23.6 | 27.8 | 39.0 | - |
| | 0.55B | ✓ | 60B | ✓ | 2 | Avg | 256 | SVD | 15B | Agg(0.3) | ✓ | 14.44 | 11.20 | 15.94 | 50.0 | 44.7 | 69.2 | 52.3 | 48.1 | 25.4 | 32.2 | 46.0 | −0.1 |
| | | | | | | | | | | | | 20.61 | 15.48 | 23.45 | 33.3 | 34.2 | 63.4 | 52.2 | 40.8 | 23.0 | 28.8 | 39.4 | - |
| | 0.70B | ✓ | 60B | ✓ | 2 | Avg | 512 | SVD | 15B | Agg(0.3) | ✓ | 14.08 | 10.94 | 15.44 | 51.1 | 46.4 | 68.7 | 52.2 | 50.0 | 26.9 | 31.6 | 46.7 | −0.3 |
| | | | | | | | | | | | | 20.20 | 15.25 | 22.98 | 34.6 | 34.1 | 63.5 | 53.0 | 41.5 | 23.6 | 27.6 | 39.7 | - |

## N    HYPOTHETICAL GENERATION SPEEDUP

**Measuring the average generation time per token**    First, we measured the generation time with various model configurations using dummy weights and inputs. We measured the elapsed time for each components, such as embedding matrices, Transformer blocks, and the classifier head. We measured decoding speed using FlashDecoding (Dao et al., 2022), a technique that has recently become standard in serving LLMs. Especially, we calculated the time per token by dividing the total time by the decoding length. Default prefix and decoding lengths are set to 512 and 2048, but we also used shorter context lengths, like 64 and 256 to simulate scenarios where parameter memory sizes become limiting. Using a single A100 40GiB GPU, we measured generation times by increasing batch sizes until an out-of-memory error occurred or memory usuage reached the predefined limit.

In Table 16, generation time was measured up to the maximum batch size that a single A100 GPU could accommodate before encountering out-of-memory errors, with prefix and decoding lengths set to 512 and 2048, respectively. Meanwhile, Table 18 presents generation times measured in a more memory-constrained deployment scenario, where the prefix and decoding lengths were reduced to 64 and 256, and the memory limit was set to 16GB. As anticipated, under severe memory constraints, the reduced parameter memory footprint of Recursive Transformers enabled substantially larger batch sizes. This observation indicates that Recursive Transformers, even without continuous batching techniques, can achieve higher throughput than vanilla Transformers due to their inherent memory efficiency.

When comparing the speed of the three models, Gemma 2B was the fastest, followed by TinyLlama 1.1B and then Pythia 1B. This order is the exact inverse of their non-embedding parameter sizes. This speed difference is attributed to the Grouped-Query and Multi-Query attention mechanisms (Ainslie et al., 2023). The main decoding bottleneck in Transformers is memory access to the key-value cache. Hence, Gemma that effectively reduces the key-value cache size through the MQA mechanism, achieves fastest speeds. Despite using GQA, TinyLlama 1.1B has a similar speed to Gemma 2B due to its shallow and deep architecture (22 layers compared to Gemma's 18 layers). This deeper architecture likely offsets the speed gains from the attention mechanism.

**Comparison of hypothetical generation throughput**    We conducted early-exiting simulations using language modeling datasets (SlimPajama, RedPajama, and PG19), assuming our models generated the tokens. For each dataset's test set, we employed an oracle-exiting algorithm to determine the earliest possible exit point for each token. We used 20K samples to obtain their exit trajectories. By combining this trajectory data with previously measured per-token processing time (considering only Transformer block computations), we estimated the hypothetical throughput across various settings and datasets. The results are detailed in Tables 17 and 19.

Our analysis reveals that Recursive Transformers achieve a 2-3$\times$ throughput gain over vanilla Transformers. Relaxed models also demonstrate significant speedup despite unoptimized LoRA computations. Currently, we merge multiple LoRAs into a single, larger LoRA to enable parallel computation of samples across different looping iterations. However, this incurs extra overhead due to redundant computations. Therefore, we observed reduced throughput gains in memory-constrained scenarios (shorter context lengths and lower memory limits). This degradation stems from the increased proportion of LoRA computation time relative to overall processing time. Because attention computation has quadratic complexity with respec to lengths, it becomes less expensive at shorter context lengths, while the complexity of LoRA computation remains constant. This highlights the impact of unoptimized LoRA computations, leading to substantial throughput reduction. However, these findings suggest that relaxed models will yield even greater performance and throughput improvements in scenarios with longer contexts where attention computation dominates.

**Approximation errors in our hypothetical throughput**    Since our throughput estimations are based on theoretical estimation, they may introduce certain approximation errors as follows:

- As our models are not fine-tuned for any downstream task, we simulated the exit trajectories of language modeling datasets by assuming they were generated by our models. While this approach is expected to closely approximate actual generation, empirical validation is necessary to confirm its accuracy.

- Throughput gains should be measured using realistic (confidence-based) early-exiting algorithms, rather than relying on the oracle-exiting algorithm. While early-exiting algorithms can introduce performance degradation due to inherent errors in confidence estimation, they also incur additional computational costs for estimating prediction confidence, necessitating further efficiency improvements.

- Our analysis solely focused on speed improvements within Transformer blocks. However, upon early exiting, the exited tokens require separate processing through the embedding layer or the classifier head for subsequent sequence generation. This necessitates non-exited tokens to wait for others, potentially reducing efficiency as the embedding layer computation may not fully utilize the maximum batch size.

- Early-exiting architectures require computing key-value caches in remaining layers for already exited tokens to prevent performance degradation (Bae et al., 2023). While this adds negligible overhead in memory-bound scenarios, it inevitably increases overhead in compute-bound scenarios where the maximum batch size is fully utilized. Our throughput estimation, however, excludes the computation time for these key-value caches in later loops (though we did account for their memory size). Incorporating these computations into a more realistic analysis of early-exiting generation is a direction for future work.

Table 16: Measurements of generation time across three models using a single A100 40GB GPU. We measured time per token for both a batch size of 1 and the maximum batch size achievable by each model. The prefix length was set to 512 tokens, and the decoded output length to 2048 tokens. We then averaged the total elapsed time by the output length of 2048. Dummy input and dummy tensors were used for measurement. Both Gemma, employing multi-query attention, and TinyLlama, utilizing grouped-query attention, demonstrated fast generation speeds and large maximum batch sizes relative to their model sizes. TinyLlama's deep and narrow architecture allowed for a significantly large maximum batch size, although its generation speed was slower due to the increased number of layers.

| Models | Model Architecture | | | | | N-emb | Recursive | | Batch | Time (ms) per token | | | |
|---|---|---|---|---|---|---|---|---|---|---|---|---|---|
| | $N_L$ | $d_{model}$ | $N_{head}$ | $N_{KV}$ | Vocab | | Block | Rank | | Total | Emb | Transformer | Head |
| Gemma 2B | 18 | 2048 | 8 | 1 | 256K | 1.98B | - | - | 1
43 | 22.994
0.657 | 0.087
0.002 | 21.344
0.616 | 0.803
0.023 |
| | 18 | 2048 | 8 | 1 | 256K | 0.99B | 2 | - | 1
43 | 13.918
0.336 | 0.088
0.002 | 11.059
0.265 | 0.827
0.023 |
| | 18 | 2048 | 8 | 1 | 256K | 1.07B | 2 | 64 | 1
41 | 15.858
0.398 | 0.080
0.002 | 13.096
0.323 | 0.825
0.024 |
| | 18 | 2048 | 8 | 1 | 256K | 1.15B | 2 | 128 | 1
41 | 15.708
0.398 | 0.080
0.002 | 12.969
0.324 | 0.822
0.024 |
| | 18 | 2048 | 8 | 1 | 256K | 1.30B | 2 | 256 | 1
39 | 15.456
0.450 | 0.083
0.002 | 12.721
0.372 | 0.818
0.025 |
| | 18 | 2048 | 8 | 1 | 256K | 1.60B | 2 | 512 | 1
39 | 15.489
0.499 | 0.078
0.002 | 12.775
0.422 | 0.817
0.025 |
| | 18 | 2048 | 8 | 1 | 256K | 0.66B | 3 | - | 1
43 | 10.546
0.263 | 0.081
0.002 | 7.394
0.182 | 0.827
0.023 |
| | 18 | 2048 | 8 | 1 | 256K | 0.74B | 3 | 64 | 1
43 | 11.871
0.306 | 0.080
0.002 | 8.724
0.182 | 0.827
0.023 |
| | 18 | 2048 | 8 | 1 | 256K | 0.82B | 3 | 128 | 1
43 | 11.768
0.294 | 0.080
0.002 | 8.649
0.221 | 0.825
0.023 |
| | 18 | 2048 | 8 | 1 | 256K | 0.97B | 3 | 256 | 1
41 | 12.018
0.311 | 0.081
0.002 | 8.848
0.226 | 0.823
0.024 |
| | 18 | 2048 | 8 | 1 | 256K | 1.27B | 3 | 512 | 1
39 | 12.087
0.325 | 0.082
0.002 | 8.932
0.237 | 0.822
0.025 |
| TinyLlama 1.1B | 22 | 2048 | 32 | 4 | 32K | 0.97B | - | - | 1
329 | 22.016
0.819 | 0.082
0.000 | 21.010
0.815 | 0.188
0.001 |
| | 22 | 2048 | 32 | 4 | 32K | 0.48B | 2 | - | 1
233 | 12.657
0.446 | 0.077
0.000 | 10.370
0.413 | 0.209
0.001 |
| | 22 | 2048 | 32 | 4 | 32K | 0.53B | 2 | 64 | 1
211 | 15.243
0.454 | 0.079
0.000 | 12.908
0.421 | 0.211
0.002 |
| | 22 | 2048 | 32 | 4 | 32K | 0.58B | 2 | 128 | 1
209 | 15.456
0.454 | 0.082
0.000 | 13.118
0.421 | 0.213
0.002 |
| | 22 | 2048 | 32 | 4 | 32K | 0.68B | 2 | 256 | 1
209 | 15.223
0.457 | 0.081
0.000 | 12.908
0.423 | 0.208
0.002 |
| | 22 | 2048 | 32 | 4 | 32K | 0.86B | 2 | 512 | 1
209 | 15.383
0.461 | 0.080
0.000 | 13.062
0.428 | 0.211
0.002 |
| Pythia 1B | 16 | 2048 | 8 | 8 | 50K | 0.81B | - | - | 1
53 | 13.280
1.227 | 0.080
0.002 | 12.286
1.206 | 0.235
0.005 |
| | 16 | 2048 | 8 | 8 | 50K | 0.40B | 2 | - | 1
61 | 8.423
0.856 | 0.081
0.001 | 6.378
0.606 | 0.262
0.005 |
| | 16 | 2048 | 8 | 8 | 50K | 0.44B | 2 | 64 | 1
63 | 10.554
0.875 | 0.082
0.001 | 8.519
0.626 | 0.260
0.005 |
| | 16 | 2048 | 8 | 8 | 50K | 0.48B | 2 | 128 | 1
59 | 10.167
0.892 | 0.076
0.001 | 8.196
0.642 | 0.256
0.005 |
| | 16 | 2048 | 8 | 8 | 50K | 0.55B | 2 | 256 | 1
59 | 10.410
0.913 | 0.079
0.001 | 8.402
0.662 | 0.258
0.005 |
| | 16 | 2048 | 8 | 8 | 50K | 0.70B | 2 | 512 | 1
53 | 12.609
0.956 | 0.091
0.002 | 10.311
0.702 | 0.267
0.006 |

Table 17: Hypothetical generation speedup of Recursive Transformers across three models. We utilized the measurements of tokens per second calculated in Table 16. We only considered the time spent within Transformer blocks, simulating generation on the SlimPajama, RedPajama, and PG19 test sets. We used a vanilla transformer model, both with and without continuous sequence-wise batching, as our baselines. Our Recursive models further enhance throughput by applying continuous depth-wise batching, leveraging looping and early-exiting techniques. The throughput improvements over the vanilla Transformer and sequence-wise batching are denoted as $\Delta_V$ and $\Delta_{Seq}$, respectively. To aid in understanding the speedup, we also provide the performance of intermediate layers and the maximum batch size.

| Models | N-emb | | Uptrain | | | Looping | | LoRA | | Early-Exit Train | | | Batching | | Few-shot Accuracy | | | | Throughput↑ | | | | |
|---|---|---|---|---|---|---|---|---|---|---|---|---|---|---|---|---|---|---|---|---|---|---|---|
| | | PT | $N_{tok}$ | KD | Block | Init | Rank | Init | $N_{tok}$ | CE | KD | Type | Exit | Last | Mid 1 | Mid 2 | Batch | SlimP | RedP | PG19 | $\Delta_V$ | $\Delta_{Seq}$ |
| | 1.99B | ✓ | 75B | ✗ | - | - | - | - | - | - | - | - | ✗ | 57.3 | - | - | 43 | 655 | 1228 | 1357 | ×1.00 | ×0.71 |
| | 1.99B | ✓ | 75B | ✗ | - | - | - | - | - | - | - | CSB | ✗ | 57.3 | - | - | 43 | 1622 | 1604 | 1357 | ×1.41 | ×1.00 |
| | 0.99B | ✓ | 60B | ✓ | 2 | Step | - | - | 15B | Agg(0.1) | ✓ | CDB | ✓ | 54.0 | 48.8 | - | 43 | 3159 | 3050 | 2421 | ×2.66 | ×1.88 |
| | 1.07B | ✓ | 60B | ✓ | 2 | Avg | 64 | SVD | 15B | Agg(0.1) | ✓ | CDB | ✓ | 54.0 | 40.8 | - | 41 | 2357 | 2255 | 1858 | ×2.00 | ×1.41 |
| | 1.15B | ✓ | 60B | ✓ | 2 | Avg | 128 | SVD | 15B | Agg(0.1) | ✓ | CDB | ✓ | 54.6 | 40.2 | - | 41 | 2355 | 2250 | 1844 | ×1.99 | ×1.41 |
| | 1.30B | ✓ | 60B | ✓ | 2 | Avg | 256 | SVD | 15B | Agg(0.1) | ✓ | CDB | ✓ | 55.2 | 40.5 | - | 39 | 2047 | 1976 | 1740 | ×1.78 | ×1.26 |
| | 1.60B | ✓ | 60B | ✓ | 2 | Avg | 512 | SVD | 15B | Agg(0.1) | ✓ | CDB | ✓ | 56.2 | 41.7 | - | 39 | 1806 | 1754 | 1598 | ×1.59 | ×1.13 |
| | 1.07B | ✓ | 60B | ✓ | 2 | Avg | 64 | SVD | 15B | Agg(0.3) | ✓ | CDB | ✓ | 53.1 | 43.3 | - | 41 | 2454 | 2357 | 1929 | ×2.08 | ×1.47 |
| | 1.15B | ✓ | 60B | ✓ | 2 | Avg | 128 | SVD | 15B | Agg(0.3) | ✓ | CDB | ✓ | 53.6 | 43.4 | - | 41 | 2445 | 2346 | 1926 | ×2.07 | ×1.47 |
| | 1.30B | ✓ | 60B | ✓ | 2 | Avg | 256 | SVD | 15B | Agg(0.3) | ✓ | CDB | ✓ | 54.6 | 43.2 | - | 39 | 2123 | 2056 | 1804 | ×1.85 | ×1.31 |
| Gemma | 1.60B | ✓ | 60B | ✓ | 2 | Avg | 512 | SVD | 15B | Agg(0.3) | ✓ | CDB | ✓ | 55.2 | 44.0 | - | 39 | 1870 | 1819 | 1655 | ×1.65 | ×1.17 |
| | 0.66B | ✓ | 60B | ✓ | 3 | Step | - | - | 15B | Agg(0.1) | ✓ | CDB | ✓ | 51.9 | 49.0 | 43.5 | 43 | 3120 | 3041 | 2729 | ×2.74 | ×1.94 |
| | 0.74B | ✓ | 60B | ✓ | 3 | Avg | 64 | SVD | 15B | Agg(0.1) | ✗ | CDB | ✓ | 51.4 | 40.8 | 36.1 | 43 | 2334 | 2274 | 2059 | ×2.06 | ×1.45 |
| | 0.82B | ✓ | 60B | ✓ | 3 | Avg | 128 | SVD | 15B | Agg(0.1) | ✗ | CDB | ✓ | 51.7 | 41.1 | 36.1 | 43 | 2290 | 2230 | 2007 | ×2.02 | ×1.42 |
| | 0.97B | ✓ | 60B | ✓ | 3 | Avg | 256 | SVD | 15B | Agg(0.1) | ✗ | CDB | ✓ | 54.1 | 42.2 | 36.1 | 41 | 2281 | 2219 | 1984 | ×2.00 | ×1.41 |
| | 1.27B | ✓ | 60B | ✓ | 3 | Avg | 512 | SVD | 15B | Agg(0.1) | ✗ | CDB | ✓ | 55.7 | 43.5 | 37.0 | 39 | 2181 | 2122 | 1900 | ×1.91 | ×1.35 |
| | 0.74B | ✓ | 60B | ✓ | 3 | Avg | 64 | SVD | 15B | Agg(0.3) | ✗ | CDB | ✓ | 50.1 | 42.9 | 37.7 | 43 | 2427 | 2372 | 2143 | ×2.14 | ×1.51 |
| | 0.82B | ✓ | 60B | ✓ | 3 | Avg | 128 | SVD | 15B | Agg(0.3) | ✗ | CDB | ✓ | 51.2 | 43.2 | 38.0 | 43 | 2376 | 2321 | 2084 | ×2.09 | ×1.48 |
| | 0.97B | ✓ | 60B | ✓ | 3 | Avg | 256 | SVD | 15B | Agg(0.3) | ✗ | CDB | ✓ | 53.0 | 44.8 | 38.7 | 41 | 2359 | 2300 | 2039 | ×2.07 | ×1.46 |
| | 1.27B | ✓ | 60B | ✓ | 3 | Avg | 512 | SVD | 15B | Agg(0.3) | ✗ | CDB | ✓ | 54.1 | 45.7 | 39.2 | 39 | 2251 | 2191 | 1975 | ×1.98 | ×1.40 |
| | 0.97B | ✓ | - | - | - | - | - | - | - | - | - | - | ✗ | 43.3 | - | - | 329 | 1205 | 1220 | 1194 | ×1.00 | ×0.99 |
| | 0.97B | ✓ | - | - | - | - | - | - | - | - | - | CSB | ✗ | 43.3 | - | - | 329 | 1227 | 1225 | 1194 | ×1.01 | ×1.00 |
| | 0.48B | ✓ | 60B | ✓ | 2 | Step | - | - | 15B | Agg(0.1) | ✓ | CDB | ✓ | 44.8 | 41.8 | - | 233 | 2038 | 2023 | 1933 | ×1.66 | ×1.64 |
| | 0.53B | ✓ | 60B | ✓ | 2 | Avg | 64 | SVD | 15B | Agg(0.1) | ✓ | CDB | ✓ | 45.1 | 33.7 | - | 211 | 1733 | 1719 | 1617 | ×1.40 | ×1.39 |
| | 0.58B | ✓ | 60B | ✓ | 2 | Avg | 128 | SVD | 15B | Agg(0.1) | ✓ | CDB | ✓ | 45.2 | 33.9 | - | 209 | 1733 | 1717 | 1609 | ×1.40 | ×1.39 |
| TinyLlama | 0.68B | ✓ | 60B | ✓ | 2 | Avg | 256 | SVD | 15B | Agg(0.1) | ✓ | CDB | ✓ | 45.6 | 33.9 | - | 209 | 1728 | 1714 | 1606 | ×1.39 | ×1.38 |
| | 0.86B | ✓ | 60B | ✓ | 2 | Avg | 512 | SVD | 15B | Agg(0.1) | ✓ | CDB | ✓ | 46.1 | 34.2 | - | 209 | 1716 | 1702 | 1581 | ×1.38 | ×1.37 |
| | 0.53B | ✓ | 60B | ✓ | 2 | Avg | 64 | SVD | 15B | Agg(0.3) | ✓ | CDB | ✓ | 44.6 | 36.1 | - | 211 | 1810 | 1796 | 1688 | ×1.46 | ×1.45 |
| | 0.58B | ✓ | 60B | ✓ | 2 | Avg | 128 | SVD | 15B | Agg(0.3) | ✓ | CDB | ✓ | 44.8 | 36.0 | - | 209 | 1802 | 1787 | 1668 | ×1.45 | ×1.44 |
| | 0.68B | ✓ | 60B | ✓ | 2 | Avg | 256 | SVD | 15B | Agg(0.3) | ✓ | CDB | ✓ | 44.8 | 36.2 | - | 209 | 1793 | 1779 | 1668 | ×1.45 | ×1.44 |
| | 0.86B | ✓ | 60B | ✓ | 2 | Avg | 512 | SVD | 15B | Agg(0.3) | ✓ | CDB | ✓ | 45.8 | 36.5 | - | 209 | 1778 | 1763 | 1637 | ×1.43 | ×1.42 |
| | 0.81B | ✓ | 75B | ✗ | - | - | - | - | - | - | - | - | ✗ | 49.3 | - | - | 53 | 702 | 785 | 822 | ×1.00 | ×0.93 |
| | 0.81B | ✓ | 75B | ✗ | - | - | - | - | - | - | - | CSB | ✗ | 49.3 | - | - | 53 | 829 | 827 | 822 | ×1.07 | ×1.00 |
| | 0.40B | ✓ | 60B | ✓ | 2 | Step | - | - | 15B | Agg(0.1) | ✓ | CDB | ✓ | 45.4 | 42.0 | - | 61 | 1339 | 1333 | 1281 | ×1.71 | ×1.60 |
| | 0.44B | ✓ | 60B | ✓ | 2 | Avg | 64 | SVD | 15B | Agg(0.1) | ✓ | CDB | ✓ | 46.1 | 37.1 | - | 63 | 1205 | 1203 | 1140 | ×1.54 | ×1.43 |
| Pythia | 0.48B | ✓ | 60B | ✓ | 2 | Avg | 128 | SVD | 15B | Agg(0.1) | ✓ | CDB | ✓ | 46.2 | 37.8 | - | 59 | 1156 | 1180 | 1108 | ×1.49 | ×1.39 |
| | 0.55B | ✓ | 60B | ✓ | 2 | Avg | 256 | SVD | 15B | Agg(0.1) | ✓ | CDB | ✓ | 46.5 | 38.0 | - | 59 | 1138 | 1139 | 1071 | ×1.45 | ×1.35 |
| | 0.70B | ✓ | 60B | ✓ | 2 | Avg | 512 | SVD | 15B | Agg(0.1) | ✓ | CDB | ✓ | 47.2 | 38.2 | - | 53 | 1051 | 1077 | 1021 | ×1.36 | ×1.27 |
| | 0.44B | ✓ | 60B | ✓ | 2 | Avg | 64 | SVD | 15B | Agg(0.3) | ✓ | CDB | ✓ | 45.1 | 39.0 | - | 63 | 1254 | 1252 | 1190 | ×1.60 | ×1.49 |
| | 0.48B | ✓ | 60B | ✓ | 2 | Avg | 128 | SVD | 15B | Agg(0.3) | ✓ | CDB | ✓ | 45.9 | 39.0 | - | 59 | 1200 | 1226 | 1153 | ×1.55 | ×1.45 |
| | 0.55B | ✓ | 60B | ✓ | 2 | Avg | 256 | SVD | 15B | Agg(0.3) | ✓ | CDB | ✓ | 46.0 | 39.4 | - | 59 | 1180 | 1180 | 1112 | ×1.50 | ×1.40 |
| | 0.70B | ✓ | 60B | ✓ | 2 | Avg | 512 | SVD | 15B | Agg(0.3) | ✓ | CDB | ✓ | 46.7 | 39.7 | - | 53 | 1088 | 1114 | 1058 | ×1.41 | ×1.32 |

Table 18: Generation time measurements of Gemma models on a single A100 GPU with 16GB memory constraint. We measured time per token for both a batch size of 1 and the maximum batch size achievable by each model. The prefix length was set to 64 tokens, and the decoded output length to 256 tokens. We then averaged the total elapsed time by the output length of 256. Dummy input and dummy tensors were used for measurement.

| Models | Model Architecture | | | | | N-emb | Recursive | | Batch | Time (ms) per token | | | |
|---|---|---|---|---|---|---|---|---|---|---|---|---|---|
| | $N_L$ | $d_{model}$ | $N_{head}$ | $N_{KV}$ | Vocab | | Block | Rank | | Total | Emb | Transformer | Head |
| Gemma 2B | 18 | 2048 | 8 | 1 | 256K | 1.98B | - | - | 1 | 22.577 | 0.084 | 20.937 | 0.801 |
| | | | | | | | | | 111 | 0.207 | 0.001 | 0.188 | 0.010 |
| | 18 | 2048 | 8 | 1 | 256K | 0.99B | 2 | - | 1 | 13.576 | 0.079 | 10.819 | 0.815 |
| | | | | | | | | | 123 | 0.118 | 0.001 | 0.091 | 0.009 |
| | 18 | 2048 | 8 | 1 | 256K | 1.07B | 2 | 64 | 1 | 15.372 | 0.080 | 12.675 | 0.813 |
| | | | | | | | | | 117 | 0.140 | 0.001 | 0.112 | 0.009 |
| | 18 | 2048 | 8 | 1 | 256K | 1.15B | 2 | 128 | 1 | 15.631 | 0.082 | 12.899 | 0.816 |
| | | | | | | | | | 115 | 0.141 | 0.001 | 0.113 | 0.010 |
| | 18 | 2048 | 8 | 1 | 256K | 1.30B | 2 | 256 | 1 | 15.317 | 0.079 | 12.639 | 0.811 |
| | | | | | | | | | 111 | 0.143 | 0.001 | 0.115 | 0.010 |
| | 18 | 2048 | 8 | 1 | 256K | 1.60B | 2 | 512 | 1 | 15.379 | 0.080 | 12.692 | 0.807 |
| | | | | | | | | | 103 | 0.158 | 0.001 | 0.127 | 0.011 |
| | 18 | 2048 | 8 | 1 | 256K | 0.66B | 3 | - | 1 | 10.528 | 0.080 | 7.411 | 0.817 |
| | | | | | | | | | 131 | 0.087 | 0.001 | 0.058 | 0.010 |
| | 18 | 2048 | 8 | 1 | 256K | 0.74B | 3 | 64 | 1 | 11.957 | 0.081 | 8.855 | 0.815 |
| | | | | | | | | | 123 | 0.105 | 0.001 | 0.075 | 0.009 |
| | 18 | 2048 | 8 | 1 | 256K | 0.82B | 3 | 128 | 1 | 11.898 | 0.080 | 8.787 | 0.816 |
| | | | | | | | | | 121 | 0.103 | 0.001 | 0.074 | 0.009 |
| | 18 | 2048 | 8 | 1 | 256K | 0.97B | 3 | 256 | 1 | 11.734 | 0.079 | 8.654 | 0.813 |
| | | | | | | | | | 117 | 0.106 | 0.001 | 0.076 | 0.009 |
| | 18 | 2048 | 8 | 1 | 256K | 1.27B | 3 | 512 | 1 | 11.986 | 0.080 | 8.856 | 0.809 |
| | | | | | | | | | 107 | 0.125 | 0.001 | 0.090 | 0.010 |
| TinyLlama 1.1B | 22 | 2048 | 32 | 4 | 32K | 0.97B | - | - | 1 | 23.898 | 0.080 | 22.909 | 0.189 |
| | | | | | | | | | 1049 | 0.131 | 0.000 | 0.129 | 0.001 |
| | 22 | 2048 | 32 | 4 | 32K | 0.48B | 2 | - | 1 | 14.129 | 0.080 | 11.846 | 0.202 |
| | | | | | | | | | 1121 | 0.070 | 0.000 | 0.064 | 0.001 |
| | 22 | 2048 | 32 | 4 | 32K | 0.53B | 2 | 64 | 1 | 14.897 | 0.080 | 12.627 | 0.202 |
| | | | | | | | | | 1105 | 0.073 | 0.000 | 0.068 | 0.001 |
| | 22 | 2048 | 32 | 4 | 32K | 0.58B | 2 | 128 | 1 | 15.090 | 0.081 | 12.778 | 0.205 |
| | | | | | | | | | 1089 | 0.074 | 0.000 | 0.069 | 0.001 |
| | 22 | 2048 | 32 | 4 | 32K | 0.68B | 2 | 256 | 1 | 14.962 | 0.081 | 12.659 | 0.201 |
| | | | | | | | | | 1065 | 0.076 | 0.000 | 0.071 | 0.001 |
| | 22 | 2048 | 32 | 4 | 32K | 0.86B | 2 | 512 | 1 | 15.284 | 0.083 | 12.950 | 0.206 |
| | | | | | | | | | 1017 | 0.080 | 0.000 | 0.075 | 0.001 |
| Pythia 1B | 16 | 2048 | 8 | 8 | 50K | 0.81B | - | - | 1 | 13.341 | 0.081 | 12.326 | 0.239 |
| | | | | | | | | | 229 | 0.176 | 0.000 | 0.171 | 0.002 |
| | 16 | 2048 | 8 | 8 | 50K | 0.40B | 2 | - | 1 | 8.336 | 0.079 | 6.303 | 0.261 |
| | | | | | | | | | 241 | 0.121 | 0.000 | 0.086 | 0.002 |
| | 16 | 2048 | 8 | 8 | 50K | 0.44B | 2 | 64 | 1 | 10.408 | 0.081 | 8.353 | 0.262 |
| | | | | | | | | | 233 | 0.133 | 0.000 | 0.097 | 0.002 |
| | 16 | 2048 | 8 | 8 | 50K | 0.48B | 2 | 128 | 1 | 10.426 | 0.082 | 8.378 | 0.259 |
| | | | | | | | | | 221 | 0.137 | 0.000 | 0.101 | 0.002 |
| | 16 | 2048 | 8 | 8 | 50K | 0.55B | 2 | 256 | 1 | 10.509 | 0.080 | 8.471 | 0.256 |
| | | | | | | | | | 205 | 0.151 | 0.000 | 0.115 | 0.002 |
| | 16 | 2048 | 8 | 8 | 50K | 0.70B | 2 | 512 | 1 | 11.254 | 0.080 | 9.241 | 0.257 |
| | | | | | | | | | 165 | 0.177 | 0.001 | 0.139 | 0.002 |

Table 19: Hypothetical generation speedup of Recursive Transformers across three models. We utilized the measurements of tokens per second calculated in Table 18. We only considered the time spent within Transformer blocks, simulating generation on the SlimPajama, RedPajama, and PG19 test sets. We used a vanilla transformer model, both with and without continuous sequence-wise batching, as our baselines. Our Recursive models further enhance throughput by applying continuous depth-wise batching, leveraging looping and early-exiting techniques. The throughput improvements over the vanilla Transformer and sequence-wise batching are denoted as $\Delta_V$ and $\Delta_{Seq}$, respectively. To aid in understanding the speedup, we also provide the performance of intermediate layers and the maximum batch size.

| | | Uptrain | | | Looping | | LoRA | | Early-Exit Train | | | Batching | | Few-shot Accuracy | | | | Throughput↑ | | | | |
|---|---|---|---|---|---|---|---|---|---|---|---|---|---|---|---|---|---|---|---|---|---|---|
| Models | N-emb | PT | $N_{tok}$ | KD | Block | Init | Rank | Init | $N_{tok}$ | CE | KD | Type | Exit | Last | Mid 1 | Mid 2 | Batch | SlimP | RedP | PG19 | $\Delta_V$ | $\Delta_{Seq}$ |
| | 1.99B | ✓ | 75B | ✗ | - | - | - | - | - | - | - | - | ✗ | 57.3 | - | - | 111 | 1740 | 3059 | 4796 | ×1.00 | ×0.63 |
| | 1.99B | ✓ | 75B | ✗ | - | - | - | - | - | - | - | CSB | ✗ | 57.3 | - | - | 111 | 5287 | 5060 | 4796 | ×1.58 | ×1.00 |
| | 0.99B | ✓ | 60B | ✓ | 2 | Step | - | - | 15B | Agg(0.1) | ✓ | CDB | ✓ | 54.0 | 48.8 | - | 43 | 3159 | 3050 | 2421 | ×2.50 | ×1.59 |
| | 1.07B | ✓ | 60B | ✓ | 2 | Avg | 64 | SVD | 15B | Agg(0.1) | ✓ | CDB | ✓ | 54.0 | 40.8 | - | 41 | 2357 | 2255 | 1858 | ×1.87 | ×1.19 |
| | 1.15B | ✓ | 60B | ✓ | 2 | Avg | 128 | SVD | 15B | Agg(0.1) | ✓ | CDB | ✓ | 54.6 | 40.2 | - | 41 | 2355 | 2250 | 1844 | ×1.87 | ×1.19 |
| | 1.30B | ✓ | 60B | ✓ | 2 | Avg | 256 | SVD | 15B | Agg(0.1) | ✓ | CDB | ✓ | 55.2 | 40.5 | - | 39 | 2047 | 1976 | 1740 | ×1.86 | ×1.18 |
| | 1.60B | ✓ | 60B | ✓ | 2 | Avg | 512 | SVD | 15B | Agg(0.1) | ✓ | CDB | ✓ | 56.2 | 41.7 | - | 39 | 1806 | 1754 | 1598 | ×1.73 | ×1.10 |
| | 1.07B | ✓ | 60B | ✓ | 2 | Avg | 64 | SVD | 15B | Agg(0.3) | ✓ | CDB | ✓ | 53.1 | 43.3 | - | 41 | 2454 | 2357 | 1929 | ×1.95 | ×1.24 |
| | 1.15B | ✓ | 60B | ✓ | 2 | Avg | 128 | SVD | 15B | Agg(0.3) | ✓ | CDB | ✓ | 53.6 | 43.4 | - | 41 | 2445 | 2346 | 1926 | ×1.95 | ×1.24 |
| | 1.30B | ✓ | 60B | ✓ | 2 | Avg | 256 | SVD | 15B | Agg(0.3) | ✓ | CDB | ✓ | 54.6 | 43.2 | - | 39 | 2123 | 2056 | 1804 | ×1.93 | ×1.22 |
| Gemma | 1.60B | ✓ | 60B | ✓ | 2 | Avg | 512 | SVD | 15B | Agg(0.3) | ✓ | CDB | ✓ | 55.2 | 44.0 | - | 39 | 1870 | 1819 | 1655 | ×1.79 | ×1.14 |
| | 0.66B | ✓ | 60B | ✓ | 3 | Step | - | - | 15B | Agg(0.1) | ✓ | CDB | ✓ | 51.9 | 49.0 | 43.5 | 43 | 3120 | 3041 | 2729 | ×2.62 | ×1.66 |
| | 0.74B | ✓ | 60B | ✓ | 3 | Avg | 64 | SVD | 15B | Agg(0.1) | ✗ | CDB | ✓ | 51.4 | 40.8 | 36.1 | 43 | 2334 | 2274 | 2059 | ×1.87 | ×1.19 |
| | 0.82B | ✓ | 60B | ✓ | 3 | Avg | 128 | SVD | 15B | Agg(0.1) | ✗ | CDB | ✓ | 51.7 | 41.1 | 36.1 | 43 | 2290 | 2230 | 2007 | ×1.90 | ×1.20 |
| | 0.97B | ✓ | 60B | ✓ | 3 | Avg | 256 | SVD | 15B | Agg(0.1) | ✗ | CDB | ✓ | 54.1 | 42.2 | 36.1 | 41 | 2281 | 2219 | 1984 | ×1.86 | ×1.18 |
| | 1.27B | ✓ | 60B | ✓ | 3 | Avg | 512 | SVD | 15B | Agg(0.1) | ✗ | CDB | ✓ | 55.7 | 43.5 | 37.0 | 39 | 2181 | 2122 | 1900 | ×1.62 | ×1.03 |
| | 0.74B | ✓ | 60B | ✓ | 3 | Avg | 64 | SVD | 15B | Agg(0.3) | ✗ | CDB | ✓ | 50.1 | 42.9 | 37.7 | 43 | 2427 | 2372 | 2143 | ×1.94 | ×1.23 |
| | 0.82B | ✓ | 60B | ✓ | 3 | Avg | 128 | SVD | 15B | Agg(0.3) | ✗ | CDB | ✓ | 51.2 | 43.2 | 38.0 | 43 | 2376 | 2321 | 2084 | ×1.97 | ×1.25 |
| | 0.97B | ✓ | 60B | ✓ | 3 | Avg | 256 | SVD | 15B | Agg(0.3) | ✗ | CDB | ✓ | 53.0 | 44.8 | 38.7 | 41 | 2359 | 2300 | 2039 | ×1.92 | ×1.22 |
| | 1.27B | ✓ | 60B | ✓ | 3 | Avg | 512 | SVD | 15B | Agg(0.3) | ✗ | CDB | ✓ | 54.1 | 45.7 | 39.2 | 39 | 2251 | 2191 | 1975 | ×1.67 | ×1.06 |
| | 0.97B | ✓ | - | - | - | - | - | - | - | - | - | - | ✗ | 43.3 | - | - | 1049 | 6856 | 7481 | 4090 | ×1.00 | ×0.96 |
| | 0.97B | ✓ | - | - | - | - | - | - | - | - | - | CSB | ✗ | 43.3 | - | - | 1049 | 7709 | 7481 | 4090 | ×1.05 | ×1.00 |
| | 0.48B | ✓ | 60B | ✓ | 2 | Step | - | - | 15B | Agg(0.1) | ✓ | CDB | ✓ | 44.8 | 41.8 | - | 233 | 2038 | 2023 | 1933 | ×1.70 | ×1.62 |
| | 0.53B | ✓ | 60B | ✓ | 2 | Avg | 64 | SVD | 15B | Agg(0.1) | ✓ | CDB | ✓ | 45.1 | 33.7 | - | 211 | 1733 | 1719 | 1617 | ×1.38 | ×1.32 |
| | 0.58B | ✓ | 60B | ✓ | 2 | Avg | 128 | SVD | 15B | Agg(0.1) | ✓ | CDB | ✓ | 45.2 | 33.9 | - | 209 | 1733 | 1717 | 1609 | ×1.36 | ×1.30 |
| TinyLlama | 0.68B | ✓ | 60B | ✓ | 2 | Avg | 256 | SVD | 15B | Agg(0.1) | ✓ | CDB | ✓ | 45.6 | 33.9 | - | 209 | 1728 | 1714 | 1606 | ×1.34 | ×1.28 |
| | 0.86B | ✓ | 60B | ✓ | 2 | Avg | 512 | SVD | 15B | Agg(0.1) | ✓ | CDB | ✓ | 46.1 | 34.2 | - | 209 | 1716 | 1702 | 1581 | ×1.28 | ×1.23 |
| | 0.53B | ✓ | 60B | ✓ | 2 | Avg | 64 | SVD | 15B | Agg(0.3) | ✓ | CDB | ✓ | 44.6 | 36.1 | - | 211 | 1810 | 1796 | 1688 | ×1.45 | ×1.38 |
| | 0.58B | ✓ | 60B | ✓ | 2 | Avg | 128 | SVD | 15B | Agg(0.3) | ✓ | CDB | ✓ | 44.8 | 36.0 | - | 209 | 1802 | 1787 | 1668 | ×1.41 | ×1.35 |
| | 0.68B | ✓ | 60B | ✓ | 2 | Avg | 256 | SVD | 15B | Agg(0.3) | ✓ | CDB | ✓ | 44.8 | 36.2 | - | 209 | 1793 | 1779 | 1668 | ×1.39 | ×1.33 |
| | 0.86B | ✓ | 60B | ✓ | 2 | Avg | 512 | SVD | 15B | Agg(0.3) | ✓ | CDB | ✓ | 45.8 | 36.5 | - | 209 | 1778 | 1763 | 1637 | ×1.33 | ×1.27 |
| | 0.81B | ✓ | 75B | ✗ | - | - | - | - | - | - | - | - | ✗ | 49.3 | - | - | 229 | 4273 | 5346 | 5149 | ×1.00 | ×0.89 |
| | 0.81B | ✓ | 75B | ✗ | - | - | - | - | - | - | - | CSB | ✗ | 49.3 | - | - | 229 | 5813 | 5724 | 5149 | ×1.13 | ×1.00 |
| | 0.40B | ✓ | 60B | ✓ | 2 | Step | - | - | 15B | Agg(0.1) | ✓ | CDB | ✓ | 45.4 | 42.0 | - | 61 | 1339 | 1333 | 1281 | ×1.77 | ×1.57 |
| | 0.44B | ✓ | 60B | ✓ | 2 | Avg | 64 | SVD | 15B | Agg(0.1) | ✓ | CDB | ✓ | 46.1 | 37.1 | - | 63 | 1205 | 1203 | 1140 | ×1.44 | ×1.28 |
| Pythia | 0.48B | ✓ | 60B | ✓ | 2 | Avg | 128 | SVD | 15B | Agg(0.1) | ✓ | CDB | ✓ | 46.2 | 37.8 | - | 59 | 1156 | 1180 | 1108 | ×1.32 | ×1.17 |
| | 0.55B | ✓ | 60B | ✓ | 2 | Avg | 256 | SVD | 15B | Agg(0.1) | ✓ | CDB | ✓ | 46.5 | 38.0 | - | 59 | 1138 | 1139 | 1071 | ×1.22 | ×1.08 |
| | 0.70B | ✓ | 60B | ✓ | 2 | Avg | 512 | SVD | 15B | Agg(0.1) | ✓ | CDB | ✓ | 47.2 | 38.2 | - | 53 | 1051 | 1077 | 1021 | ×0.98 | ×0.87 |
| | 0.44B | ✓ | 60B | ✓ | 2 | Avg | 64 | SVD | 15B | Agg(0.3) | ✓ | CDB | ✓ | 45.1 | 39.0 | - | 63 | 1254 | 1252 | 1190 | ×1.50 | ×1.33 |
| | 0.48B | ✓ | 60B | ✓ | 2 | Avg | 128 | SVD | 15B | Agg(0.3) | ✓ | CDB | ✓ | 45.9 | 39.0 | - | 59 | 1200 | 1226 | 1153 | ×1.37 | ×1.22 |
| | 0.55B | ✓ | 60B | ✓ | 2 | Avg | 256 | SVD | 15B | Agg(0.3) | ✓ | CDB | ✓ | 46.0 | 39.4 | - | 59 | 1180 | 1180 | 1112 | ×1.27 | ×1.12 |
| | 0.70B | ✓ | 60B | ✓ | 2 | Avg | 512 | SVD | 15B | Agg(0.3) | ✓ | CDB | ✓ | 46.7 | 39.7 | - | 53 | 1088 | 1114 | 1058 | ×1.02 | ×0.90 |

