# OpenReview forum: "Relaxed Recursive Transformers: Effective Parameter Sharing with Layer-wise LoRA"
_ICLR.cc/2025/Conference — ICLR 2025 Poster_

### Official Review · Reviewer_LZ2e · 2024-10-31

**Soundness:** 2
**Presentation:** 3
**Contribution:** 3
**Rating:** 6
**Confidence:** 5

**Summary:**

This paper revisits the concept of "layer tying" as a form of parameter sharing in Transformers for developing smaller models. These models utilize a single block of unique layers repeated in a loop derived from pre-trained models to reduce model size while maintaining high performance. The paper presents Relaxed Recursive Transformers incorporating depth-wise LoRA modules to enhance these recursive models. These modules add flexibility to the layer-tying process but maintain the model's compactness. Furthermore, the paper proposes a new inference technique, Continuous Depth-wise Batching, which leverages the recursive nature of these models combined with early exiting strategies to increase throughput by 2-3 times.

**Strengths:**

1. The introduction of Recursive Transformers with a single repeated layer block offers a novel approach to reducing model size and complexity without substantial loss in performance.

2. The incorporation of Relaxed Recursive Transformers with LoRA modules successfully improves flexibility and performance.

3. The proposed Continuous Depth-wise Batching method, which takes advantage of the recursive model structure and early exiting, increases inference throughput.

**Weaknesses:**

1. The approach of uptraining a full-size pre-trained model is questionable, especially given that the goal is to achieve comparable performance to the original model. In practical applications, deployment considerations often prioritize model size rather than training data size or consumption. Using a smaller, lower-quality dataset for these experiments may unfairly favor smaller (1B) models, potentially skewing the results and making the claimed improvements less persuasive.

2. Furthermore, this uptraining step could be prohibitive for settings where on-device retraining is required. The reliance on extensive uptraining, which consumes 60 billion tokens, presents challenges for practical deployment, particularly when computational resources are constrained.

3. The current methodology appears to train reduced models from scratch, using the same number of tokens, while recursive transformers are initialized with pre-trained layers. This discrepancy in training procedures introduces a potential bias in favor of the recursive transformers.

4. The paper lacks a clear rationale for determining the number of initial layers to be recursively repeated and the frequency of such recursions. Furthermore, an automated approach for selecting the optimal recursion size would greatly increase the novelty and technical contribution of the proposed method.

5. All experiments are conducted with models around the 1B parameter mark, yet hardware advancements make models larger than 1B feasible for many applications. Testing the proposed approach on larger models is essential to validate its scalability and effectiveness beyond the 1B parameter range, ensuring the method’s relevance for state-of-the-art large models.

6. The selection of layers for initializing looped layers, as indicated in Figure 5, appears to depend heavily on the specific model used. A clear explanation of how to automate this layer selection process would substantially improve usability and consistency across different models. Without this, the method’s applicability may be limited by model-specific manual tuning.

7. The impact of using different base models (other than a 2B model) for recursive transformer initialization has not been evaluated. Testing a range of base model sizes would provide insight into how the choice of initial model affects recursive initialization outcomes, offering a more thorough assessment of the method’s robustness.

8. The experiments focus exclusively on few-shot tasks, but in real-world applications, 1B models are often fine-tuned for specific tasks. Testing the proposed method on downstream QA or classification tasks with full training on task-specific data would provide a more realistic evaluation of its performance and applicability across different use cases.

**Questions:**

Please refer to the weaknesses section.

---

> ### Author Response · Authors · 2024-11-21
>
> We thank the reviewer for their thoughtful comments. We appreciate that the reviewer finds our methods, particularly the relaxation technique and Continuous Depth-wise Batching, to be novel and effective. We provide responses to the reviewer’s specific questions and concerns (quoted) below.
>
> ---
>
> > **[W1] Uptraining a full-size pretrained model is questionable.** Using a smaller, lower-quality dataset may unfairly favor smaller (recursive) models, potentially skewing the results.
>
> **[A1]** Thank you for pointing out this potentially confusing aspect. Our decision was motivated not by a desire to make an advantageous scenario for the recursive model, but rather to **mitigate the potential impact of distribution shift between the uptraining and pretraining datasets**. The only uptraining dataset we use is the publicly available SlimPajama dataset (which is the same pretraining dataset for TinyLlama, but not for Pythia or Gemma). As shown in Table 4 of Appendix H, finetuning the original Gemma and Pythia models on SlimPajama leads to slight performance *drops* on certain few-shot tasks, indicating that **using SlimPajama lowers the upper bound of achievable performance for Recursive Gemma and Pythia models.**
>
> In order to disentangle the effects of using (a) *an inferior uptraining dataset* vs. (b) *a recursive architecture*, we therefore believe that using the full-sized Gemma and Pythia models uptrained on SlimPajama is the fairest comparison point. On the other hand, for TinyLlama, where the pretraining data of the original LLM matches the uptraining dataset, it is appropriate to use the original LLM performance as the primary baseline.
>
> In order to openly share the full details of the experimental setup, we have also included the detailed results of all full-size models both before and after uptraining in Table 4 of Appendix H. It is also useful to note that the overall effects of uptraining the base model are fairly small: average Gemma performance drops by 3.1 absolute percentage points, Pythia drops by only 0.2 absolute percentage points, and TinyLlama improves by 4.1 absolute percentage points. We have further clarified this point in the revised paper.
>
> ---
>
> > **[W2]** The reliance on **extensive uptraining, which consumes 60 billion tokens, presents challenges for practical deployment** (e.g., on device training).
>
> **[A2]** While we agree that on-device deployment is an interesting use case for Recursive Transformers, in this work we assumed that the training itself is not performed on the edge device. We leverage uptraining, often referred to as **“continual pretraining”**, which is a common method of further “pretraining” models for specific purposes beyond their original training.
>
> Moreover, to demonstrate the efficiency of our approach, **we show that the conversion to a recursive model requires only a relatively small training cost.** Notably, we evaluate models uptrained on **15 billion or 60 billion tokens which amounts to only 0.5% to 2% of the approximately 3 trillion tokens typically used for pretraining models of comparable scale**.
>
> We emphasize that this uptraining, which is likely performed on modern accelerators, shouldn’t limit the deployment to on-device use cases in any way (it is more likely to help with that deployment due to the reduced parameter count). We hypothesize that on-device training recipes that work for pretrained LMs are likely to work out-of-the-box for uptrained Recursive Transformers. We leave that to future work.

---

> ### Author Response · Authors · 2024-11-21
>
> > **[W3] Reduced-size models are trained from scratch**, while recursive transformers are initialized with pretrained layers.
>
> **[A3]** To understand the performance of our Recursive Transformer, we established two non-recursive baselines: full-size models and reduced-size models. **The reduced size model performance is meant to serve as a lower bound** which we can use to better judge the efficacy of (1) *unique looping and parameter sharing techniques* that are made possible by our approach and (2) *leveraging pretrained layers*. Though initialized from scratch, note that the reduced-size model is still trained via a distillation loss from the full-size, pre-trained model, akin to our recursive models.
>
> **To further ablate the effect of each of these two components, we conducted additional experiments using the Pythia 410M model** presented in the following table (we used the Pile dataset--pretraining dataset of Pythia models--for uptraining.) Intuitively, we observed **significant performance gains by leveraging pretrained layers, with further improvement achieved through recursion**. We believe this additional experiment provides valuable insight into the performance contributions of the two approaches proposed for constructing Recursive Transformers. We have included this in the revised paper.
>
> ---
> | Model | N–emb | Pretrain | Recursion | Init | Uptrain | Loss | LBD | HS | PQ | WG | ARC-e | ARC-c | OBQA | Avg |
> |---|------|---|---|---|---|---|---|---|---|---|---|---|---|---|
> | Pythia |  300M | $\checkmark$ | - | - | - | - | 44.96 | 40.97 | 66.97 | 53.28 | 44.40 | 25.51 | 30.20 | 43.76 |
> |=====|=====|======|========|======|=====|=====|====|====|====|====|====|====|====|====|
> | Pythia |  150M | $\times$ | - | - | 15B | 2.6468 | 31.48 | 29.53 | 61.37 | 52.49 | 39.14 | 22.44 | 27.00 | 37.63 |
> | Pythia |  150M | $\times$ | 2 | Random | 15B | 2.6252 | 31.55 | 29.94 | 62.30 | 50.88 | 40.28 | 23.98 | 28.20 | 38.02 |
> |=====|=====|======|========|======|=====|=====|====|====|====|====|====|====|====|====|
> | Pythia |  150M | $\checkmark$ |  - | - | 15B | 2.4405  | 40.48 | 34.19 | 63.42 | 50.99 | 41.84 | 23.12 | 28.40 | 40.35 |
> | Pythia |  150M | $\checkmark$ |  2 | Stepwise | 15B | 2.4006  | 43.41 | 35.59 | 64.58 | 53.04 | 41.58 | 23.81 | 28.80 | 41.54 |
>
> ---
>
> > **[W4]** Lack of clear **rationale for determining the number of initial layers to be repeated and the frequency of such recursions**. An **automated approach for selecting the optimal recursion size** would increase the technical contribution.
>
> **[A4]** Thank you for raising this important point. In our paper, **we carefully adjusted the number of looped layers and recursion steps to ensure the overall model depth remained the same with the original LLMs.** For instance, with Gemma 2B (which has 18 layers), we explored configurations like 9 layers with 2 recursions or 6 layers with 3 recursions. There are two key factors behind this choice: (1) it enables a **clearer and fairer comparison with the full-size baseline by maintaining equivalent model depths;** and (2) this setup allows us to **effectively leverage the benefits of layer-wise LoRA initialization in Relaxed Recursive Transformers.** By preserving the total depth, we could demonstrate a smoother transition between vanilla and recursive models through adjustments to the LoRA rank.
>
> While we agree that defining an optimal recursion size is an interesting question, **what is “optimal” ultimately depends on the user’s specific deployment scenario.** Our analysis reveals an inverse relationship between the number of shared blocks and the achievable performance for a given training budget. Therefore, **selecting the appropriate recursion depth requires careful consideration of the desired performance level and the available training resources.**
>
> ---
>
> > **[W5] Testing the proposed approach on larger models** is essential to validate its scalability and effectiveness beyond the 1B parameter range.
>
> **[A5]** Further exploration into the performance and speed benefits attainable in recent larger models presents a compelling avenue for future research. We anticipate that preserving more sophisticated capabilities in these larger models **may necessitate a greater number of uptraining steps**. The findings from TinyLlama suggest that **utilizing the same dataset used for pretraining will be crucial for facilitating performance restoration**. Despite these challenges, **we are confident that our observations regarding initialization and relaxation will generalize broadly** across diverse architectures and model scales, as evidenced by consistent performance trends across three distinct architectures and a wide range of model sizes.

---

> ### Author Response · Authors · 2024-11-21
>
> > **[W6]  The selection of layers for initializing looped layers appears to depend heavily on the specific model used.** A clear explanation of **how to automate this layer selection process** would substantially improve usability and consistency across different models.
>
> **[A6] We respectfully disagree with the assertion that the initialization method for looped layers is highly dependent on the specific model used.** In fact, **Figure 5 demonstrates that the “Stepwise” method**, which anchors the initial and final layers while selecting intermediate layers at a certain step interval, **consistently performs best across all the models we examined**. This includes models of varying sizes and architectures, as well as different recursion frequencies **for the recursive conversion** (please refer to Table 5 in Appendix I for detailed results).
>
> Furthermore, **Figure 6 illustrates that the “Average” initialization method consistently yields the best performance across all models for the relaxed recursive conversion** (for detailed results, please refer to Table 7 in Appendix J). Therefore, we believe that **our discovery of a robust and broadly applicable initialization strategy represents a significant and valuable contribution.** This finding offers a practical advantage to other researchers, potentially saving them from having to undertake the same extensive exploration process that we conducted.
>
> ---
>
> > **[W7]** The impact of using different base models (other than a 2B model) for recursive transformer initialization has not been evaluated.
>
> **[A7]** As we discussed above, **we observed a consistent superiority in initialization strategies** (Stepwise for recursive conversion, and Average for relaxed recursive conversion) **across both 1B and 2B base model scales**. We conducted **additional experiments on two smaller model sizes (Pythia 410M and 160M)** to evaluate recursive initialization techniques.
>
> ---
>
> | Model | N-emb | Recursion | Init | Uptrain | Loss | LBD | HS | PQ | WG | ARC-e | ARC-c | OBQA | Avg |
> |---|---|---|---|---|---|---|---|---|---|---|---|---|---|
> | Pythia | 300M | - | - | - | - | 44.96 | 40.97 | 66.97 | 53.28 | 44.40 | 25.51 | 30.20 | 43.76 |
> | Pythia | 150M | 2 | Stepwise | 15B |  2.4006  | 43.41 | 35.59 | 64.58 | 53.04 | 41.58 | 23.81 | 28.80 | 41.54 |
> | Pythia | 150M | 2 | Lower | 15B | 2.4393 | 42.98 | 34.32 | 63.93 | 52.41 | 42.34 | 24.15 | 25.00 | 40.73 |
> | Pythia | 150M | 2 | Average | 15B | 2.4471 | 39.84 | 34.17 | 64.31 | 52.25 | 41.04 | 24.66 | 26.60 | 40.41 |
> |======|=====|========|=======|=======|=====|====|====|====|====|====|====|====|====|
> | Pythia | 85M | - | - | - | - | 13.53 | 30.67 | 58.22 | 48.62 | 36.62 | 25.00 | 28.60 | 34.47 |
> | Pythia | 43M | 2 | Stepwise| 15B | 2.7684 | 21.02 | 29.28 | 60.01 | 48.93 | 37.92 | 23.98 | 28.00  | 35.59 |
> | Pythia | 43M | 2 | Lower | 15B | 2.7846  | 21.46 | 29.61 | 59.90 | 50.67| 38.52 | 22.95 | 28.00 | 35.87 |
> | Pythia | 43M | 2 | Average | 15B | 2.7800 | 22.36 | 29.07 | 60.17 | 49.96 | 37.24 | 23.29 | 26.60 | 35.53 |
>
> ---
>
> Our supplementary experiments, conducted on models with smaller sizes and different layer numbers (12 layers for Pythia 160M and 24 layers for Pythia 410M), **further validate the superior performance of the Stepwise method for looped layer initialization** (in light of the inherent randomness in few-shot accuracy, a comparison based on the **loss value** would provide a more stable measure of performance.) **These findings reinforce the robustness of our key observations regarding initialization methods**, complementing our original extensive experiments. We have incorporated these results into the revised version of the paper.
>
> ---
>
> > **[W8] Testing the proposed method on downstream QA or classification tasks** with full training on task-specific data would provide a more realistic evaluation of its performance.
>
> **[A8]** In this work, we focused specifically on the language modeling task and evaluating few-shot performance. Our goal was to provide a comprehensive exploration of various approaches within this domain, including initialization and training recipes for Recursive Transformers, as well as the integration of an early-exiting framework to demonstrate potential throughput gains. Evaluating our findings on a broader range of downstream tasks remains a promising avenue for future work.

---

> ### Author Response · Authors · 2024-11-25
> **Gentle Reminder to Reviewer LZ2e**
>
> Dear Reviewer LZ2e,
>
> Thank you once again for the time and effort you’ve dedicated to reviewing our paper.
>
> As the discussion period is coming to a close, we kindly remind you that there are two days remaining for any further comments or questions. We hope that our responses so far have clarified misleading points and addressed your concerns.
>
> If there are any remaining points or feedback you’d like us to consider, we’d be very grateful for the chance to discuss them before the deadline.
> Thank you again for your valuable insights and support throughout this process.
>
> Warm regards,
> Authors

---

> ### Author Response · Authors · 2024-11-28
> **Kind Reminder to Reviewer LZ2e**
>
> Dear Reviewer LZ2e,
>
> Thank you again for your time and effort in reviewing our paper. We greatly appreciate your valuable feedback and suggestions.
>
> We would like to gently remind you that the discussion period is coming to a close.
>
> In our rebuttal, we addressed your concerns by:
> - **Clarifying the rationale for full-size model baselines,**
> - **Explaining that training is assumed not to be performed on the edge device,**
> - **Clarifying the reduced-size model baseline, and Showing the effects of individual components in Recursive Transformers,**
> - **Elaborating on the choice of recursion numbers, and Explaining that optimal depths ultimately depends on the user’s deployment scenario,**
> - **Discussing scalability on larger models,**
> - **Explaining the consistent trend of our initialization method, while being not highly dependent on specific settings,**
> - **Conducting additional experiments on two different model sizes,**
> - **Discussing the applicability to downstream tasks.**
>
> If you have any remaining concerns, please do not hesitate to share them with us. We are more than willing to address them promptly.
>
> Thank you very much for your consideration.
>
> Best regards, Authors

---

> > ### Comment · Reviewer_LZ2e · 2024-11-28
> >
> > Thank you for your detailed reply, which addressed most of my concerns. Above all, I have decided to raise my score. I appreciate the effort put into the rebuttal and the improvements made in response to the feedback.

---

> ### Author Response · Authors · 2024-11-29
>
> We are delighted to hear that most of your concerns have been addressed. We appreciate your response and the updated score.

---

### Official Review · Reviewer_uD6H · 2024-11-02

**Soundness:** 3
**Presentation:** 4
**Contribution:** 3
**Rating:** 8
**Confidence:** 5

**Summary:**

This paper introduces a novel framework for parameter sharing in LLMs, namely relaxed recursive transformers. The method revist the way of layer tying and further convert LLMs into recursive transformers by sharing parameters across layers. Moreover, the performance is further improved by relaxing the layer tying constraint. The method is demonstrated to be well performed on various LLMs, it also get significant throughput gains by paring with early exiting, namely continuous depth-wise batching.

**Strengths:**

1. The paper is well motivated on a hot topic in the field of LLMs.
2. The paper is well organized and the insight of the method is clearly stated.
3. The experimental results are convincing. I appreciate the way the authors present the takeaways for relaxed recursive transformers.

**Weaknesses:**

1, It lacks evaluation on larger scales, which makes the scalability of the method remain questionalbe. For instance, I expect results on LLMs with 7B, 70B or larger scales.

**Questions:**

1. I wonder the performance of relaxed recursive transformers on larger scales.
2. How does the authors compare the proposed parameter sharing methods with other model deployment optimization methods, such as quantization, sparcification，pruning or distillation?

---

> ### Author Response · Authors · 2024-11-21
>
> We thank the reviewer for their careful review and thoughtful comments. We appreciate that the reviewer finds our work to be well motivated, well organized, and convincing experimentally. We provide responses to the reviewer’s specific questions and concerns (quoted) below.
>
> ---
>
> > **[W1] Lacks evaluation on larger scales**, raising concerns about the scalability of the approach.
>
> **[A1]** We acknowledge that exploring larger models is a valuable direction for future research. While preserving their capabilities **might require larger uptraining steps, leveraging the same dataset used for pretraining could significantly facilitate performance recovery**, as demonstrated in TinyLlama. Moreover, throughout this work, we evaluated our methods across three distinct architectures and a wide range of base model scales. **These comprehensive evaluations reinforce our confidence in the broad applicability and adaptability of our proposed approaches.**
>
> ---
>
> > **[Q1]** How do the authors **compare with quantization, sparsification, pruning, or distillation?**
>
> **[A2]** Thank you for pointing out the important points. Such approaches **can be seamlessly integrated with our method (i.e., complementary approaches):** we can have a recursive, “sparse” architecture. However, in this work, **we primarily focus on demonstrating the effectiveness of recursive patterns within “dense” architectures** (a domain that remains relatively unexplored) and providing guidelines on how to initialize or relax sharing constraints.
>
> That said, while **we take the first step at studying Relaxed Recursive Transformer with dense Transformer layers**, we do believe that **incorporating quantization and pruning techniques within the looped blocks are promising directions for future research.**
>
> Moreover, regarding **distillation** methods, we view them as **valuable optimization techniques that can assist in converting to Recursive Transformers**. In fact, we have explored various distillation losses to this end (refer to Table 9 in Appendix L). We have clarified them in the revised paper.

---

> ### Author Response · Authors · 2024-11-25
> **Gentle Reminder to Reviewer uD6H**
>
> Dear Reviewer uD6H,
>
> Thank you once again for the time and effort you’ve dedicated to reviewing our paper.
>
> As the discussion period is coming to a close, we kindly remind you that there are two days remaining for any further comments or questions. We hope that our responses so far have clarified misleading points and addressed your concerns.
>
> If there are any remaining points or feedback you’d like us to consider, we’d be very grateful for the chance to discuss them before the deadline.
> Thank you again for your valuable insights and support throughout this process.
>
> Warm regards,
> Authors

---

> ### Author Response · Authors · 2024-11-28
> **Kind Reminder to Reviewer uD6H**
>
> Dear Reviewer uD6H,
>
> Thank you again for your time and effort in reviewing our paper. We greatly appreciate your valuable feedback and suggestions.
>
> We would like to gently remind you that the discussion period is coming to a close.
>
> In our rebuttal, we addressed your concerns by:
> - **Discussing scalability on larger models,**
> - **Discussing compatibility with sparsity-based approaches.**
>
> If you have any remaining concerns, please do not hesitate to share them with us. We are more than willing to address them promptly.
>
> Thank you very much for your consideration.
>
> Best regards, Authors

---

> > ### Comment · Reviewer_uD6H · 2024-11-29
> >
> > Dear authors，
> >
> > Thank you for the response on my questions. I appreciate your work and will keep my score. I am looking forward to its application on larger scale.

---

> ### Author Response · Authors · 2024-11-29
>
> Thank you again for your valuable feedback, and we appreciate your positive assessment of our work. We hope to demonstrate the scalability of Recursive Transformers in future research.
>
> Best, Authors

---

### Official Review · Reviewer_9eTH · 2024-11-04

**Soundness:** 3
**Presentation:** 3
**Contribution:** 4
**Rating:** 6
**Confidence:** 3

**Summary:**

The paper presents an innovative approach to parameter efficiency in Large Language Models through Recursive Transformers, by iteratively reusing a single block. The concept is extended to Relaxed Recursive Transformers by incorporating layer-specific LoRA modules, introducing flexibility to the parameter-sharing scheme. The authors demonstrate the conversion of pre-trained LLMs into more compact recursive versions while maintaining performance and achieving improved throughput through depth-wise batching.

**Strengths:**

- The core concept of recursive layer sharing combined with LoRA-based relaxation is straightforward yet powerful.
- The paper provides extensive empirical validation across multiple model architectures, recursive layer initialization strategies, LoRA initialization strategies, and evaluation metrics, including thorough ablation studies and practical deployment.
- The authors not only show the benchmarking results of Relaxed Recursive Transformers but also explore hypothetical training speedup with continuous depth-wise batching strategy and early-exiting.

**Weaknesses:**

**Major**
- It compares to different-sized dense baselines. However, the evaluation lacks comparison with sparse designs, particularly layer-skipping mechanisms, and layerwise parameter-sharing approaches [1-2].
- The paper does not fully investigate how recursive architectures affect model training dynamics and optimization landscapes. The relationship between shared parameters and gradient flow during the uptraining phase. Also, whether shared parameter amounts affect model learning capacity could be explored.
- The study would benefit from experiments on optimal recursion depths across varying model scales and task domains.

**Minor**
- The experiments are primarily conducted on relatively small models with 1-2B parameters. The effectiveness and scalability of the approach on larger models remain unproven. This raises questions about whether the benefits would hold at scales where current LLMs demonstrate more sophisticated capabilities.
- Typos: "achieved a 10%p performance" on page6.

[1] Mixture-of-Depths: Dynamically allocating compute in transformer-based language models

[2] Zamba: A Compact 7B SSM Hybrid Model

**Questions:**

- The choice of LoRA for relaxing parameter sharing appears somewhat arbitrary. And prefix-tuning results show limiting performance gain. Could you elaborate more on the relaxation technique selections and other alternatives (*e.g.* Adapter, IA3)?
- For Section 3.8, how would the throughput gains hold up when using practical confidence-based early-exiting rather than the idealized oracle approach that assumes perfect prediction matching?

---

> ### Author Response · Authors · 2024-11-21
>
> We thank the reviewer for their careful review and thoughtful comments. We appreciate that the reviewer finds the core concept of our work to be straightforward yet powerful, and that our empirical and theoretical validation of Recursive Transformers are comprehensive. We provide responses to the reviewer’s specific questions and concerns (quoted) below.
>
> ---
>
> > **[W1]** Lack of **comparison with sparse designs** (layer-skipping mechanisms like Mixture-of-Depth [1]) and **layerwise parameter-sharing** (Zamba [2]).
>
> **[A1]** Thank you for raising these discussion points. Certainly, other sparsity-based approaches can also give good model compression results. In fact, **many of these techniques are complementary to our approach: for example, we can seamlessly have a recursive, sparse architecture.** In this work, we choose to focus on recursive dense designs (a domain that remains relatively unexplored) that also have very promising, practical performance traits (i.e., allowing for continuous depth-wise batching for faster throughput). This also allows us to focus our analysis on the different initialization and relaxation strategies within a set parameter sharing recipe, while allowing for meaningful direct comparisons.
>
> That said, while in this work **we take the first step at studying Relaxed Recursive Transformer with dense Transformer layers**, we do believe that **incorporating Mixture-of-expert, layer-skipping and SSM components within the looped blocks are promising directions for future research.** We agree that these works provide valuable context to help clarify our contribution points, so we have included this discussion in the revised version.
>
> ---
>
> > **[W2]** Need to fully investigate **how recursive architectures influence model training dynamics and optimization landscapes**, particularly the relationship between shared parameters and gradient flow. Also the need to explore the **impact of shared parameter amounts on model learning capacity.**
>
> **[A2]** While the scope of our contribution is empirical, previous research [3,4] does suggest that **aggregating gradients from different depths does provide a regularization effect** that can stabilize training and promote generalization.  Moreover, we believe a form of **“self-distillation” occurs within these looped models**. As demonstrated by surprisingly strong performance of intermediate loop outputs in Section 3.7, **intermediate outputs are effectively distilled from the final output, with all gradients backpropagating to the same parameters.**
>
> Sharing parameters with fixed depth does limit model capacity compared to the original model in theory, though it is unclear to what extent this matters in practice.  Empirically, we explored the relationship between the degree of parameter sharing vs. model performance in Table 10 and Figure 17 of the Appendix. These experiments do suggest that **increasing parameter sharing does reduce performance (and it is likely that this is a result of reduced model capacity, as the ablations on the LoRA rank also support)**. This might be improved by keeping the same parameter count, but increasing the number of recursions (i.e., more looping blocks). Further exploration of higher compression ratios remains a promising avenue for future work.
>
> ---
>
> > **[W3]** Experiments on **optimal recursion depths across varying model scales and task domains.**
>
> **[A3]** While model scale and task domain certainly influence the optimal choice of recursion depth, **what is “optimal” ultimately depends on the user’s specific deployment scenario.** Our analysis reveals an inverse relationship between the number of shared blocks and the achievable performance for a given training budget. Therefore, **selecting the appropriate recursion depth requires careful consideration of the desired performance level and the available training resources.**
>
> ---
>
> > **[W4] Scalability of the approach on larger models** remains unproven.
>
> **[A4]** We agree that investigating the performance and speed benefits in larger models is a promising direction for future work. Preserving more sophisticated capabilities **might necessitate a greater number of uptraining steps**. As evidenced by TinyLlama, we also expect that **utilizing the same dataset used for pretraining is best for restoring performance**. Moreover, we are confident that **our findings regarding initialization and relaxation will generalize broadly**, as demonstrated across three distinct architectures and  a diverse range of base model scales.
>
> ---
>
> > **[W5] Typo** in “achieved a 10%p performance” on page 6.
>
> **[A5]** This is not a typo, rather we meant **10 percentage points** performance gain—that is, the absolute difference of two percentages. We have clarified this notation in the paper.

---

> ### Author Response · Authors · 2024-11-21
>
> > **[Q1]** **Elaborate more on the relaxation technique selection and other alternatives,** such as Adapter and IA3.
>
> **[A6]** The choice of LoRA for relaxation is not arbitrary, instead **it was selected for its efficiency in relaxation and compatibility with batched inference**. Especially, unlike Adapters [5] and IA3 [6], **LoRA's parallel attachment structure facilitates efficient serving**. As discussed in Appendix A, LoRA modules allow for the utilization of optimized CUDA kernels employed in multi-task learning and parallel computation techniques from MoE distributed training systems.
>
> While **Prefix Tuning**'s limited trainable parameters reduced its efficacy, **its superior compatibility with batched inference beyond LoRA motivated its exploration** (refer to Appendix K). The revised version includes further discussions on relaxation alternatives, such as layer-specific bias and various Adapter and Prefix Tuning variants.
>
> ---
>
> > **[Q2]** How would the **throughput gain hold up when using practical confidence-based early-exiting?**
>
> **[A7]** Thank you for a good question. We included a discussion on this topic in Appendix A. Exploiting practical confidence-based algorithms **might yield lower throughput gains than theoretically predicted**. This discrepancy could arise from inaccuracies in confidence estimation, the overhead of the estimation process itself, and the unaccounted time for missing KV cache computations for exited tokens in the upper layers during simulation.
>
> To enhance efficiency, **we've elaborated potential solutions in Appendix A**. For instance, the missing KV cache computations can be addressed by leveraging continuous depth-wise batching, allowing the KV cache for exited positions in subsequent loops to be performed in parallel with the computations for the next sequence sample.
>
> ---
>
> **References:**
>
> [1] Raposo, et al. “Mixture-of-Depths: Dynamically allocating compute in transformer-based language model.” arXiv 2024.
>
> [2] Glorioso, et al. “Zamba: A compact 7B SSM hybrid model.” arXiv 2024.
>
> [3] Dehghani, et al. “Universal Transformers.” ICLR 2019.
>
> [4] Bai, et al. “Deep Equilibrium Models.” NeurIPS 2019.
>
> [5] Houlsby, et al. “Parameter-Efficient Transfer Learning for NLP.” ICML 2019.
>
> [6] Liu, et al. “Few-Shot Parameter-Efficient Fine-Tuning is Better and Cheaper than In-Context Learning.” NeurIPS 2022.

---

> ### Author Response · Authors · 2024-11-25
> **Gentle Reminder to Reviewer 9eTH**
>
> Dear Reviewer 9eTH,
>
> Thank you once again for the time and effort you’ve dedicated to reviewing our paper.
>
> As the discussion period is coming to a close, we kindly remind you that there are two days remaining for any further comments or questions. We hope that our responses so far have clarified misleading points and addressed your concerns.
>
> If there are any remaining points or feedback you’d like us to consider, we’d be very grateful for the chance to discuss them before the deadline.
> Thank you again for your valuable insights and support throughout this process.
>
> Warm regards,
> Authors

---

> ### Author Response · Authors · 2024-11-28
> **Kind Reminder to Reviewer 9eTH**
>
> Dear Reviewer 9eTH,
>
> Thank you again for your time and effort in reviewing our paper. We greatly appreciate your valuable feedback and suggestions.
>
> We would like to gently remind you that the discussion period is coming to a close.
>
> In our rebuttal, we addressed your concerns by:
> - **Discussing compatibility with sparsity-based approaches,**
> - **Discussing how the recursion influences model training dynamics,**
> - **Explaining that optimal depths ultimately depends on the user’s deployment scenario,**
> - **Discussing scalability on larger models,**
> - **Elaborating the rationale on the choice of LoRA modules,**
> - **Explaining potential solutions for limitations encountered when using confidence-based early-exiting.**
>
> If you have any remaining concerns, please do not hesitate to share them with us. We are more than willing to address them promptly.
>
> Thank you very much for your consideration.
>
> Best regards, Authors

---

> > ### Comment · Reviewer_9eTH · 2024-11-29
> >
> > Thank you for the detailed explanation. Given the positive initial assessment, I will maintain my score. This work presents an interesting empirical investigation into recursive LLM design. I look forward to future work that addresses enhanced scalability and explores interactions with other efficient model architectures.

---

> ### Author Response · Authors · 2024-11-29
>
> We appreciate your positive assessment of our work. We acknowledge that enhancing scalability and exploring incorporation with other efficient architecture are important future works, and we firmly believe that recursive LLM holds significant potential for achieving these goals.
>
> Thanks again for your valuable comments.
>
> Best, Authors

---

### Official Review · Reviewer_WB6T · 2024-11-05

**Soundness:** 3
**Presentation:** 3
**Contribution:** 3
**Rating:** 6
**Confidence:** 4

**Summary:**

The paper introduces a new approach to reducing the deployment costs of large language models by implementing recursive parameter sharing. The proposed Recursive Transformer model reuses a single block of unique layers across multiple iterations, effectively minimizing model size while maintaining strong performance. To enhance flexibility, Relaxed Recursive Transformers incorporate low-rank adaptation (LoRA) modules at each layer, allowing for slight layer-specific modifications that improve performance with minimal overhead. Additionally, the authors propose a Continuous Depth-wise Batching technique, which leverages the recursive architecture to increase inference throughput by enabling multiple depths to be processed simultaneously, achieving up to a 2-3× improvement in efficiency over traditional models.

**Strengths:**

1. Propose a novel model structure that can effectively compress pre-trained large language models for deployment.
2. LoRA adaptation makes the model structure more flexible to adapt to different depths.
3. Depth-wise Batching translates the advantages of the proposed recursive transformer block to real deployment acceleration.

**Weaknesses:**

1. Experiment design is not so convincing, which can be further improved.
2. Authors do not push the recursive computation block’s advantage to the limit, which can be interesting to explore
3. Some highly-related papers are missing in the reference

**Questions:**

1. In experiment design (Line 268), authors mention “As a result, we set the target performance for Gemma and Pythia models as the performance achieved by uptraining a full-size pretrained model with an equivalent number of tokens.”. This design is unreasonable, especially since your training (SlimPajama) and testing datasets (llm harness evaluation) are different. You need to provide the original LLM’s performance to help readers understand the real performance of the proposed recursive transformer instead of fine-tuning a worse model to mislead readers.
2. To test the effectiveness of your model compression method, it is highly recommended to achieve a larger compression ratio. E.g. compressing 7B model to 1B. Current experiments only compress 2B models to 1B, which lacks practical meaning as they are all models with similar sizes.
Reusing the transformer computation block has been explored before [1] without properly cited within this paper.

[1] Liu, Zechun, Changsheng Zhao, Forrest Iandola, Chen Lai, Yuandong Tian, Igor Fedorov, Yunyang Xiong et al. "Mobilellm: Optimizing sub-billion parameter language models for on-device use cases." arXiv preprint arXiv:2402.14905 (2024).

---

> ### Author Response · Authors · 2024-11-21
>
> We thank the reviewer for their careful review and thoughtful comments. We appreciate that the reviewer finds our work to be novel, flexible, and translatable to real deployment advantages. We provide responses to the reviewer’s specific questions and concerns (quoted) below.
>
> ---
>
> > **[W1]** The experiment design is not convincing, especially **setting full-size model baselines of Gemma and Pythia to uptrained original models** with an equivalent number of tokens. You need to provide the original LLM’s performance instead.
>
> **[A1]** Thank you for pointing out this potentially confusing aspect. The only uptraining dataset we use is the publicly available SlimPajama dataset (which is the same pretraining dataset for TinyLlama, but not for Pythia or Gemma). As shown in Table 4 of Appendix H, finetuning the original Gemma and Pythia models on SlimPajama leads to slight performance *drops* on certain few-shot tasks, indicating that **using SlimPajama lowers the upper bound of achievable performance for Recursive Gemma and Pythia models**.
>
> In order to disentangle the effects of using (a) *an inferior uptraining dataset* vs. (b) *a recursive architecture*, we therefore believe that using the full-sized Gemma and Pythia models uptrained on SlimPajama is the fairest comparison point. On the other hand, for TinyLlama, where the pretraining data of the original LLM matches the uptraining dataset, it is appropriate to use the original LLM performance as the primary baseline.
>
> In order to openly share the full details of the experimental setup, we have also included the detailed results of all full-size models both before and after uptraining in Table 4 of Appendix H. It is also useful to note that the overall effects of uptraining the base model are fairly small: average Gemma performance drops by 3.1 absolute percentage points, Pythia drops by only 0.2 absolute percentage points, and TinyLlama improves by 4.1 absolute percentage points. We have further clarified this point in the revised paper.
>
> ---
>
> > **[W2]** **Pushing the recursive computation block’s advantage to the limit (i.e., larger compression ratio)** can be interesting to explore. Current experiments only compress 2B models to 1B.
>
> **[A2]** We have also conducted experiments with **Recursive Gemma models using 3 blocks** (refer to **Table 10 and Figure 17** for their performance, and **Tables 15 and 17** for their throughput.) The proposed initialization and training methods demonstrated consistent trends even with three blocks. Moreover, we observed the following trends:
>
> - **(1) Performance degradation with 3 blocks was less significant than anticipated.** In fact, in some cases, the relaxed version of the 3-block model outperformed the 2-block models with similar parameter counts.
> - **(2)** Models with 3 loops exhibited a **higher likelihood of exiting at earlier loops, leading to a more favorable trade-off between performance and decoding speed.**
>
> A more thorough exploration of higher compression ratios (and with larger original models / upper bound performance) is left for future work. Nevertheless, as the robustness of the proposed method has been validated across various settings, we believe that the number of recursion blocks can be adjusted appropriately to suit specific deployment scenarios, such as device constraints, computational cost limits, or desired performance levels.
>
> ---
>
> > **[W3] Missing reference: MobileLLM [1].**
>
> **[A3]** Thank you for bringing this reference to our attention!. We **have included a discussion of MobileLLM [1]**, as well as other prior works that other reviewers have flagged (i.e., **EdgeFormer [2]** for our relaxation methods, and **Zamba [3]** for cross-layer parameter sharing in a SSM hybrid model), in the revised version of the paper.
>
> ---
>
> **References:**
>
> [1] Liu, et al. "Mobilellm: Optimizing sub-billion parameter language models for on-device use cases." ICML 2024.
>
> [2] Ge, et al. “EdgeFormer: A parameter-efficient transformer for on-device seq2seq generation.” EMNLP 2022.
>
> [3] Glorioso, et al. “Zamba: A compact 7B SSM hybrid model.” arXiv 2024.

---

> ### Author Response · Authors · 2024-11-25
> **Gentle Reminder to Reviewer WB6T**
>
> Dear Reviewer WB6T,
>
> Thank you once again for the time and effort you’ve dedicated to reviewing our paper.
>
> As the discussion period is coming to a close, we kindly remind you that there are two days remaining for any further comments or questions. We hope that our responses so far have clarified misleading points and addressed your concerns.
>
> If there are any remaining points or feedback you’d like us to consider, we’d be very grateful for the chance to discuss them before the deadline.
> Thank you again for your valuable insights and support throughout this process.
>
> Warm regards,
> Authors

---

> ### Author Response · Authors · 2024-11-28
> **Kind Reminder to Reviewer WB6T**
>
> Dear Reviewer WB6T,
>
> Thank you again for your time and effort in reviewing our paper. We greatly appreciate your valuable feedback and suggestions.
>
> We would like to gently remind you that the discussion period is coming to a close.
>
> In our rebuttal, we addressed your concerns by:
> - **Clarifying the rationale for full-size model baselines,**
> - **Detailing the findings from experiments with various looping blocks,**
> - **Including the missing references.**
>
> If you have any remaining concerns, please do not hesitate to share them with us. We are more than willing to address them promptly.
>
> Thank you very much for your consideration.
>
> Best regards, Authors

---

> ### Author Response · Authors · 2024-12-02
> **Kind Reminder to Reviewer WB6T**
>
> Dear Reviewer WB6T,
>
> This is a friendly reminder that the discussion period for the review ends tomorrow.
>
> We hope our response and the revised manuscript have addressed your concerns. We would appreciate it if you could confirm whether any further clarification is needed. We are happy to answer any remaining questions before the discussion period closes.
>
> If there are no further questions, we would be grateful if you could reflect this in your re-evaluation.
>
> Thank you once again for your time and consideration.
>
> Best regards, Authors

---

### Author Response · Authors · 2024-11-21

Thank you all for taking the time and effort to review our paper and provide thoughtful and constructive feedback. We have carefully addressed each reviewer's comments and uploaded a revised version of the paper, which includes the following additions:

- **Clarification of baseline models:** We have further clarified the rationale behind the full-size and reduced-size non-recursive baselines in Section 3.2.
- **Discussion of sparse designs:** Appendix A now includes a discussion of how sparsity-based approaches (pruning, quantization, and layer-skipping) compare to our proposed methods and how they might be integrated with our structures.
- **Analysis of individual effects in Recursive Transformer:** We conducted additional experiments on Pythia 410M to isolate and demonstrate the individual effects of leveraging pretrained layers and recursive patterns (see Appendix O).
- **Evaluation of initialization methods in other model scales:** To confirm the robustness of our proposed initialization methods, we expanded our experiments to include two different model scales---Pythia 410M and 160M (see Appendix P).
- **Missing references:** We have added missing references related to parameter sharing and relaxation techniques.

Thank you again for your valuable feedback.  We appreciate your time and insights. If you have any further questions or concerns, please feel free to share them. We are happy to engage in further discussions.

---

### Meta-Review · Area_Chair_JjUy · 2024-12-21

**Metareview:**

The paper proposes relaxed recursive transformers, a new approach to parameter sharing in LLMs by iteratively reusing a single block of layers. The incorporation of LoRA enhances model flexibility without significant performance loss. Additionally, the paper introduces continuous depth-wise batching for improved inference throughput. Reviewers praised the innovative combination of recursive architecture and LoRA, as well as the comprehensive experiments. I recommend acceptance.

**Additional Comments On Reviewer Discussion:**

Some concerns were raised about scalability to larger models and limited evaluation of diverse tasks by reviewers, the rebuttal addressed these issues effectively with additional experiments, clarifications, and extended analysis.

---

### Decision · Program_Chairs · 2025-01-22

Accept (Poster)